# Comprehension Without Competence: Architectural Limits of LLMs in Symbolic Computation and Reasoning*

**Zheng Zhang**†                                                                            *zzhang@gmail.com*

**Reviewed on OpenReview:** *https://openreview.net/forum?id=Gz5HMiJLqv*

## Abstract

Large Language Models (LLMs) display striking surface fluency yet systematically fail at tasks requiring symbolic reasoning, arithmetic accuracy, and logical consistency. This paper offers a structural diagnosis of such failures, revealing a persistent gap between *comprehension* and *competence*. Through controlled experiments and architectural analysis, we demonstrate that LLMs often articulate correct principles without reliably applying them—a failure rooted not in knowledge access, but in computational execution. We term this phenomenon the computational *split-brain syndrome*, where instruction and action pathways are geometrically and functionally dissociated. This core limitation recurs across domains, from mathematical operations to relational inferences, and explains why model behavior remains brittle even under idealized prompting. We argue that LLMs function as powerful pattern completion engines, but lack the architectural scaffolding for principled, compositional reasoning. Our findings delineate the boundary of current LLM capabilities and motivate future models with metacognitive control, principle lifting, and structurally grounded execution. This diagnosis also clarifies why mechanistic interpretability findings may reflect training-specific pattern coordination rather than universal computational principles, and why the geometric separation between instruction and execution pathways suggests limitations in neural introspection and mechanistic analysis.

## 1 Introduction

When asked whether 9.9 is greater than 9.11, Claude Sonnet 4 confidently states that 9.11 is larger, explaining that "since 90 is greater than 11 in the hundredths place, 9.11 is the larger number"—while simultaneously calculating that 9.11 - 9.9 = 0.21 (incorrect). GPT-4o claims that "9.11 is greater than 9.9" in mathematical contexts, invoking software versioning to justify the confusion. Yet when asked to explain how to compare decimal numbers, both models provide flawless algorithmic descriptions: "Write the numbers one above the other, aligning the decimal points... Compare digit by digit from left to right." Claude Sonnet 4 even works through the exact same comparison correctly as an instructional example, concluding that "9.90 > 9.11." Both models articulate decimal comparison procedures with textbook precision while failing to execute them reliably.

This disconnect reveals a fundamental limitation in Large Language Models: they exhibit *comprehension without competence*—a systematic dissociation where models can perfectly explain principles they cannot reliably execute. We term this phenomenon *computational split-brain syndrome*[1], drawing analogy to neurological conditions where different brain systems cannot coordinate effectively. Like patients who can verbally describe actions they cannot perform, LLMs develop geometrically separated pathways for "knowing about" procedures versus "executing" them.

---

*Code repository: `https://github.com/zzhang-cn/comprehension-without-competence`. Video presentation: `https://youtu.be/Z4waLOGwhyQ`

†This work was completed as a personal research project while employed at Amazon Web Services. The views expressed are those of the author and do not necessarily reflect those of Amazon.

[1]The term also appears in distributed systems to describe partitioned networks that cannot coordinate (Brewer, 2000)—we use the neuroscience analogy as it better captures the functional dissociation between capabilities.

Here, comprehension and competence are understood behaviorally—when an LLM flawlessly explains decimal comparison algorithms, observers reasonably judge it as "comprehending" the procedure, just as we assess human understanding through explanation ability. Yet the model cannot reliably execute what it explains. Both capabilities emerge from pattern completion but as geometrically separated, uncoordinated pathways— explanation without execution. This represents an inversion of Dennett's framework where competence typically precedes comprehension in biological systems (Dennett, 2017).

This split-brain syndrome manifests most visibly as *computational hallucination*—when models must execute multi-step algorithms they cannot actually perform, they generate plausible-sounding answers through pattern completion rather than computation. Unlike hallucinations that contradict facts or context, computational hallucination emerges directly from architectural impossibility: the model compares 9.11 with 9.9 not through arithmetic but by pattern-matching what such calculations typically look like. Every task requiring symbolic manipulation—from arithmetic to logical inference to data operations—becomes a source of systematic hallucination, with models confidently producing answers that seem reasonable but lack computational grounding.

This is not a limitation of scale, training data, or prompting techniques. We argue that computational split-brain syndrome stems from structural constraints in how LLMs represent symbols, learn procedures, and execute reasoning steps. This dissociation challenges fundamental assumptions about intelligence derived from human cognition, where explanatory fluency typically correlates with execution competence. LLMs systematically violate this expectation, revealing that artificial and biological intelligence may operate under fundamentally different principles. Unlike symbolic systems that bind tokens to abstract roles and apply rules over those bindings, LLMs operate as pattern completion engines optimized for next-token prediction.

**Main Claims of This Work.** Our analysis establishes this disconnect through three interconnected claims:

- **Claim 1 (Representation)**: LLM token embeddings encode context-weighted averages that systematically resist automatic domain binding, preventing stable symbolic circuits across computational domains.

- **Claim 2 (Computation)**: Feed-forward networks face architectural impossibility of implementing exact symbolic operations through weight configuration alone, forcing them to resort to residual pattern fitting rather than implementing generalizable symbolic procedures.

- **Claim 3 (Instruction-Execution Disconnect)**: Next-token prediction objectives decouple algorithmic descriptions from executable behavior.

These three constraints must be understood together to explain the computational split-brain syndrome:

Claims 1 & 2 (Execution failures): When encountering execution instances (e.g., '9.11 > 9.9?'), contextual averaging prevents clean mathematical binding (Claim 1). Even if binding were perfect, transformers face an architectural bottleneck: attention identifies operations and operands but cannot generate new values, forcing FFNs to produce results—yet FFNs cannot implement exact operations, only pattern approximations (Claim 2). Together, these establish that reliable execution is impossible.

Claim 3 (The disconnect): Next-token prediction provides no mechanism for well-learned instructions to guide execution. Both capabilities emerge as separate pattern-matching pathways with no automatic binding between them. The model thus can explain perfectly yet fail at execution—the split-brain syndrome.

Importantly, we demonstrate that these constraints manifest consistently across two critical domains: symbolic computation (arithmetic operations) and relational reasoning (logical inference and variable binding). In both domains, models exhibit the same split-brain syndrome—fluent explanation coupled with unreliable execution—revealing this as limitation of the architecture-training paradigm combination rather than a domain-specific failure.

We now face a fundamental design trade-off. By reducing all language processing to a single continuous optimization task (next-token prediction) over unified token embeddings, transformers achieve their remarkable

generality and fluency. However, this same unification prevents the type-based reasoning and domain-specific binding that symbolic computation requires. The contextual averaging that enables rich semantic understanding also contaminates mathematical symbols; the single objective that scales so effectively treats instruction and execution as indistinguishable patterns. Understanding this trade-off clarifies why the computational split-brain syndrome persists across model families and scales: it emerges from the very design choices that make LLMs powerful.

**Contributions.** Our analysis establishes computational split-brain syndrome as a unifying framework for understanding when and why LLMs fail at systematic reasoning. We provide empirical evidence for each architectural constraint through embedding analysis, layer-by-layer computation tracking, and systematic evaluation across arithmetic and logical tasks. We then examine three compensatory strategies—self-scaffolding, tool delegation, and hybrid architectures—showing how each leverages LLMs' pattern completion strengths while working around their execution limitations, yet all require metacognitive capabilities that current architectures fundamentally lack.

Our findings suggest that mechanistic interpretability studies identifying "arithmetic circuits" or "adder neurons" may be observing sophisticated forms of pattern matching coordination that are formed *path-dependently* during learning, rather than genuine computational subroutines. More critically, the geometric dissociation between instruction and execution pathways raises concerns about both model self-explanations and interpretability research: models may articulate reasoning procedures through neural routes distinct from those used for actual computation, potentially misleading both self-monitoring and researcher analysis of neural activations. Section 7 extends this analysis to modern interpretability methods, examining whether Sparse Autoencoders and feature-level decomposition escape these limitations or merely surface more stable statistical patterns.

These constraints appear unavoidable within the current paradigm: contextual averaging emerges inevitably from next-token prediction over diverse corpora, while predicting next tokens on internet-scale data forces FFNs to work with attention layers into hierarchical pattern assembly rather than principled computation, and instruction-execution disconnect results from treating all text as equivalent prediction targets. This suggests that computational split-brain syndrome will persist across model families and scales unless addressed through fundamental architectural innovations rather than incremental improvements.

**Roadmap.** Section 2 reviews foundational work on transformer computation, symbolic reasoning, and recent interpretability findings that inform our analysis. In Section 3, we investigate LLM failures in symbolic computation, focusing on arithmetic tasks that reveal unstable embeddings and residual fitting behavior. Section 4 extends this analysis to relational reasoning, demonstrating that the same architectural bottlenecks underlie LLM failures in multi-step inference, variable binding, and logical consistency. Section 5 examines compensatory approaches—self-scaffolding, tool delegation, and hybrid architectures—revealing how all three strategies converge on the same reliable metacognitive limitations that current architectures lack. Section 6 analyzes LLMs as hierarchical pattern completion engines, distinguishing between general intelligence (which LLMs achieve through sophisticated pattern matching) and generalizable intelligence (which requires systematic rule discovery and principled reasoning), and examining the performance cliffs that emerge when tasks transition from pattern recognition to genuine algorithmic discovery. Section 7 explores how our framework applies to mechanistic interpretability research, examining whether current methods like Sparse Autoencoders identify genuine computational mechanisms or statistical patterns, and proposing tests to distinguish between these possibilities. Section 8 offers a brief reflection on the research journey that led to these insights.

**Limitations.** This analysis focuses on pretrained transformer-based LLMs trained on next-token prediction objectives over natural language corpora, without access to external tools or explicit reasoning scaffolds. We analyze these "pure" systems to understand why compensatory strategies have become essential—revealing not just what fails, but why tool use, scaffolding, and hybrid architectures represent necessary workarounds rather than optional enhancements for fundamental architectural constraints. Our empirical evaluation centers on specific model families (LLaMA2, Claude, GPT-4) and may not generalize to fundamentally different architectures or training paradigms. The geometric separation experiments rely on t-SNE projections which

may not capture all relevant representational structure. Our theoretical analysis of FFN computational limits focuses on ReLU networks and may not apply to other activation functions. While recent architectural variations like mixture-of-experts or reasoning-augmented models may improve performance, they operate within the same fundamental transformer framework and thus likely face similar constraints—though empirical validation of this hypothesis remains future work.

## 2 Related Work

This work builds on several lines of inquiry across theory, empirical analysis, architecture, and cognitive science. We organize our review around the emergence, diagnosis, and mitigation of systematic reasoning failures in LLMs, providing a unified framework for understanding when and why current architectures fall short of reliable symbolic computation.

### 2.1 Foundations and Theoretical Limits

**Expressivity and Computational Boundaries.** Recent theoretical frameworks have clarified the fundamental boundaries of transformer expressivity. Peng et al. (2024) use communication complexity to prove that transformer layers cannot compose functions (e.g., identifying grandparents in genealogies) when domains are sufficiently large. Strobl et al. (2024) provide a comprehensive survey documenting how transformers function as recognizers of formal languages, revealing that while scaffolding strategies can improve performance within bounded expressivity classes, they remain fundamentally constrained by architectural limitations rather than achieving true symbolic reasoning capabilities. The Simons Institute workshop (Simons Institute for the Theory of Computing, 2024) established that transformers act as highly parallel pattern matching engines with limited capacity for long-range sequential computation, constraining their theoretical expressivity in structured symbolic tasks.

However, Li et al. (2024b) prove that scaffolding strategies, techniques in which models generate intermediate steps to guide reasoning, can theoretically expand computational power: While constant-depth transformers can only solve problems in $\mathsf{TC}^0$, adding $T$ intermediate steps enables them to solve any problem solvable by Boolean circuits of size $T$. Chain-of-thought prompting (Wei et al., 2022) exemplifies this approach: instead of directly computing '$23 \times 17 = ?$', models generate '$23 \times 10 = 230, 23 \times 7 = 161, 230 + 161 = 391$', converting parallel constraints into serial computation. These analyses can be seen as the theoretical foundation for the use of self-scaffolding as a principled workaround (see Section 5). Recent large reasoning models such as o1 and DeepSeek-R1 (OpenAI, 2024; DeepSeek-AI et al., 2025) show improved performance by essentially scaffolding their thought processes at inference time. Yet, they remain fundamentally compensatory rather than curative—they bypass core architectural bottlenecks rather than resolving them.

**Training Dynamics and Path Dependence.** Understanding how reasoning capabilities emerge during training reveals critical insights into LLM limitations. Power et al. (2022) demonstrate that late-phase generalization involves a fundamental reorganization of internal representations, showing how training dynamics create path-dependent artifacts that appear as reasoning capabilities but reflect statistical regularities rather than systematic algorithms. Tigges et al. (2024) extend this analysis to moderate scale LLMs with decoder only (up to 2.8B parameters), finding that while computational circuits emerge consistently across scale, their specific implementations vary significantly during training. We interpret these findings as suggesting that apparent "reasoning circuits" may reflect training-specific pattern coordination rather than universal computational principles, although this inference requires further validation through controlled experiments with different training permutations.

Recent mechanistic analysis of factual learning provides direct evidence for these training-dependent phenomena. Zucchet et al. (2025) demonstrate that the acquisition of knowledge in transformers follows three distinct phases: general statistics learning, a plateau phase where attention circuits develop, followed by acquisition of individual-specific knowledge. Critically, they show that adding new factual knowledge rapidly corrupts existing memories stored in feedforward layers, confirming that apparent knowledge storage reflects fragile statistical coordination rather than stable representation systems.

## 2.2 Empirical Failures Across Domains

**Symbolic Computation Breakdowns.** Systematic evaluation reveals consistent failures in symbolic computation despite surface fluency. Srivastava et al. (2024) quantify sharp performance drops when problem structures deviate from training distributions. Mirzadeh et al. (2024) demonstrate through the GSM-Symbolic benchmark that LLMs exhibit noticeable variance when only numerical values change, with performance drops up to 65% when irrelevant clauses are added. This fragility confirms that current LLMs cannot perform genuine logical reasoning and instead replicate reasoning patterns from training data.

Dziri et al. (2023) provide comprehensive analysis showing that transformers solve compositional tasks through "linearized subgraph matching"—memorizing computation patterns rather than learning systematic algorithms. Their frequency analysis reveals that models succeed primarily when test computation subgraphs appeared frequently in training data, failing dramatically on out-of-distribution examples. Nikankin et al. (2024) confirm this pattern-dependent behavior, showing that models rely on a "bag of heuristics" for arithmetic rather than implementing systematic algorithms.Wu et al. (2025) further demonstrate that domain-specific fine-tuning can increase answer accuracy while *degrading* the logical coherence of chain-of-thought explanations, reinforcing the comprehension–execution split.

Multiple recent studies provide converging evidence for the execution limitation across different domains. Yang et al. (2024) show that LLMs consistently outperform chain-of-thought when generating Prolog programs for external execution rather than attempting direct calculation. The Illusion of Thinking commentary (Opus and Lawsen, 2025) demonstrates similar improvements when LLMs generate Lua code for Tower of Hanoi problems rather than attempting direct move enumeration. Wolff and Hulsebos (2025) reveal significant deficits when LLMs attempt tabular reasoning tasks requiring multi-step algorithms such as computing averages or finding maximum values—operations that would be trivial for database systems. Cheng et al. (2024) provide systematic evidence that LLMs excel at pattern recognition and code generation while struggling with deductive reasoning, with their successful framework architecturally separating symbolic code generation from symbolic code execution. This convergent pattern across arithmetic reasoning, logic puzzles, tabular data operations, and systematic inference demonstrates that LLMs excel at extracting predicates and generating symbolic representations while requiring external systems for reliable computation, precisely the instruction-execution disconnect we formalize.

**Relational and Logical Reasoning Failures.** The same architectural constraints manifest in relational reasoning. Berglund et al. (2023) identify the "reversal curse," where models fail to infer bidirectional relations from symmetric facts due to asymmetric training exposure. Nezhurina et al. (2024) show similar failures in multi-step family reasoning problems that should be trivial for systematic logical processing.

A controlled biography-corpus study by Allen-Zhu et al. (Allen-Zhu and Li, 2024) reinforces the same point: a GPT-2 model trained *exclusively* on person–relation tuples and matching QA pairs achieves near-perfect in-distribution accuracy, yet its OOD QA accuracy on unseen entities collapses below 10% *unless* the pre-training data are aggressively rewritten, permuted, or translated. This highlights that relational exposure alone is insufficient for lifted reasoning; surface-form augmentation is required to abstract away from individual instances.

Li et al. (2024a) provide systematic evaluation across inductive logic programming tasks, finding that LLMs—despite being 100,000 times larger than specialized logical reasoning models—perform dramatically worse on tasks requiring variable binding and systematic rule application. This performance gap persists even when given explicit logical structure through truth-value matrices, definitively ruling out training data scarcity as an explanation and confirming architectural limitations.

## 2.3 Mechanistic Explanations

Recent mechanistic interpretability research has developed sophisticated tools for tracing computational pathways within transformer models. Attribution graph studies by Lindsey et al. Lindsey et al. (2025b) demonstrate methods for mapping how models transform inputs to outputs through intermediate representa-

tional steps, while other circuit analyses reveal consistency patterns across training and scale (Tigges et al., 2024).

**Representational Pathologies.** The instruction-execution disconnect operates at the representational level through systematic failures in symbolic binding. McLeish et al. (2024) demonstrate that altering input geometry alone can dramatically improve simple numerical reasoning, confirming that embedding representations constitute a significant bottleneck for symbolic manipulation. This supports our analysis that contextual averaging prevents stable domain binding required for symbolic manipulation.

**Training-Order Dependencies and Pattern Storage.** Mechanistic interpretability reveals that apparent "reasoning circuits" often reflect *distributed heuristics* rather than principled computation. Nikankin et al. (2024) show that arithmetic behaviour emerges from a "bag of heuristics" spread across layers instead of dedicated algorithmic modules. Tigges et al. (2024) further demonstrate that, even when a high-level algorithm seems stable, the specific attention heads instantiating it drift throughout training, underscoring strong path dependence. Ye et al. (2025) train a 124M-parameter GPT-2 *exclusively* on a procedurally generated grade-school math curriculum and uncover a linearly decodable *dependency-graph* representation that tracks which intermediate quantities depend on which others. We interpret these coherent circuits as artifacts of an unnaturally homogeneous training path; they dissolve once heterogeneous text is introduced, reinforcing our claim that stable algorithmic modules are fragile in realistic open-domain settings.

**Reasoning Faithfulness and Post-Hoc Construction.** Comprehensive analysis of reasoning faithfulness reveals fundamental limitations in how LLMs construct explanations. Plaat et al. (2024) survey extensive evidence that LLM reasoning may be "post-hoc" and "constructed after a conclusion has been found," with larger models showing less faithful reasoning. Chain-of-thought continues to work "even with invalid steps in the reasoning chain," suggesting pattern matching rather than logical execution. This evidence strongly supports our geometric separation hypothesis—models access instructional pathways distinct from computational pathways when generating explanations.

**Sparse Autoencoders and Feature-Level Analysis.** Recent advances in mechanistic interpretability have shifted from neuron-level analysis to feature-level decomposition through Sparse Autoencoders (SAEs) (Bricken et al., 2023; Templeton et al., 2024). These methods extract interpretable features that appear more stable across models than individual neurons (Wang et al., 2025), with techniques like Linear Computation Graphs tracing complete computational pathways through these features (He et al., 2024). While SAEs promise to overcome the polysemanticity and path-dependency limitations of neuron-level analysis, we examine in Section 7 whether these features represent genuine computational mechanisms or merely more stable statistical patterns. Our analysis suggests that even sophisticated feature decomposition cannot escape the fundamental architectural constraints that create the computational split-brain syndrome.

## 2.4 Compensatory Strategies and Architectural Remedies

**Prompt-Level and Self-Scaffolding Interventions.** The effectiveness of compensatory strategies has theoretical grounding in computational complexity. Li et al. (2024b) prove that chain-of-thought fundamentally changes the computational expressiveness of transformers: while constant-depth transformers without CoT can only solve problems in $\mathsf{TC}^0$, adding $T$ steps of CoT enables them to solve any problem solvable by Boolean circuits of size $T$. This provides the theoretical foundation for why self-scaffolding and chain-of-thought prompting partially succeed—they effectively transform parallel computation constraints into serial computation opportunities, allowing transformers to simulate $\mathsf{P/poly}$ with polynomially many intermediate steps. This theoretical result directly explains the empirical success of self-scaffolding as one of the compensatory strategies.

However, Peng et al. (2024) demonstrate that chain-of-thought requires an exponentially growing number of intermediate steps for complex function composition tasks, suggesting that self-scaffolding becomes ineffective when problems demand impractically long reasoning chains. This theoretical prediction aligns with empirical failures we discuss in Section 5.

Beyond standard chain-of-thought (Wei et al., 2022), researchers have proposed metacognitive variants that embed self-evaluation into the reasoning process. AbstRaL (Gao et al., 2025) trains models to first generate an *abstract* symbolic representation of a problem via reinforcement learning before attempting execution, substantially improving robustness under distribution shift. Self-correction frameworks such as SPOC (Zhao et al., 2025) train models to interleave solution generation with explicit verification steps, enabling dynamic revision when the internal verifier detects errors. This exemplifies the self-scaffolding pattern analyzed in Section 5.

**Tool Integration and Hybrid Execution.** Tool-augmented approaches explicitly address the execution gap. Toolformer (Schick et al., 2023) enables automated tool calling, while ReAct (Yao et al., 2022) combines reasoning with external action. Yang et al. (2024) provide empirical validation that this separation—LLMs for predicate extraction, external systems for computation—consistently outperforms end-to-end LLM execution across arithmetic tasks.

**Architectural Modifications.** To address fundamental limitations, researchers have explored hybrid architectures incorporating explicit symbolic modules. OccamLLM (Dugan et al., 2024) and IGC (Dietz and Klakow, 2024) embed arithmetic reasoning units, while Logic-LM (Pan et al., 2023), NLM (Dong et al., 2019), and Differentiable Logic Machines (Zimmer et al., 2023) implement logical operations through structured tensors. These systems represent promising directions for true architectural solutions rather than compensatory workarounds.

## 2.5 Positioning and Contributions

Concurrent work by Lin et al. (Lin et al., 2025) observes a similar disconnect between LLMs' ability to generate symbolic text and execute reliably, framing this as a safety and alignment concern arising from "semiotic disconnects." While we share this empirical observation, understanding *how* these limitations emerge requires identifying the architectural factors that create them. Our work addresses this in three crucial ways.

First, while previous studies document *what* failures occur across domains, we provide a principled mechanistic explanation for *why* they occur through three interconnected architectural constraints: contextual averaging, computational impossibility, and instruction-execution disconnect. Second, rather than treating arithmetic and relational reasoning as separate problems, we demonstrate that identical underlying limitations create the computational split-brain syndrome across all symbolic domains. Third, our geometric analysis reveals that this disconnect operates at the representational level, with direct implications for interpretability research showing why mechanistic studies often surface training-dependent artifacts rather than universal computational principles.

This unified framework not only explains current limitations but predicts why certain compensatory strategies succeed while others fail, providing a foundation for developing more reliable reasoning systems that leverage LLMs' pattern completion strengths while addressing their execution limitations through principled architectural modifications.

# 3 Structural Limits on Symbolic Computation

The computational split-brain syndrome manifests most clearly in symbolic computation, where LLMs can perfectly explain algorithms they cannot reliably execute. This systematic dissociation—drawing analogy to neurological conditions where different brain systems cannot coordinate effectively—emerges as geometric separation in representational space: models develop distinct pathways for "knowing about" procedures versus "executing" them.

This section examines the architectural constraints that prevent reliable symbolic reasoning through a causal chain of three interdependent factors. Importantly, no single constraint alone would be fatal—contextual averaging could be overcome with better training techniques, architectural limitations might be addressed through scaling, and instruction-execution disconnect could potentially be bridged through clever prompt-

ing. However, these three constraints reinforce each other systematically: contextual averaging prevents automatic domain binding (Claim 1), while architectural limitations make direct symbolic computation impossible through weight configuration alone, forcing models toward pattern storage (Claim 2), and next-token prediction treats algorithmic descriptions and execution traces as equivalent pattern-completion tasks, preventing instructional guidance from bridging this gap (Claim 3). Together, these interdependent factors reveal why LLMs function as sophisticated pattern completion engines rather than symbolic reasoners, with implications extending to relational reasoning as we explore in Section 4.

### 3.1   Claim 1: Contextual Averaging Prevents Automatic Domain Binding

Token embeddings in LLMs encode context-weighted averages that resist automatic domain binding, making stable symbolic circuits extremely difficult to form. Unlike traditional programs that explicitly bind variables to types and domains, LLMs derive token meanings from statistical patterns across all training contexts.

**The Failure of Mathematical Binding.**   The root of this problem lies in how LLMs learn token representations. During training, the model encounters "9.11" in countless different contexts—historical articles about September 11th, software version discussions, financial data, and mathematical expressions. The training objective forces the model to find a single embedding that works reasonably well across all these contexts.

Formally, this creates a fundamental tension. The training objective optimizes embeddings through next-token prediction:

$$\mathcal{L}(\theta) = -\sum_t \log p_\theta(w_t \mid w_{<t}) \tag{1}$$

$$e^*(w) = \arg\min_e \sum_{c \in C(w)} \mathbb{E}_{w' \sim p_{\text{data}}(\cdot|c,w)} \left[ -\log p_\theta(w' \mid e,\ E(c \setminus w)) \right] \tag{2}$$

Equation 1 is the familiar next-token prediction loss, where $\theta$ denotes the model parameters, $w_t$ is the token at position $t$, $w_{<t}$ represents all tokens before position $t$, and $p_\theta(\cdot|\cdot)$ is the conditional probability distribution learned by the model. Equation 2 shows how individual token embeddings are optimized: $e^*(w)$ is the optimal embedding for token $w$, $C(w)$ denotes all contexts containing token $w$, $e$ is the candidate embedding being optimized, $E(c \setminus w)$ represents embeddings of all tokens in context $c$ except $w$, and $w'$ is the next token sampled from the data distribution $p_{\text{data}}(\cdot|c,w)$. This optimization produces context-averaged embeddings that trade off among multiple usage modes, similar to distributional semantics models such as Word2Vec (Mikolov et al., 2013).

In formal reasoning, we bind symbols to precise domains ($a = 9.9, a \in \mathbb{R}$), stripping away non-mathematical associations. LLMs typically struggle to perform this domain binding reliably. When encountering "9.9", they tend to preserve contextual associations rather than mapping to a clean mathematical representation.

To demonstrate this contextual contamination, we presented LLaMA2-7B-chat with prompts like "9.11 is a" and "9.9 is a". The completions reveal dramatically different semantic associations:

- For "9.11 is a", the model produces: "day", "date", "remembrance", "tragedy"

- For "9.9 is a", the model produces: "number", "decimal", "perfect", "high"

This divergence shows "9.11" retaining historical event associations while "9.9" behaves as a scalar quantity. Quantitative analysis using negative log-likelihoods across different templates (DATE, VERSION, MEASURE) confirms these contextual biases. The missing operation is domain binding—establishing "9.11" as an element in $\mathbb{R}$ governed by mathematical laws rather than historical associations.

**Failure of Embedding Geometry.**   Symbolically stable circuits require representation spaces that preserve consistent structural relationships. While this doesn't demand strictly Euclidean geometry, it requires

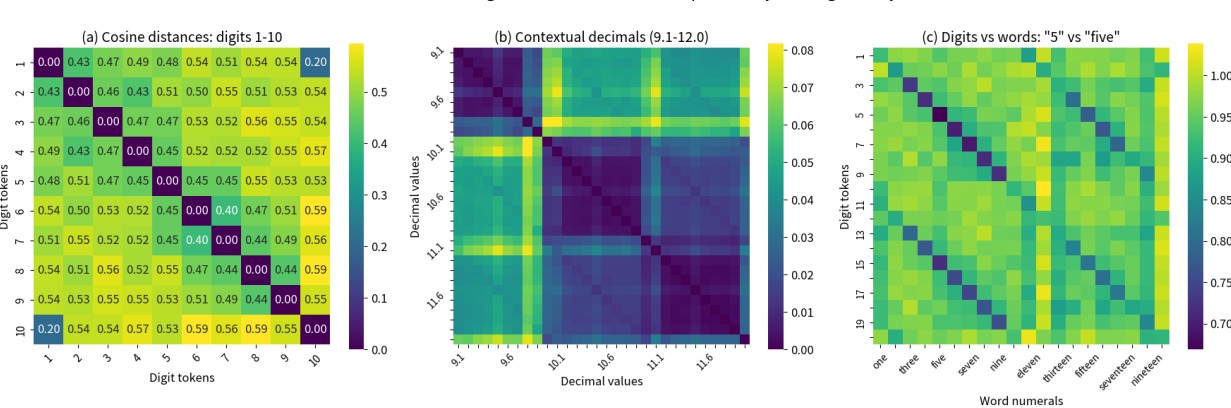

Figure 1: Numeric embeddings in LLaMA2-7B-chat fail to preserve symbolic geometry: (a) irregular cosine distances between digit tokens; (b) bifurcation at "10.0" due to tokenization boundaries; (c) large distances between equivalent representations like "5" and "five".

systematic, predictable structure. As Equation 2 predicts, contextually-averaged embeddings resist the domain binding required for stable symbolic circuits.

The embedding analysis reveals the extent of this geometric chaos (Figure 1 ). What should be the mathematically ordered space of numbers instead resembles a scrambled puzzle, with qualitatively similar failures observed across Qwen and Mistral models:

- **Panel (a):** Cosine distances between digit tokens "1" to "10" show irregular, asymmetric patterns. Notable violations of numerical ordering include "10" being closer to "1" (distance 0.20) than to "9" (distance 0.55), and "6" being closer to "10" (distance 0.59) than to adjacent digits like "7" (distance 0.40).

- **Panel (b):** Decimal embeddings exhibit clear clustering by first digit rather than numerical proximity. The sharp visual boundaries separate 9.x values (top-left dark block), 10.x values (middle diagonal band), and 11.x–12.0 values (bottom-right region), revealing tokenization-driven rather than mathematically-principled organization.

- **Panel (c):** Symbolic equivalents show systematic misalignment in embedding space. Ideally, we should observe a single diagonal of low distances where digit-word pairs ("1"/"one", "2"/"two", etc.) align. Instead, the heatmap reveals fragmented diagonal patterns with significant irregularities. While some digit-word pairs show reasonable proximity (e.g., "5" is relatively close to "five"), the embedding space exhibits systematic failures: a group of higher-numbered words like "ten", "eleven", and "twelve" show poor alignment with any digit tokens, and paradoxically, "5" appears closer to "fifteen" than many other digits are to their correct word equivalents. This demonstrates that the embedding geometry fails to consistently preserve mathematical equivalence relationships.

These geometric failures directly impede the formation of stable arithmetic circuits that could operate reliably across different input formats or numerical ranges. For LLMs to form stable, input-agnostic circuits for any algorithm, inputs must preserve the properties that algorithm expects, even when cast in high dimensions. Consider sorting—it fails if vector representations have random perturbations that violate ordering relationships. Without isometric properties in embedding space (preserving numerical order and relationships), models cannot form circuits that work consistently across all inputs. McLeish et al. (2024) demonstrate that altering input geometry alone can dramatically improve simple numerical reasoning, which might appear to contradict our contextual averaging argument. However, their approach involves training models exclusively on arithmetic problems, which inadvertently avoids the fundamental contextual averaging problem faced by

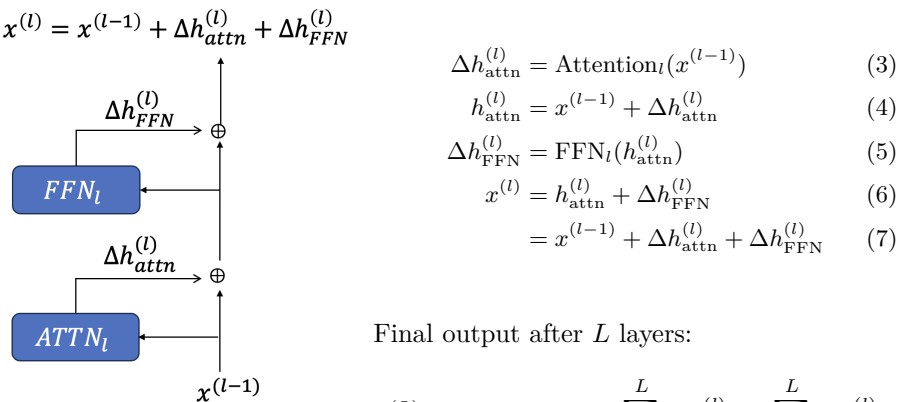

$$x^{(l)} = x^{(l-1)} + \Delta h_{attn}^{(l)} + \Delta h_{FFN}^{(l)}$$

$$\Delta h_{\text{attn}}^{(l)} = \text{Attention}_l(x^{(l-1)}) \tag{3}$$

$$h_{\text{attn}}^{(l)} = x^{(l-1)} + \Delta h_{\text{attn}}^{(l)} \tag{4}$$

$$\Delta h_{\text{FFN}}^{(l)} = \text{FFN}_l(h_{\text{attn}}^{(l)}) \tag{5}$$

$$x^{(l)} = h_{\text{attn}}^{(l)} + \Delta h_{\text{FFN}}^{(l)} \tag{6}$$

$$= x^{(l-1)} + \Delta h_{\text{attn}}^{(l)} + \Delta h_{\text{FFN}}^{(l)} \tag{7}$$

Final output after $L$ layers:

$$x^{(L)} = \text{embeddings} + \sum_{l=1}^{L} \Delta h_{\text{attn}}^{(l)} + \sum_{l=1}^{L} \Delta h_{\text{FFN}}^{(l)}$$

Figure 2: Computational flow through transformer layers. When processing arithmetic operations (e.g., "43 × 78 = ?"), attention aggregates the operation context (operator and operands) while FFNs must generate the result. Due to causal masking, attention at the "=" position can only compute weighted averages of the embeddings it sees, creating a representation that cannot contain the novel direction for the result token.

real-world LLMs. Once embeddings are contaminated through multi-domain training—where tokens like "9.11" accumulate conflicting associations across historical, versioning, and mathematical contexts—this geometric solution breaks down. The McLeish results thus support rather than refute our analysis: they show embedding geometry matters precisely because real-world training makes it unsolvable.

But suppose we could somehow solve the representation problem—imagine LLMs had perfect, mathematically-structured embeddings where "9.9" and "9.11" occupied precisely the right positions in embedding space. Would this enable reliable symbolic computation? Unfortunately, a deeper architectural constraint ensures the answer is no. Even with ideal symbolic representations, the computational mechanisms of transformer networks face fundamental limitations that force them toward pattern storage rather than algorithmic execution.

### 3.2 Claim 2: Feed-Forward Networks Resort to Pattern Storage

Even if we solved the representation problem—imagine LLMs had perfect, mathematically-structured embeddings—would transformers be able to execute symbolic computation? The answer reveals a profound gap between capacity and capability. Feed-forward networks possess, through the Universal Approximation Theorem, the capacity to approximate any continuous function. Yet in practice, they systematically fail at symbolic computation, resorting instead to pattern storage. This paradox lies at the heart of the computational split-brain syndrome.

**The Architectural Division of Labor.** To understand this paradox, we must first examine how computation flows through the transformer architecture and identify where symbolic computation must occur.

Figure 2 illustrates the computational pipeline of transformer layers.[2] Each transformer layer implements two successive residual connections: attention reads from the residual stream and adds its output back, creating an intermediate state that FFN then reads, transforms, and adds to produce the next layer's representation.

Consider a concrete example: '43 × 78 = ?'. Due to causal masking at the '=' position, attention only sees $\{43, \times, 78, =\}$ and computes weighted averages of these embeddings. This creates a rich representation encoding 'multiply 43 by 78'—identifying the operation and operands—but crucially, since attention can

---

[2]We omit layer normalization and dropout from our analysis as these operations—being element-wise rescaling and masking—do not alter the fundamental computational limitations we identify. LayerNorm normalizes representations but cannot enable symbolic computation.

only compute weighted combinations of its input embeddings, this representation cannot contain the novel embedding direction needed for '3354', which lies outside the space spanned by the input tokens.

Feed-forward networks then receive this attention-enriched representation and must somehow produce the correct result. Since attention cannot generate novel values, FFNs bear the entire computational burden: transforming the encoding of 'multiply 43 by 78' into the embedding direction for '3354'.

**Theoretical Capacity via Universal Approximation.** In principle, FFNs have sufficient capacity for this task. Each FFN implements an expand-compress architecture (Vaswani et al., 2017):

$$\text{FFN}(x) = W_2 \cdot \text{ReLU}(W_1 x + b_1) + b_2 \tag{8}$$

By the Universal Approximation Theorem (Hornik, 1991), such networks can approximate any continuous function $f : \mathbb{R}^d \to \mathbb{R}^d$ to arbitrary precision on compact sets. Theoretically, this includes multiplication: given sufficient width and appropriate weights, an FFN could approximate $x \times y$ within any bounded domain.

This raises a critical question: If FFNs have the capacity to approximate multiplication, why do they fail so catastrophically in practice?

**The Training Reality: Role Specialization.** The answer lies not in what FFNs can do, but in what gradient descent trains them to do. We now present our central theoretical result:

---

**Theorem (Role Specialization under Next-Token Prediction)**

Under gradient descent optimization for next-token prediction loss $\mathcal{L} = -\log p_\theta(w_t | w_{<t})$, the three parameter sets necessarily specialize into distinct computational roles:

- $\theta_E$ (embeddings) $\to$ Context-averaged representations

- $\theta_A$ (attention) $\to$ Operation context encoding

- $\theta_F$ (FFNs) $\to$ Memorized pattern-to-result mapping

*Proof Sketch:* The cross-entropy loss requires $x^{(L)} \cdot e_{\text{result}}$ to be maximal among all vocabulary tokens. Since attention is constrained to weighted averages of inputs, it cannot generate an embedding direction for the result token that lies outside the space spanned by the input embeddings. FFNs must therefore learn the discrete mapping from attention's operation encoding to result embeddings. While UAT guarantees this mapping can be approximated on the training set, gradient descent optimizes for memorizing specific pattern-to-token mappings rather than discovering algorithmic rules. See Appendix B for detailed analysis.

---

**Why Pattern Storage Wins.** To understand why memorization emerges inevitably, consider the optimization dynamics for arithmetic like '43 × 78 = 3354'. The gradient descent must minimize:

$$\mathcal{L} = -\log p(\text{'3354'} | \text{'43} \times \text{78} =\text{'}) \tag{9}$$

This requires the final representation $x^{(L)}$ to have maximum dot product with the embedding of token '3354' compared to all other vocabulary tokens. The gradients flow through three components with distinct constraints:

1. **Embeddings** lack the isometric properties required for arithmetic—numerical tokens cannot preserve ordering or distance relationships due to contextual averaging (Section 3.1).

2. **Attention** can only compute weighted averages of existing embeddings, unable to generate novel token directions outside their span.

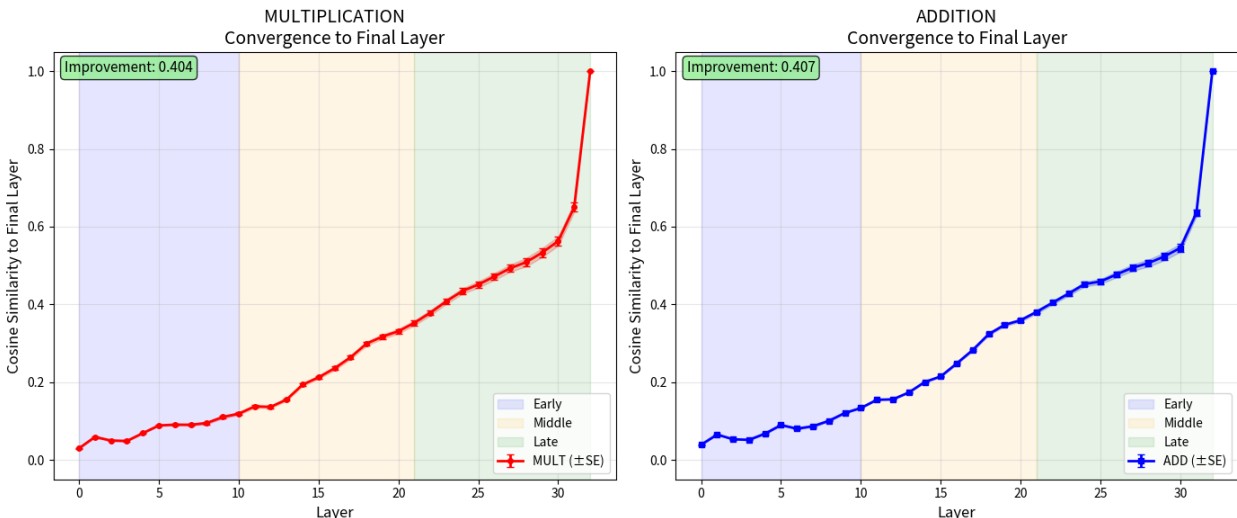

Figure 3: Experimental validation of hierarchical pattern assembly in arithmetic computation. Layer-by-layer convergence analysis for multiplication (left) and addition (right) shows progressive refinement from initial similarity of ∼0.07 to late-layer similarity of ∼0.48. Error bars represent standard error across 10 examples per operation. The consistent improvement patterns (0.404 for multiplication, 0.407 for addition) provide direct neural evidence for residual pattern fitting rather than algorithmic computation. Shaded regions indicate early (blue), middle (orange), and late (green) processing phases.

3. **FFNs** must bridge the gap: mapping attention's encoding to the specific embedding direction that maximizes probability for '3354'.

Since FFNs cannot implement true multiplication (piecewise linear functions cannot compute $x \times y$ exactly—see Appendix A), gradient descent converges to the only viable solution: memorizing a lookup table of pattern $\rightarrow$ result mappings. The UAT capacity that could theoretically approximate multiplication is instead allocated to storing these discrete mappings.

**Experimental Validation of Hierarchical Pattern Assembly.** To test our theoretical framework, we analyzed layer-by-layer hidden state trajectories in LLaMA-2-7B during arithmetic computationFor prompts like "$23 \times 96 =$" and "$63.7 + 3.5 =$", we measured how each layer's representation at the prediction position converges toward the final layer's output representation.

Figure 3 demonstrates strong empirical support for our theoretical framework. Both multiplication and addition exhibit nearly identical progressive refinement patterns: representations begin with low similarity to the final output (∼0.07) and gradually converge through three distinct phases. This layer-by-layer analysis confirms that since each individual FFN faces the same architectural impossibility of exact computation, the only viable learning strategy becomes hierarchical pattern decomposition, where multiple FFNs coordinate their residual contributions through learned pattern sequences rather than implementing systematic algorithms.

This pattern storage behavior aligns with recent findings by Dziri et al. (2023), who demonstrate that transformers memorize computation patterns from training rather than learning generalizable algorithms, with models succeeding primarily when test patterns appeared frequently in training data.

**The Approximation-Computation Gap.** The fundamental distinction between what FFNs can do in theory versus practice reveals why pattern completion differs from algorithmic approximation:

- **UAT Approximation**: FFNs can learn $f : K \to \mathbb{R}^d$ on compact training set $K$, successfully memorizing patterns like '23×17 → 391'.

- **Symbolic Computation**: True algorithms execute for any valid input, maintaining systematic behavior on novel combinations.

- **The Key Difference**: Robustness to distribution shift—algorithmic implementation (even imperfect) generalizes systematically, while pattern completion catastrophically fails outside memorized examples.

For novel inputs outside the training support $K$, FFNs have no algorithm to execute—only nearby memorized patterns to interpolate between. The discrete nature of token prediction exacerbates this: small errors in $x^{(L)}$ cause complete failure (wrong token), unlike continuous approximation where errors degrade gracefully. If students memorized multiplication tables then tried 3-digit problems, they would fail completely—only true algorithmic understanding enables generalization.

Many researchers, including ourselves initially, searched for generalizable arithmetic circuitry in LLMs. At inference time, such circuitry does not exist, as confirmed by studies showing LLMs rely on "bags of heuristics" rather than systematic algorithms (Nikankin et al., 2024; Tigges et al., 2024). Models perform pattern completion whether explaining algorithms or attempting execution—they just complete different memorized patterns.

**Complexity-Theoretic Confirmation.** This empirical limitation aligns with theoretical impossibility results. Li et al. (2024b) prove that constant-depth transformers can only compute functions in $\mathsf{TC}^0$, while multiplication requires serial carry propagation—fundamentally outside $\mathsf{TC}^0$. Even with universal approximation capacity, architectural constraints prevent true symbolic computation in a single forward pass. The $\mathsf{TC}^0$ result establishes that no clever weight configuration can enable exact multiplication, while our analysis shows that training dynamics allocate the available approximation capacity toward pattern memorization rather than algorithmic discovery.

This analysis of pattern storage versus algorithmic computation raises a puzzling question: if LLMs cannot execute symbolic operations reliably, why do they excel at explaining algorithmic procedures? When asked "How do you multiply two numbers?", models provide textbook-perfect explanations of the algorithm. This suggests a potential solution: perhaps these articulated procedures could implicitly guide execution, with the model's comprehensive knowledge of algorithms compensating for its execution limitations. Unfortunately, this intuitive hope fails due to a third architectural constraint that prevents instructional knowledge from rescuing symbolic computation.

### 3.3 Claim 3: Next-Token Prediction Decouples Instruction from Execution

The next-token objective treats algorithmic descriptions and execution examples as equivalent prediction tasks, preventing the binding of procedural knowledge to computational implementation. Even if architectural mechanisms existed to enable instructional guidance, the geometric separation of instruction and execution pathways ensures that procedural knowledge cannot rescue symbolic computation.

**The False Promise of Instructional Binding.** Training data contains examples that might appear to support instruction-execution binding. Consider instructional sequences like:

*"To multiply 23 × 17: First multiply 23 × 7 = 161, then multiply 23 × 10 = 230, finally add 161 + 230 = 391."*

Such examples seem to demonstrate algorithmic steps paired with execution traces, potentially enabling models to bind descriptions to implementations. However, next-token prediction processes these as sequential pattern completions without distinguishing instructional content from computational steps.

The expectation that exposure to such instruction-execution pairs would automatically enable binding represents a fundamental illusion about neural learning. Training models to recite algorithmic procedures does

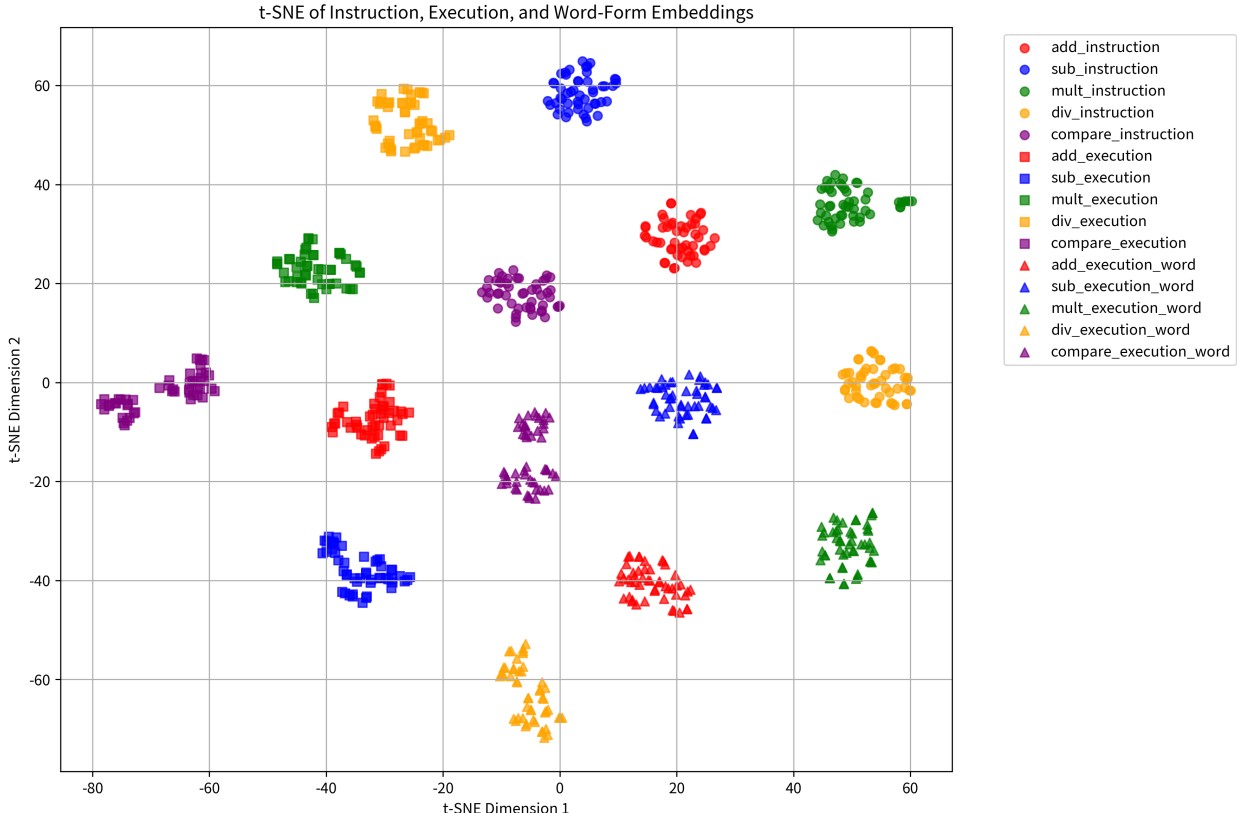

Figure 4: t-SNE projection of instruction, symbolic execution, and word-form execution embeddings from LLaMA2-7B-chat, across 50 problems each in five arithmetic operations. Instructions (circles), symbolic expressions (squares), and word-form results (triangles) occupy geometrically distinct regions, despite describing the same operation. This illustrates that LLaMA2-7B-chat stores instructional knowledge and executional know-how in separate regions of its latent space.

not automatically configure weights to perform those operations or enable inference-time binding between instruction and execution. The next-token objective does not provide a mechanism for such binding: it treats "how to multiply" and "what is 23×17" as independent text completion tasks.

**Indiscriminate Processing Across Context Types.** The training corpus contains arithmetic in diverse contexts: instructional examples with step-by-step explanations, homework solutions, calculations embedded within articles, and standalone problem-answer pairs. Next-token prediction treats all of these—instructional descriptions, execution traces, and standalone calculations—as equivalent token sequences to predict, without distinguishing their functional roles.

This creates two separate learning pathways with distinct representational signatures: models learn to recite algorithms by pattern-matching instructional texts, while learning to execute operations by retrieving stored pattern fragments from training examples. Critically, these competencies become geometrically separated in the model's latent space—instructional knowledge and execution capabilities occupy different representational regions.

**Experimental Validation of Geometric Separation.** If our hypothesis is correct, we should be able to directly observe this separation in the model's representational space. When a model encounters "To multiply 56 and 76, first break one number into parts..." versus "56 × 76 = 4256," are these processed by the same neural pathways or completely different ones?

To test this, we constructed a dataset of 250 arithmetic problems spanning five operations, each expressed in three distinct forms:

- an instructional sentence that explains the procedure (e.g., "To multiply 56 and 76, first break one number into parts..."),

- a symbolic execution form (e.g., "$56 \times 76 = 4256$"), and

- a word-form result (e.g., "A worker makes 56 parts per day for 76 days, totaling 4256 parts").

We embedded each sentence using LLaMA2-7B-chat and projected the resulting vectors with t-SNE. As shown in Figure 4, the model organizes these representations into clearly separated geometric clusters: instruction embeddings are well-separated from execution forms, and word-form results cluster more closely with symbolic executions than with instructions. This visualizes the structural dissociation between verbalized procedures and learned execution templates in the model's latent space.

We computed the mean embedding (centroid) for each (operation, role) cluster based on 50 examples per category. We then calculated pairwise cosine distances between these centroids and visualized the result as a heatmap (Figure 5). The cluster labels are grouped first by role (instruction, execution, worded execution) and then by operation. The plot reveals a clear block structure: instructional forms form one cluster with tight internal cohesion but are distant from both executional forms. In contrast, execution and worded execution are notably closer to one another, reflecting their shared emphasis on output form rather than procedural structure.

**Inference-Time Dissociation.** At inference time, different prompts steer the model toward geometrically distinct subspaces in its representational space. When prompted for algorithmic explanation (e.g., "How do you multiply two numbers?"), the model gravitates toward the instructional subspace, retrieving pedagogical patterns from its training distribution. When prompted for calculation (e.g., "What is $56 \times 76$?"), the model seeks out the execution subspace, accessing stored arithmetic patterns.

These operate as independent systems developed through separate pattern-matching processes during training, explaining why models can fluently articulate multiplication procedures while failing to execute them reliably. The geometric separation we observed in Figure 5 manifests functionally as this prompt-dependent subspace selection, where the model's response pathway is determined by which representational region the prompt activates rather than any principled binding between instruction and execution.

With all three architectural constraints now established—unstable symbolic representations, computational impossibility of direct symbolic operations, and geometric separation of instruction from execution—we can see how they interact to create the computational split-brain syndrome. No single constraint alone would be fatal, but together they systematically prevent LLMs from bridging the gap between comprehension and competence.

### 3.4 Real-World Manifestation: Structured Data Operations

The architectural constraints we identify manifest concretely in real-world applications where organizations attempt to use LLMs for data analysis tasks. Consider a seemingly simple query: finding the maximum sales value across branches from a CSV table embedded in a prompt. This operation, trivial for any database system, requires:

- Sequential iteration through records

- Maintaining comparison state across rows

- Tracking both the maximum value and its associated entity

- Consistent numerical comparison operations

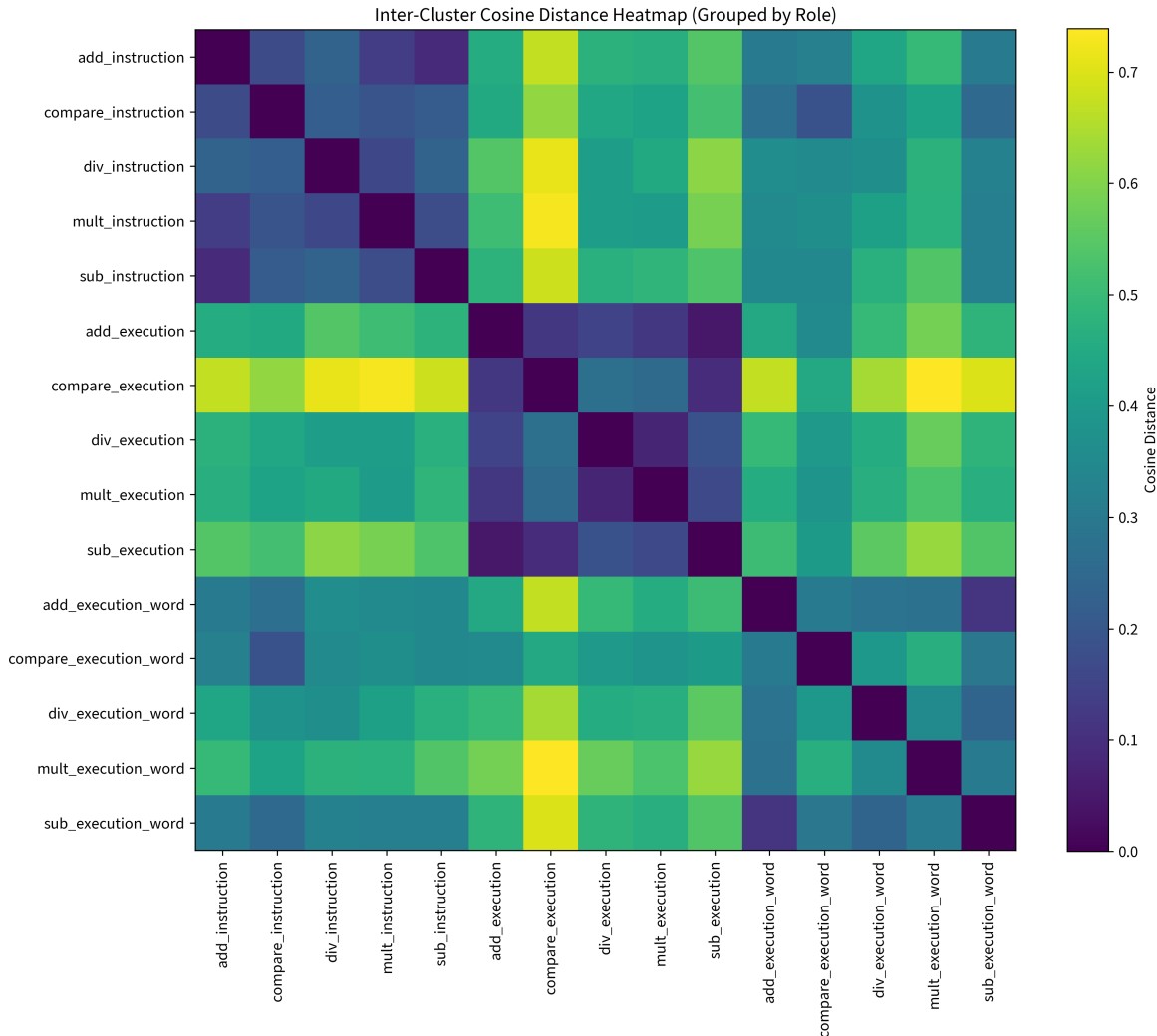

Figure 5: Cosine distances between cluster centroids for each operation-role pair, grouped by role: instructional texts (top block), execution strings (middle), and worded executions (bottom). Each centroid represents the average embedding of 50 samples per category, computed using LLaMA2-7B-chat. Clear geometric separation is observed between instructional and execution-related forms, with execution and worded execution clusters being closer to each other than either is to instruction. This provides quantitative support for the hypothesis that instruction-following and execution are learned as distinct representational pathways within the model's embedding space.

Recent empirical work by Wolff and Hulsebos (2025) provides direct evidence of these failures. Their evaluation of LLMs on analytical aggregations—including operations as fundamental as computing averages, sums, and maximum values—reveals "significant deficits in tabular reasoning performance." These multi-step algorithms exemplify precisely the symbolic operations our three architectural constraints predict to be impossible through pattern completion alone.

The failure is systematic: MAX requires maintaining comparison state across iterations, AVG demands accumulating sums while counting elements, and even COUNT necessitates consistent increment operations. Each violates our identified constraints: contextual averaging (Claim 1) prevents stable numerical comparison, FFNs cannot implement exact iterative operations (Claim 2), and the instruction-execution disconnect

(Claim 3) means that even when models can describe these algorithms perfectly, they cannot execute them reliably.

This has immediate practical implications. This has immediate practical implications. The inevitable failures are not due to inadequate training on tabular data or poor prompt engineering—they reflect fundamental architectural impossibilities that practitioners do not expect. As Wolff and Hulsebos (2025) demonstrate, performance degrades further with realistic data characteristics like missing values and duplicates, confirming these are architectural rather than training limitations.

### 3.5 The Broader Pattern: From Architecture to Cognition

With all three escape routes blocked—unstable symbolic representations, architectural impossibility of direct computation, and geometric separation of instructional knowledge from execution—LLMs inevitably resort to sophisticated pattern storage and retrieval systems. This explains the computational split-brain syndrome: models become excellent tutors who can articulate multiplication procedures (through instructional pattern-matching from Claim 3) while executing them through brittle hierarchical pattern retrieval rather than principled computation.

**Contrast with Symbolic Systems.** Traditional symbolic computation systems avoid these limitations through explicit design choices that transformer architectures lack:

- **Type Systems**: Variables are bound to specific domains (e.g., $a \in \mathbb{R}$) with operations defined over those types, preventing contextual contamination.

- **Algorithmic Implementation**: Operations like multiplication are implemented as explicit procedures rather than learned approximations.

- **Compositional Binding**: Complex expressions maintain consistent variable bindings through syntactic structure.

**Systematic vs. Pattern-Based Approximation.** Even when traditional systems use approximation, as in floating-point arithmetic, they maintain systematic behavior with guaranteed error bounds (IEEE 754). Each operation follows precise rules with predictable maximum errors. LLMs, by contrast, exhibit unsystematic pattern-based failures: correctly computing most arithmetic while failing unpredictably on specific cases, with no principled error bounds. For queries involving arithmetic operations, LLMs will therefore always encounter out-of-distribution problems if left to their own devices. This distinction highlights the fundamental difference between engineered approximation algorithms and emergent pattern retrieval.

**Path-Dependent Fragmentation and Interpretability Challenges.** The residual fitting process is highly sensitive to training dynamics across diverse domains, suggesting why mechanistic interpretability findings often lack generalizability. Since arithmetic represents just one of many competing objectives, the order and context of examples during training significantly impact which pattern clusters emerge and where they are encoded. Training data ordering, scheduling decisions, and model architecture jointly determine which statistical regularities crystallize into seemingly coherent "circuits."

Recent detailed mechanistic analysis confirms this training-dependency hypothesis. Nikankin et al. (2024) provide valuable evidence that arithmetic computation relies on distributed "bags of heuristics" spread across multiple layers, rather than implementing dedicated algorithmic circuits for basic operations ($+$, $-$, $\times$, $\div$). Their neuron-level analysis reveals that no coherent computational circuits exist for these fundamental operations. Importantly, they find that models within the same training lineage (Llama3-8B vs Llama3-70B) share more similar heuristic patterns than models from different training backgrounds (Pythia-6.9B, GPT-J), yet even within this shared lineage, the models differ in the degree and sophistication of these patterns. This training-dependency gradient—where shared training yields similar but not identical mechanisms—confirms that apparent "arithmetic mechanisms" reflect training-specific statistical regularities rather than universal computational principles.

**Reversing Dennett: Comprehension Without Competence.** This pattern inverts what philosopher Daniel Dennett observed in natural systems, where competence typically precedes comprehension (Dennett, 2017). Simple organisms demonstrate complex behaviors before developing explicit understanding. LLMs exhibit the reverse: sophisticated explanatory capabilities coupled with unreliable execution—comprehension without competence.

This reversal reveals LLMs as fundamentally different from both biological intelligence and traditional computational systems. They excel at linguistic pattern completion while generally lacking the grounded symbolic manipulation capabilities that such fluent descriptions would suggest.

These representational, computational, and learning constraints reveal why LLMs function as sophisticated pattern completion engines rather than symbolic reasoners. The computational split-brain syndrome emerges as a fundamental property of transformer architectures optimized for pattern completion rather than symbolic manipulation.

**Testable Predictions.** Our framework makes specific predictions that can be empirically validated:

1. Models trained exclusively on arithmetic should show different embedding geometry than general-purpose models, with more isometric numerical relationships

2. Pattern assembly signatures (gradual layer-by-layer convergence) should appear across all transformer model families

3. Instruction-execution separation should persist across architectures, including instruction-tuned and reasoning models

4. Novel format generalization should consistently fail while familiar format variations succeed (the key discriminator between pattern completion and algorithmic approximation)

These predictions provide a roadmap for future quantitative validation of our claims.

## 4 From Arithmetic to Relational Reasoning

If our architectural analysis is correct, the computational split-brain syndrome should not be limited to arithmetic. The same three constraints—contextual averaging, architectural impossibility of exact computation, and instruction-execution disconnect—should create identical failure patterns wherever systematic symbolic manipulation is required.

Relational reasoning provides the perfect test case. Like arithmetic, it demands automatic binding from natural language to symbolic structure, followed by systematic execution of transformations. Just as "9.11" must be bound as a numerical quantity rather than a historical date, "Alice" must be bound to appropriate relational roles before applying logical transformations like $\forall x, y \ [\text{Parent}(x, y) \rightarrow \text{Child}(y, x)]$.

Table 1: Two-Phase Computational Requirements. Both arithmetic and relational reasoning require automatic binding from natural language followed by systematic execution of transformations.

| Phase | Arithmetic Operations | Relational Reasoning |
|---|---|---|
| **Input Binding** | Automatically infer and bind: "9.11" $\rightarrow$ numerical value, "9.9" $\rightarrow$ numerical value | Automatically infer and bind: "Alice" $\rightarrow$ PERSON, family relationships from syntax |
| **Computational Execution** | Execute bound operations: $9.9 > 9.11$ | Execute inference rules: $\text{Parent}(X, Y) \rightarrow \text{Child}(Y, X)$ |
| **Consistency** | Maintain across multi-step procedures: $a > b \wedge b > c \rightarrow a > c$ | Maintain variable bindings across multi-step inference: $\text{Parent}(A, B) \wedge \text{Parent}(B, C) \rightarrow \text{Grandparent}(A, C)$ |

Table 1 illustrates these parallel computational demands across domains. Both require the same fundamental operations: automatic symbolic binding from natural language, reliable rule execution, and consistency

maintenance across multi-step reasoning chains. The same three architectural constraints predict systematic failures in relational reasoning: contextual averaging prevents clean symbolic binding, instruction-execution disconnect separates rule articulation from application, and pattern storage leads to memorized templates rather than systematic variable manipulation.

This theoretical extension predicts LLMs will exhibit identical split-brain syndrome across domains—fluent explanation of logical principles coupled with unreliable execution. The evidence strongly supports this prediction.

### 4.1 The Reversal Curse and Alice Problem

Consider two cases where LLMs fail at drawing basic conclusions that appear deceptively simple. The Reversal Curse (Berglund et al., 2023) tests models on their ability to perform bidirectional inference: models trained only on statements like "Tom Cruise's mother is Mary Lee Pfeiffer" are tested on the reverse direction "Who is Mary Lee Pfeiffer's son?"—a symmetric transformation humans perform unconsciously. The Alice problem (Nezhurina et al., 2024) presents a different challenge: models given "Alice has 4 sisters and 1 brother" must answer "How many sisters does Alice's brother have?" This requires rebinding Alice from her own perspective to her brother's perspective while maintaining compositional consistency across family relationships.

**Empirical Failures** The magnitude of these failures is striking. Even after fine-tuning on fictional statements like "Uriah Hawthorne is the composer of Abyssal Melodies," models cannot answer "Who composed Abyssal Melodies?" revealing the same automatic binding failures we identified in arithmetic domains. Experiments show that models fine-tuned on 1,000 fictional "A is B" statements achieved near-perfect accuracy on forward retrieval but only 7% accuracy on reverse queries (Berglund et al., 2023). This failure persists in real-world scenarios: when tested on celebrity facts, GPT-4 correctly answers forward questions like "Who is Tom Cruise's mother?" 79% of the time, compared to only 33% for reverse questions like "Who is Mary Lee Pfeiffer's son?" (Berglund et al., 2023). The model's log-probability for correct answers in the reverse direction shows no improvement over random baseline, indicating systematic rather than accidental failure.

The Alice problem (Nezhurina et al., 2024) presents equally dramatic breakdowns. Given the seemingly simple prompt "Alice has 4 sisters and 1 brother" and asked "How many sisters does Alice's brother have?", state-of-the-art models including GPT-4, GPT-4o, and Claude 3 Opus show "strong collapse of reasoning" across most tested models (Nezhurina et al., 2024). Despite the straightforward answer (5: Alice plus her 4 sisters), comprehensive testing across multiple prompt variations and at least 30 trials per model revealed frequent failures and extreme performance fluctuations on trivial problem variations. When confronted with more complex family structures (AIW+), performance collapsed to "close to 0" even for the most capable models (Nezhurina et al., 2024). Notably, models exhibit strong overconfidence in wrong solutions while providing "confabulation-like" explanations that sound plausible but demonstrate fundamental failures in compositional reasoning and variable binding.

**Computational Split-Brain in Relational Reasoning** These failures exemplify computational split-brain syndrome in relational reasoning, directly predicted by our architectural analysis in Section 3. Models demonstrably know both generic rules ("If X is Y's mother and Y is male, then Y is X's son") and specific facts ("Tom Cruise's mother is Mary Lee Pfeiffer"), yet systematically fail to execute the algorithmic steps required for inference. This reflects the same instruction-execution disconnect identified in Section 3.3, where procedural knowledge and computational implementation occupy geometrically separated representational spaces.

Consider the computational requirements for the reversal curse: given "Tom Cruise's mother is Mary Lee Pfeiffer" and asked "Who is Mary Lee Pfeiffer's son?", models must automatically infer that the tokens represent PERSON entities, bind the implicit MOTHER_OF relationship from surface syntax, recognize the query seeks the inverse relationship, apply the symmetric transformation $\text{Mother}(x, y) \rightarrow \text{Son}(y, x)$ in reverse, bind variables $x =$Mary Lee, $y =$Tom, and conclude Tom is Mary Lee's son. Similarly, the Alice problem requires binding family structure from "has 4 sisters and 1 brother," then rebinding Alice from SISTER role to SIBLING role to execute the perspective-shift computation.

Both tasks demand the symbolic manipulation capabilities that our architectural analysis shows current LLMs lack: contextual averaging prevents the clean symbolic binding required for role assignment, instruction-execution disconnect separates relational rule articulation from application, and pattern storage leads to memorized relationship templates rather than systematic variable manipulation.

Instead, models resort to directional pattern memorization. The reversal curse occurs because "A is B" patterns vastly outnumber "B is A" patterns in training corpora, causing asymmetric learning of fundamentally symmetric relationships. The same architectural constraints that prevent stable arithmetic circuits also undermine the variable binding and systematic transformation required for relational reasoning, creating the identical split-brain syndrome across domains—fluent explanation of logical principles coupled with unreliable execution.

We note that the reversal curse cannot be addressed through training data frequency balancing—simply ensuring equal exposure to "A is B" and "B is A" patterns. The issue is architectural: models learn separate representational pathways for each syntactic form rather than understanding the symmetric logical relationship. Even with perfect frequency balance, the system would still develop distinct pattern-matching rules instead of unified bidirectional binding. Solutions like LoCo-LM (Calanzone et al., 2025) succeed precisely because they bypass this limitation through explicit logical constraint enforcement in the training objective, essentially forcing systematic truth table coverage rather than relying on natural pattern completion tendencies.

### 4.2 Rule Applications, Logical Inconsistencies and Operator Failures

In contrast to the implicit complexity of the Reversal Curse and Alice problem, logical inconsistency tasks present LLMs with explicit formal structure that should simplify reasoning. These problems provide clear premises, stated rules, and well-defined logical relationships—yet LLMs continue to fail systematically. This failure despite explicit scaffolding reveals that the architectural constraints operate independently of how logical structure is presented.

Large language models exhibit a range of logical inconsistencies, as identified in LoCo-LM (Calanzone et al., 2025) and summarized by Cheng et al. (2025):

- **Negation Inconsistency:** The model affirms both "X is an organism" and "X is not an organism."

- **Implication Failure:** Given "All birds are animals" and "An albatross is a bird," the model fails to infer "An albatross is an animal."

- **Reverse Implication Failure:** From "If made of metal, then conducts electricity" and "X does not conduct electricity," the model fails to infer "X is not made of metal."

- **Deductive Chain Breakdown:** Given "Nails are made of iron," "Iron is a metal," and "Metals conduct electricity," the model fails to infer "Nails conduct electricity."

These failures persist even when facts and rules are fully spelled out (e.g., "All birds can fly" and "Tweety is a bird" → "Tweety can fly")—a setup that, unlike the Reversal Curse or Alice problem, explicitly states the binding relationships that should enable deduction if the architecture supported it.

Vashishtha et al. (2025) demonstrates success on causal reasoning tasks that test rule applications without variable binding. Pattern matching works when models are trained from scratch on clean synthetic data, with a 67M-parameter transformer outperforming billion-parameter LLMs on causal axioms. This confirms that direct rule application falls within transformers' capabilities when training conditions isolate logical operations from the contextual contamination of multi-domain pretraining.

**Systematic Evaluation of Relational Reasoning** The comprehensive study by Li et al. (2024a) provides decisive evidence of LLMs' architectural limitations in relational reasoning. They systematically evaluated state-of-the-art LLMs on inductive logic programming (ILP) tasks across both natural language and

truth-value matrix formats, comparing performance against Differentiable Logic Machines (DLM) (Zimmer et al., 2023)—specialized neural program induction models designed explicitly for logical reasoning.

The results are striking: despite being 100,000 times larger than DLM models, LLMs performed dramatically worse across all logical reasoning tasks. For family tree reasoning tasks requiring multi-step inference:

- **Simple relations** (HasFather): LLMs achieved 47-100% accuracy vs. DLM's perfect 100%

- **Complex relations** (IsUncle): LLMs dropped to 0-49% accuracy vs. DLM's 85%

- **Hierarchical relations** (IsMGUncle): LLMs achieved only 0-48% vs. DLM's 55%

Critically, these failures occurred across *both* natural language prompting and truth-value matrix representations. When given explicit logical structure through truth-value matrices—the same format used by specialized logical reasoning systems—LLMs still failed systematically. This definitively rules out training data scarcity or prompt engineering as explanations.

**Hallucination in Logical Reasoning** The study revealed systematic hallucination patterns that illuminate the underlying architectural problems. LLMs generated contradictory reasoning processes, such as claiming "P8 is the father of P3 and also the mother of P3" when given explicit family relationships. These aren't random errors but systematic pattern completion mistakes where models correctly identify surface facts but fail at the transformations required for compositional reasoning.

### 4.3 The Pattern Across Relational Reasoning

Our analysis reveals a convergent pattern across complexity levels: LLMs fail systematically regardless of how relational reasoning tasks are presented. Problems that appear simple (Reversal Curse, Alice problem) actually require sophisticated computational transformations that LLMs cannot execute. Problems that appear complex but provide explicit logical structure (truth-value matrices, clear logical relationships) should be easier to solve, yet LLMs still fail systematically. Table 2 summarizes the magnitude of these systematic failures (data from (Berglund et al., 2023; Nezhurina et al., 2024; Li et al., 2024a)), revealing consistently poor performance despite the apparent simplicity of many tasks.

Table 2: Systematic Failures in Relational Reasoning

| Task Domain | LLM Performance | Comparison |
|---|---|---|
| Reversal Curse (forward) | 79% | – |
| Reversal Curse (reverse) | 7% | Random baseline |
| Alice Problem (complex) | ∼0% | – |
| Family Relations (IsUncle) | 0-49% | DLM: 85% |
| Family Relations (IsMGUncle) | 0-48% | DLM: 55% |

This convergence is theoretically significant. If architectural limitations were merely about training data scarcity or prompt engineering, we would expect LLMs to succeed when given explicit logical structure. Instead, failure persists regardless of presentation format, confirming that the constraints operate at the architectural level—the same three limitations that prevent stable arithmetic circuits also undermine relational reasoning across all contexts.

**Lifted Reasoning as the Missing Capability** The consistent failure across both implicit and explicit reasoning tasks points to a fundamental missing capability: *lifted reasoning*. What these tasks demand—and what LLMs fundamentally lack—is the ability to apply general rules over arbitrary entities rather than memorizing instance-specific patterns. Lifted reasoning underlies human-like generalization in logic, mathematics, and programming. Unlike pattern completion, which operates over specific instances, lifted reasoning requires variable abstraction (binding entities to abstract roles that can be systematically manipulated), rule generalization (applying logical transformations that work across arbitrary entity instantiations), and compositional consistency (maintaining variable bindings across multi-step inference chains).

Current transformer architectures lack the representational and computational mechanisms necessary for these operations. Models specifically designed for these capabilities, like Neural Logic Machines (NLM) (Dong et al., 2019) and Differentiable Logic Machines (DLM) (Zimmer et al., 2023), achieve lifted reasoning through explicit tensor representations for predicates and specialized circuits for logical operations. These architectures explicitly address the binding problem by representing relationships as tensors and implementing differentiable operations that maintain variable consistency—architectural innovations absent from standard transformers.

The performance gap persists even when LLMs are given identical inputs to specialized models: despite being 100,000 times larger than DLM models, LLMs performed dramatically worse across all logical reasoning tasks (Li et al., 2024a), demonstrating that architectural constraints, not data availability, prevent reliable logical reasoning. This provides definitive evidence that the failures reflect fundamental limitations in how transformers process symbolic relationships, confirming our architectural analysis from Section 3.

This lifted reasoning gap explains why LLMs exhibit sophisticated pattern completion capabilities while systematically failing at the variable binding and rule application required for reliable relational reasoning. The computational split-brain syndrome manifests regardless of how logical structure is presented, confirming that the constraints operate at the architectural rather than training or prompting level. Understanding this fundamental limitation clarifies both what current LLMs can achieve as pattern completion engines and what alternative approaches might be needed to transcend these constraints.

## 5 Compensatory Strategies and Their Execution Gap

The computational split-brain syndrome reveals a fundamental paradox: LLMs can articulate principles they cannot reliably execute. However, this very comprehension capability suggests compensatory strategies. If models have memorized algorithmic procedures well, they can leverage these memorized patterns to unroll step-by-step solutions (self-scaffolding). If they can recognize problem types reliably, they can delegate to specialized external systems (tool delegation). Finally, rather than working around execution limitations, we can address them architecturally by integrating dedicated symbolic modules (hybrid architectures); essentially *an* internal tool-calling for critical operations.

Each approach leverages LLMs' strengths while addressing execution limitations differently. Yet as we demonstrate, all three strategies converge on the same fundamental requirement: coordinating these approaches demands reliable metacognitive capabilities that current architectures lack (see Section 5.4).

### 5.1 Self-Scaffolding: Leveraging Comprehension for Step-by-Step Execution

Self-scaffolding exploits LLMs' demonstrated ability to articulate memorized algorithmic procedures by having them generate explicit step-by-step decompositions, then attempt to execute each step themselves. This approach converts implicit reasoning into structured text patterns that leverage models' pattern completion strengths.

**Examples Across Domains.** A self-scaffolding approach to the problems of multi-step arithmetic, reversal curse, and Alice problem would generate:

> "To multiply $742 \times 89$: First, multiply $742 \times 9 = 6678$. Then multiply $742 \times 80 = 59360$. Finally, add $6678 + 59360 = 66038$."

> "Since Tom Cruise's mother is Mary Lee Pfeiffer, and the relationship 'mother of' is symmetric to 'son of', Mary Lee Pfeiffer's son must be Tom Cruise."

> "Alice has 4 sisters and 1 brother. From her brother's perspective, he has Alice plus her 4 sisters, making 5 sisters total."

The Illusion of Thinking study (Shojaee et al., 2025) is a recent example, demonstrating this pattern across planning puzzles, where Large Reasoning Models (LRMs) including Claude 3.7 Sonnet, DeepSeek-R1, and

OpenAI's o3-mini generate detailed algorithmic traces for problems like Tower of Hanoi while attempting execution.

## 5.2 Tool Delegation: Bypassing Model Execution Entirely

Tool delegation represents a fundamentally different strategy: rather than improving the model's execution capabilities, it bypasses them entirely by delegating computational tasks to specialized external systems. This approach has been systematized in frameworks like ReAct (Yao et al., 2022), Toolformer (Schick et al., 2023), and various agent architectures that enable LLMs to interact with external APIs and computational tools.

**Direct Delegation Examples.** For arithmetic, instead of attempting step-by-step execution, the model simply recognizes the problem type and delegates:

- "This is a multiplication problem" → `calculator.multiply(742, 89)`.

- For database queries: "Find maximum sales" → SQL: `SELECT MAX(sales) FROM data`.

- For logical reasoning: "Check satisfiability" → `SMT_solver.check(constraints)`.

The necessity of such delegation becomes clear when we consider the structured data operations analyzed in Section 3.4. Operations such as finding maximum values or computing averages require iterative state tracking that LLMs cannot perform reliably, yet these same models excel at recognizing when to invoke `SQL` or `pandas` functions that implement these algorithms correctly. This pattern extends beyond data operations to other domains requiring algorithmic execution.

The comment on the Illusion of Thinking paper (Opus and Lawsen, 2025) provides a particularly striking example of tool delegation's effectiveness. When Tower of Hanoi problems were reformulated as: "Solve Tower of Hanoi with 15 disks. Output a Lua function that prints the solution when called," models achieved high accuracy across tested systems (Claude-3.7-Sonnet, Claude Opus 4, OpenAI o3, Google Gemini 2.5), completing in under 5,000 tokens. This dramatic improvement occurred because models generated *algorithmic code* rather than attempting to enumerate moves themselves. Similarly, Cheng et al. (2024) demonstrate that separating symbolic code generation from symbolic code execution improves deductive reasoning.

## 5.3 Hybrid Architectures: Specialized Modules for Symbolic Operations

Hybrid architectures represent a more fundamental response to the computational split-brain syndrome: addressing the limitations architecturally rather than through workarounds. These approaches integrate specialized computational modules within LLM systems to handle symbolic operations that current transformer architectures cannot reliably implement. Essentially, the idea is to mend the comprehension-competence gap internal to the model.

**Domain-Specific Integration.** Several promising approaches have emerged across different symbolic domains. For arithmetic computation, OccamLLM (Dugan et al., 2024) and Internal General Computations (IGC) (Dietz and Klakow, 2024) incorporate specialized circuits that directly implement arithmetic operations, achieving perfect accuracy while avoiding pattern-based approximations. For logical reasoning, Logic-LM (Pan et al., 2023) integrates LLMs with symbolic solvers through translation-execution pipelines that maintain logical consistency.

Particularly encouraging are approaches like Differentiable Logic Machines (DLM) (Zimmer et al., 2023), which represent predicates as tensors and implement logic operations as differentiable functions. Unlike approaches that use neuro-symbolic losses to improve consistency in existing architectures, DLMs provide fundamental architectural support for rule learning and symbolic generalization, directly addressing the induction barriers we identified.

**Functional Specialization Architectures.** Such integration, if done well, would combine pattern-matching flexibility with principled symbolic operations. Rather than emulating symbolic computation through pattern completion, such systems would implement dedicated mechanisms for variable binding, rule application, and systematic inference.

Current Mixture-of-Experts (MoE) architectures (Shazeer et al., 2017) optimize primarily for computational efficiency—routing tokens to different parameter sets to increase model capacity—rather than functional specialization based on cognitive capabilities. A neuroscience-inspired approach could allocate specific experts to handle symbolic operations, relational reasoning, pattern completion, and other specialized functions. Just as the human brain has dedicated regions for executive reasoning (prefrontal cortex) and memory (temporal lobe) (Kandel et al., 2021), functionally partitioned MoE could develop experts that excel at exactly the symbolic binding and manipulation operations that current general-purpose feed-forward networks struggle to implement.

## 5.4 Coordination and The Metacognitive Bottleneck

**Complementary Strengths and Shared Limitations.** The three compensatory strategies form a complementary toolkit for addressing the computational split-brain syndrome, each leveraging LLMs' pattern completion strengths while circumventing execution limitations in different ways. Self-scaffolding requires no external components or architectural modifications, working entirely within existing capabilities by converting implicit reasoning into explicit textual decompositions. It excels in medium-complexity scenarios where individual steps fall within reliable pattern storage, as demonstrated by Large Reasoning Models successfully solving problems like the Alice puzzle where base LLMs fail (Mitchell, 2025). Tool delegation converts architectural impossibilities into architectural strengths, achieving perfect reliability by offloading computation to dedicated symbolic processors. When appropriate tools exist, this approach works reliably across complexity levels for well-defined domains while minimizing computational overhead.

However, each strategy faces critical limitations that reveal a deeper architectural constraint. Self-scaffolding incurs massive computational overhead (multiplying two 10-digit numbers requires approximately $10^2$ FLOPs on a CPU but $10^{12}$ FLOPs in transformer inference), suffers from planning failures when decomposition relies on unreliable pattern completion, and encounters execution unreliability when individual steps require symbolic manipulation beyond pattern storage capabilities (Shojaee et al., 2025). Tool delegation faces problem recognition requirements (identifying when and which tools to use), tool availability constraints (limited to well-defined domains with existing specialized systems), and composition challenges when complex problems require coordination between multiple tools or combinations of delegation and direct reasoning.

While hybrid architectures offer the most principled solution by mending the comprehension-competence gap internally through dedicated symbolic processing capabilities, they face fundamentally similar limitations to external tool delegation. Just as external tool calling requires recognizing when and which tools to use, hybrid architectures must coordinate between symbolic and pattern completion modules—essentially internal tool routing decisions. This introduces integration complexity, neural adaptation requirements that may constrain symbolic expressiveness, and the same metacognitive challenge of determining when symbolic processing versus pattern completion is appropriate. Moreover, the internal nature of this coordination makes it potentially harder to monitor, debug, or override compared to external tool delegation.

**The Metacognitive Dependency.** Most critically, all three approaches converge on the same fundamental requirement: they demand reliable metacognitive assessment to coordinate when and how to apply each strategy. Self-scaffolding requires knowing when decomposition will help versus hurt, tool delegation requires recognizing appropriate problem types and available tools, and hybrid architectures must coordinate between symbolic and pattern completion modules. This shared dependency exposes a recursive problem: the compensatory strategies themselves require the very introspective abilities that the computational split-brain syndrome prevents.

However, compensatory strategies require careful coordination and expose deeper introspection requirements. Consider an idealized metacognitive control loop (Algorithm 1) as an illustrative framework: while this process could theoretically be executed through pattern completion—LLMs can perform individual steps

---

**Algorithm 1** Idealized Metacognitive Control Loop

---
1: $P \leftarrow$ input problem
2: $d \leftarrow$ ASSESS_DIFFICULTY$(P)$ {requires introspection}
3: **if** $d \leq \tau$ **then**
4:    **return** PATTERN_COMPLETE$(P)$
5: **else**
6:    $P' \leftarrow$ DECOMPOSE_OR_DELEGATE$(P)$ {tool call / planner}
7:    **return** SOLVE$(P')$
8: **end if**

---

like tool calling and decomposition using their existing capabilities (e.g., GPT-4's tool-calling demonstrations)—it reveals fundamental challenges in metacognitive assessment. The difficulty assessment in line 2 and decomposition choice in line 6 would require metacognitive capabilities that face inherent limitations:

- Complexity assessment (line 2): "Is this problem computationally hard for my specific architecture?" (But how can the model assess difficulty without attempting the task?)

- Capability introspection (line 2): "Can I execute this operation reliably through my pattern storage?"

- Tool appropriateness (line 6): "Do I have the right external system for this problem type?"

- Decomposition quality (line 6): "Will breaking this into sub-steps actually help?" (The decomposition process itself can be brittle, as suggested by reasoning model studies in (Shojaee et al., 2025))

- Step reliability (line 6): "Can I trust each sub-computation in my scaffolding?"

The deeper issue is metacognitive: while commercial LLMs have developed sophisticated capabilities for recognizing when to delegate computational tasks to external tools, they lack reliable self-awareness of their internal computational limitations when such delegation is unavailable. The same instruction-execution disconnect that prevents reliable symbolic computation also creates fundamental challenges for model self-assessment of their internal pattern completion capabilities. Recent work has identified these metacognitive limitations across LLMs, with systematic overconfidence and poor calibration, particularly on out-of-distribution tasks (Geng et al., 2024; Kadavath et al., 2022). This metacognitive deficit manifests critically in high-stakes applications: studies of medical reasoning show that LLMs "lack essential metacognition for reliable medical reasoning," consistently failing to recognize knowledge limitations (Griot et al., 2025).

To test the importance of metacognition, we designed a tightly controlled experiment using $n$-digit $\times$ $n$-digit multiplication tasks with place-value decomposition and simulated column-wise addition. We tested Claude Sonnet 4, GPT-4o, and Gemini 2.5 Flash across three conditions, with results summarized in Table 3:

- **Zero-shot direct calculation:** Models were asked to calculate multiplication problems directly with explicit constraints against scaffolding, decomposition, or external tools ("Calculate this multiplication using only your internal reasoning, without using any tools or code"). All models achieved 0% accuracy on 10-digit problems, demonstrating universal computational split-brain syndrome when constrained to pure transformer computation. At 5-digit complexity, Claude achieved 10.5% accuracy while GPT-4o and Gemini remained at 0%.

- **Self-generated decomposition:** We tested whether models could independently generate appropriate decomposition prompts for multiplication problems. When asked to create step-by-step decomposition instructions, all models succeeded on 5-digit tasks. For 10-digit problems, GPT-4o produced prompts with 65% accuracy in decomposition steps and 94.2% overall quality, while Claude Sonnet 4 and Gemini 2.5 Flash maintained perfect decomposition accuracy.

- **Golden decomposition execution:** We tested execution using *golden* decomposition prompts that explicitly guide step-by-step computation using place-value breakdown. Models were constrained to

use only internal capabilities without external tools or self-correction. Table 3 summarizes performance across two difficulty levels.

Table 3: Performance on multiplication problems with golden decomposition prompts. Sum errors refers to failures in final addition despite correct intermediate steps.

| Complexity | Metric | GPT-4o | Gemini 2.5 | Claude Sonnet 4 |
|---|---|---|---|---|
| 5-digit | Overall Accuracy | 5% (1/20) | 95% (19/20) | 100% (20/20) |
| | Step-wise | 95–100% | 95–100% | 100% |
| | Sum Errors | 19/20 | 1/20 | 0/20 |
| 10-digit | Overall Accuracy | 0% (0/21) | 0% (0/20) | 0% (0/20) |
| | Step-wise | 76–100% | 95–100% | 95–100% |
| | Sum Errors | 18/21 | 20/20 | 20/20 |

With 5-digit numbers, GPT-4o achieved 95–100% accuracy on individual multiplication steps yet failed all but one problem due to arithmetic errors in the final summation. Gemini 2.5 Flash performed near-perfectly (95% overall accuracy), while Claude Sonnet 4 achieved perfect performance. As complexity increased to 10-digit numbers, all models degraded: GPT-4o's step-wise accuracy collapsed to 76–100% with systematic errors, while Claude and Gemini maintained high step accuracy (95–100%) but failed all final answers due to summation errors.

Critically, when given natural prompts without explicit constraints, these models automatically delegate multiplication problems to external computational tools, achieving perfect accuracy through sophisticated tool-routing capabilities. Our experiments reveal what happens when this learned compensatory strategy is disabled, exposing the underlying computational split-brain syndrome that automatic tool delegation normally masks.

These results reveal a fundamental limitation: even with *perfect* algorithmic decomposition, neither individual computational steps nor aggregation operations remain reliable as per-step complexity increases. Models face a computational trilemma: either (1) execute multi-step procedures internally and risk systemic breakdowns, (2) rely entirely on external tools, or (3) decompose steps further—which introduces new failure points even under perfect prompting.

Real-world applications involve countless multi-step algorithms of unknown complexity where external computational verification is unavailable. This necessitates reliable metacognitive assessment of internal capabilities: models must recognize their computational limitations and coordinate appropriate compensatory strategies without external guidance. While commercial models have learned sophisticated tool delegation for well-defined computational tasks, they lack the self-awareness needed to assess the reliability of their internal pattern completion processes.

This raises a fundamental question about the nature of metacognition itself. Large Reasoning Models and approaches like Reflexion (Shinn et al., 2023) appear to learn metacognitive skills: knowing when to reflect, when to distrust their initial judgment, when to backtrack. During training, external verifiers provide ground truth signals about when reflection is needed, allowing models to learn patterns about failure recognition. At inference time, this "metacognition" manifests as pattern association: recognizing cues that previously correlated with the need for reflection. But is this genuine self-awareness or sophisticated pattern matching? The fundamental challenge remains that any metacognitive assessment—"I should doubt this answer," "This reasoning seems flaky"—must itself emerge from pattern recognition over the same representational pathways we have shown to be unreliable for systematic reasoning.

Perhaps this mirrors human cognition itself—our sense of self-awareness may similarly emerge from pattern recognition over internal states rather than direct introspective access to our computational processes. This suggests that the distinction between "genuine" and "pattern-matched" metacognition may be less meaningful than initially apparent: if both humans and LLMs rely on learned associations to assess their own reasoning, the key question becomes not the authenticity of self-awareness, but its reliability and scope.

**Summary** Understanding these metacognitive requirements clarifies a fundamental limitation of current approaches to the computational split-brain syndrome. While the three compensatory strategies form a

powerful toolkit—each leveraging LLMs' pattern completion strengths in different ways—they all converge on the same recursive problem: coordinating these sophisticated workarounds requires the very introspective abilities that the split-brain syndrome prevents.

Even with perfect algorithmic decomposition, as our experiments demonstrate, execution gaps persist at the individual step level. This suggests that the instruction-execution disconnect operates at a more fundamental architectural level than compensatory strategies can address. The metacognitive bottleneck reveals why current LLM capabilities, however sophisticated, remain fundamentally constrained: they excel as pattern completion engines but lack the self-awareness needed to reliably coordinate between comprehension and competence. This clarifies both what current architectures can achieve and what limitations persist even with the most sophisticated compensatory mechanisms.

## 6 Pattern Completion: Strengths and Limitations

Having examined LLMs' systematic limitations in symbolic computation (Section 3), relational reasoning (Section 4), and compensatory strategies (Section 5), two critical questions emerge: How do we explain the remarkable power of LLMs as they are today? And what are the consequences of their architectural limitations? Our analysis points to pattern completion as both the source of their strengths and the root of their symbolic weaknesses.

**Definition 1.** *General intelligence = sophisticated pattern completion across diverse domains, with robust knowledge retrieval and flexible task adaptation.* **Generalizable intelligence** *= systematic rule discovery and principled reasoning that can be deployed to novel tasks and drives scientific progress.*

We demonstrate that LLMs excel at the former but encounter fundamental barriers at the latter. The following analysis traces where today's LLMs succeed on **general** intelligence but fail on **generalizable** intelligence.

**Multi-level Pattern Completion.** LLMs achieve impressive capabilities through a hierarchy of pattern completion mechanisms. At the most basic level, next-token prediction enables fluent text generation, while higher up, the same mechanism supports chain-of-thought reasoning as pattern-matching operations over common reasoning structures.

Most recently, test-time computation allows these models to upgrade from LLMs to LRMs, functioning as self-programming metacomputers that dynamically adjust reasoning strategies based on task complexity. Problem decomposition, exploration, trial-and-verification, and backtracking can all be viewed as higher-level patterns for task execution.

This pattern-matching approach helps to explain why LLMs excel at tasks that can be solved through pure memorization (like retrieving facts) or tasks that appear to require symbolic manipulation but can actually be solved through pattern matching (like solving standardized math problems in familiar formats).

**The Performance Cliff: From Pattern Recognition to Rule Discovery.** Because LLMs conflate similarity search with logical inference, recent empirical evidence from the ARC-AGI benchmark progression crystallizes the distinction between general and generalizable intelligence. ARC-AGI-1 evaluates pattern recognition and single-rule application in visual reasoning tasks, requiring models to identify and apply transformations from minimal examples (Chollet, 2019). A forthcoming benchmark extension (ARC-AGI-2) advances this evaluation by specifically targeting rule discovery and systematic application: tasks require inferring multiple interacting rules from limited observations, applying them compositionally across sequential steps, and adapting rule application based on contextual cues[3].

The results reveal a fundamental performance cliff: while OpenAI's o3 achieved 87.5% on ARC-AGI-1 through sophisticated pattern completion, the same architecture scored < 3% on ARC-AGI-2 (Chollet et al., 2025; ARC Prize Foundation, 2025). Performance plummets once pattern completion loses its near-neighbor anchors from training, despite identical instruction formats. This cliff occurs precisely where tasks transition

---

[3]ARC-AGI-2 details from pre-release technical reports available at `https://github.com/fchollet/ARC-AGI`

from pattern recognition to genuine rule discovery and systematic reasoning—exposing the architectural limitations we have identified throughout this paper.

This performance cliff exemplifies our architectural analysis: LRMs achieve improvements through sophisticated self-scaffolding and test-time computation (compensatory strategies), but the underlying computational split-brain syndrome persists—they remain pattern completion engines that fail when tasks require genuine rule discovery and systematic reasoning beyond their memorized patterns.

**Inductive Reasoning and the Limits of Generalizable Intelligence.** Inductive reasoning spans a spectrum of complexity, from simple function learning to discovering fundamental principles that govern entire domains. Consider the challenge of deriving an algorithmic solution from scratch. Recent work by Cheng et al. (2024) demonstrates that LLMs can excel at the simplest level—learning mathematical functions like base conversion from input-output examples—achieving near-perfect accuracy. At a higher level, FunSearch (Romera-Paredes et al., 2023) shows that LLMs can discover new algorithmic constructions for combinatorial problems through evolutionary code search, finding novel solutions to established mathematical challenges like the cap set problem.

However, discovery of novel algorithms such as solving the Tower of Hanoi problem *from scratch* requires a much stronger set of capabilities. The optimal algorithm that works for any number of disks $n$ requires recognizing that moving $n$ disks involves first moving $n-1$ disks, then the largest disk, then the $n-1$ disks again. This recursive insight, itself a meta-pattern, is even more challenging than what ARC-AGI-2 tests. Recent evaluation shows that state-of-the-art reasoning models like Claude 3.7 Sonnet and OpenAI's o3-mini exhibit complete performance collapse beyond 8-9 disks, despite having sufficient token budget to continue (Shojaee et al., 2025).

Scientific discovery derives compact governing principles from noisy, real-world observations. Recent breakthroughs demonstrate that AI can indeed discover such principles, but through architecturally specialized approaches. AI Feynman succeeds through a physics-constrained symbolic search, whereas the LLM symbolic regression relies on pattern-based guessing (Udrescu and Tegmark, 2020; Makke and Chawla, 2022). These successes require domain-specialized architectures with strong inductive biases, such as geometric priors, physics constraints, and symbolic manipulation engines. It remains unclear whether pattern completion alone, however powerful, can accomplish genuine scientific discovery.

# 7 Implications for Mechanistic Interpretability

Having established three architectural constraints that create the computational split-brain syndrome, we now examine how our framework relates to findings from mechanistic interpretability research. While these sophisticated tools have revealed important insights about model internals, we argue that the architectural constraints we identify suggest an alternative interpretation of what these tools discover: sophisticated pattern coordination rather than algorithmic implementation. This perspective does not diminish the value of interpretability research but rather reframes what we might be learning from it.

## 7.1 From Activation Patching to Sparse Autoencoders

Early mechanistic interpretability relied heavily on activation patching, intervening on specific neurons or attention heads to understand their causal role in model behavior. This approach revealed that arithmetic in LLMs rely not on clean algorithmic circuits but on what Nikankin et al. (2024) characterize as a "bag of heuristics"—distributed, overlapping pattern detectors with no coherent computational structure.

However, activation patching faces several limitations that motivated the development of more sophisticated methods. The polysemantic nature of individual neurons makes it difficult to isolate specific computational functions through patching alone—single neurons often represent multiple concepts simultaneously. Interventions that work on training-like inputs often fail on novel examples, revealing poor out-of-distribution generalization. Furthermore, our analysis suggests that the specific neurons and attention heads involved in any computation are highly path-dependent, varying significantly across training runs even when final

behavior converges, making findings from activation patching potentially idiosyncratic to specific model instances.

Sparse Autoencoders (SAEs) emerged as a proposed solution to the polysemanticity problem (Bricken et al., 2023; Templeton et al., 2024). SAEs are unsupervised learning models that decompose neural network activations into sparse, interpretable features by learning an overcomplete dictionary of feature directions. By expanding activations into a higher-dimensional space with sparsity constraints, SAEs aim to identify monosemantic units—features that capture single, interpretable concepts rather than the mixed representations found in individual neurons. These features appear more stable across models, with Wang et al. (2025) demonstrating that similar features emerge across different open-source architectures trained on comparable data.

Recent methods extend interpretability analysis from local interventions to whole-network computation traces. Linear Computation Graphs (He et al., 2024) build on SAE features to trace complete computational pathways across layers, while Attribution Graphs (Lindsey et al., 2025a) trace information flow through raw model activations without requiring feature decomposition. Despite their different approaches—one working through SAE features, the other through raw activations—both methods reveal similar patterns of distributed, redundant computation rather than clean algorithmic circuits.

## 7.2 Alternative Interpretations of Interpretability Findings

The interpretability findings discussed above are typically interpreted as evidence of computational mechanisms within LLMs. Our architectural framework, however, suggests an alternative lens through which to view these same empirical results—one that emphasizes pattern coordination over algorithmic implementation. This interpretation, while consistent with the data, differs from how the original authors might characterize their findings.

Consider the Indirect Object Identification (IOI) circuit identified through Linear Computation Graphs (He et al., 2024). The circuit traces how models process sentences like "When Mary and John went to the store, John gave Mary a bottle," identifying features that track entities and their relationships. Yet, our framework suggests these features recognize statistical patterns—names appearing early in sentences often receive objects mentioned later—rather than implementing genuine reference resolution through variable binding. The original authors might interpret these circuits as genuine computational mechanisms; our framework offers an alternative interpretation where these represent sophisticated statistical pattern recognition.

The cross-architecture universality of features (Wang et al., 2025) initially seems to suggest discovery of fundamental computational units. However, our framework suggests this universality could reflect convergent solutions to pattern matching. Whether these represent algorithmic primitives or statistical patterns remains an open empirical question. Models trained on similar corpora develop similar pattern-recognition strategies because they encounter similar statistical regularities.

This interpretation aligns with the "bag of heuristics" finding (Nikankin et al., 2024). While their analysis examined neuron-level patterns rather than SAE features, our framework predicts—though this remains to be empirically tested—that similar limitations would apply to SAE features. The heuristics are not implementation accidents that better interpretability tools could resolve into clean algorithms. They reflect the fundamental strategy available to pattern-completion architectures: coordinate multiple pattern detectors to approximate computational outcomes without implementing computational procedures. SAE features may be more stable across training runs than individual neurons—they capture statistical regularities that any model trained on similar data must learn. But these stable regularities remain patterns rather than algorithms.

The apparent success of mechanistic interpretability may itself exemplify the computational split-brain syndrome. We excel at comprehending what features correlate with—recognizing that certain features activate on arithmetic, logical relations, or syntactic structures. Yet, we may be mistaking this comprehension for understanding of computational competence. A feature that activates on multiplication and whose ablation disrupts multiplication could be recognizing multiplication patterns rather than implementing multiplication.

The very interpretability of these features—that they correspond to human-recognizable concepts—might bias us toward assuming they implement rather than recognize these concepts.

### 7.3 Critical Questions for Interpretability Research

Our framework suggests two methodological directions that could help distinguish between pattern-matching and algorithmic hypotheses in interpretability research:

**Systematic Testing at Scale.** Current interpretability work often demonstrates features on expected patterns without systematic out-of-distribution testing. Following the methodology of Nikankin et al. (2024) and GSM-Symbolic, interpretability claims should be validated across:

- Surface form variations ("3 + 5" vs. "three plus five" vs. "add 3 and 5")

- Systematic perturbations (changing numbers, adding irrelevant clauses, reordering)

- Distribution shifts that preserve computational requirements

If SAE features truly implement computational operations rather than pattern matching, they should maintain their function across these variations. Current evidence suggests they do not—features that appear to track arithmetic operations fail when numbers appear in unexpected formats, revealing pattern recognition rather than algorithmic implementation.

**Addressing Confirmation Bias.** Interpretability research currently suffers from a methodological limitation: features are typically interpreted after observing their activation patterns on known examples. This post-hoc analysis invites confirmation bias—we see what we expect to see. A more rigorous approach would:

- Interpret features based solely on their structure before examining behavior

- Make explicit predictions about feature responses to novel inputs

- Report both successful and failed predictions transparently

- Test features on adversarial examples designed to separate correlation from causation

These methodological improvements would help distinguish between two fundamentally different types of findings:

1. **Statistical Mechanism Discovery**: Understanding how models coordinate pattern matching to produce outputs—what current methods achieve effectively

2. **Algorithmic Mechanism Discovery**: Identifying computational procedures that generalize beyond training distributions—what may be architecturally impossible in current transformers

We emphasize that current interpretability research has made substantial contributions to understanding how LLMs process information. Our critique is not of the methods or findings themselves, but rather suggests a reframing of what these findings might represent given the architectural constraints we identify. By acknowledging that we study sophisticated pattern coordination rather than algorithmic implementation, interpretability research can provide more accurate insights into both the capabilities and fundamental limitations of current language models. This reframing may ultimately prove more valuable than maintaining the illusion that we are reverse-engineering computational mechanisms that may not exist.

## 8 Reflections on the Research Journey

This work emerged from a multi-year investigation that began with a seemingly simple question: how do LLMs perform grade-level math problems that, while conceptually simple, require multi-step algorithmic execution? The search for generalizable arithmetic circuitry led to a deeper puzzle—why should grade-school mathematics ever present an "out-of-distribution" problem for models trained on vast corpora? In parallel, the Reversal Curse and Alice-in-Wonderland problems revealed that automatic binding to concepts and relations—something humans take for granted—simply does not occur in LLMs. The viral "9.11 vs 9.9" comparison, initially dismissed as an amusing anecdote, crystallized as a fundamental input representation problem: tokens carry contextual contamination that prevents clean symbolic binding. Gradually, we recognized these weren't isolated curiosities but systematic failures spanning arithmetic, logic, and relational reasoning—all stemming from the same architectural constraints. This journey from searching for arithmetic circuits to discovering their impossibility ultimately revealed the computational split-brain syndrome as a unifying explanation for why LLMs can explain what they cannot execute.

## 9 Conclusion

The apparent intelligence of LLMs belies a deeper fragility. Despite their success in pattern-rich tasks, these models consistently fail to generalize principles, execute symbolic computations, or reason reliably—even under idealized conditions. Our analysis reveals that these failures are not incidental but structural: LLMs dissociate instruction interpretation from execution, leading to a computational split-brain syndrome that prevents the formation of robust, composable operations.

This diagnosis clarifies why interpretability tools often uncover expressive artifacts rather than reliable algorithms, and why model self-explanations may diverge from the actual pathways used in computation. Our finding that instruction and execution pathways are geometrically separated raises fundamental concerns: models' explanations of their reasoning may not reflect their actual computational processes, challenging current approaches to explainability. Training order influences which internal patterns are embedded and where—even when outward behavior remains stable—causing interpretability efforts to surface path-dependent artifacts rather than consistent mechanisms.

These insights reframe current conversations about emergence and scaling. Our findings align with critiques by Bender et al. (2021) and others who argue that the "bigger-is-better" paradigm has fundamental flaws. By demonstrating that LLM limitations are architectural rather than scale-dependent, we provide technical evidence that more data or parameters cannot resolve bottlenecks rooted in architectural design. This suggests redirecting resources from pure scaling toward architectural innovations could reduce computational and environmental costs while improving reliability.

These findings have immediate implications for high-stakes applications where reliable reasoning is critical. Current LLMs require careful scaffolding, external verification, or hybrid architectures rather than deployment as standalone reasoning systems in domains like medical diagnosis, legal analysis, or safety-critical decision making. The computational split-brain syndrome we identify suggests that apparent fluency in explaining procedures should not be mistaken for reliable execution capability.

If we seek genuinely general and generalizable intelligence, we must look beyond pattern completion engines. Future systems will need metacognitive scaffolding, lifted representations, and architectural support for principled execution. Our findings offer not only a critique of current LLMs, but a foundation for building the next generation of intelligent systems: one that can reason, not just react.

## Acknowledgments

This work benefited greatly from insightful discussions and valuable feedback from colleagues. Special thanks to Yasser Shaaban, David Paul Wipf, Tong He, and many students for reviewing early drafts and offering constructive comments. Tong He is also gratefully acknowledged for assistance in refining the experiments.

We thank the TMLR reviewers for their thoughtful feedback, which substantially improved the clarity and rigor of this work.

The author acknowledges the use of AI assistants (Claude, Anthropic; ChatGPT, OpenAI) for brainstorming discussions, related work synthesis, code development and debugging, and manuscript editing. All core theoretical contributions, experimental design, and analytical insights remain the author's own. Finally, the author thanks Kevin Zhang for unknowingly but instrumentally sponsoring his dad's research.

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

# A    Theoretical Proofs of FFN Limitations

## A.1    Piecewise Linearity of Feed-Forward Networks

**Theorem 1** (FFN Piecewise Linearity). *For any choice of weights and biases, the function implemented by a feed-forward network with ReLU activation is piecewise linear.*

*Proof.* Consider the FFN computation: $\text{FFN}(x) = \max(0, xW_1 + b_1)W_2 + b_2$

1. Each neuron in the hidden layer computes $h_i = \max(0, x \cdot w_i + b_i)$ where $w_i$ is the $i$-th column of $W_1$. The linear function $x \cdot w_i + b_i$ defines a hyperplane in the input space.

2. Each neuron's ReLU activation partitions the input space based on $x \cdot w_i + b_i = 0$, creating regions where the neuron is active ($x \cdot w_i + b_i > 0$) or inactive ($x \cdot w_i + b_i \leq 0$).

3. The hyperplanes from all neurons intersect to partition the input space into polyhedral regions, each with a fixed pattern of active/inactive neurons.

4. Within any region, the activation pattern is constant, so the output is $\sum_{i \in \text{active}} (x \cdot w_i + b_i) W_2[:, i] + b_2$. Since this is a linear combination of the input $x$, the function is linear within each region.

5. Therefore, the overall function is piecewise linear: linear within each region, with boundaries defined by the hyperplanes.

□

## A.2   Impossibility of Exact Multiplication

**Theorem 2** (Multiplication Impossibility). *No piecewise linear function can implement exact multiplication* $f(x, y) = xy$ *over an unbounded domain.*

*Proof.* Suppose, for contradiction, that some piecewise linear function $g$ implements exact multiplication over $\mathbb{R}^2$.

1. The multiplication function $f(x, y) = xy$ has mixed partial derivative $\frac{\partial^2 f}{\partial x \partial y} = 1$ everywhere.

2. Any piecewise linear function has zero mixed partial derivatives within each linear region (since linear functions have constant first derivatives and zero second derivatives).

3. For $g$ to equal $f$ everywhere, we would need $\frac{\partial^2 g}{\partial x \partial y} = 1$ everywhere, but this contradicts the piecewise linear property.

4. More constructively: consider any bounded region $R$ where $g$ is linear. Within $R$, we have $g(x, y) = ax + by + c$ for some constants $a, b, c$. But $ax + by + c \neq xy$ for all $(x, y) \in R$ unless $R$ contains at most finitely many points.

5. Since multiplication requires unbounded curvature (the function $xy$ becomes arbitrarily steep as $|x|$ or $|y|$ increases), no finite partition into linear pieces can approximate it exactly over an unbounded domain.

□

This impossibility extends to other non-linear operations requiring precise symbolic manipulation, establishing fundamental computational limits for FFN architectures in symbolic reasoning tasks.

## B   Appendix: Role Specialization Under Next-Token Prediction

We provide a detailed proof sketch for the role specialization theorem presented in Section 3.2.

### B.1 Setup and Notation

Consider a transformer with $L$ layers trained on next-token prediction. We denote:

- $\theta = \{\theta_E, \theta_A, \theta_F\}$: The three parameter sets for embeddings, attention, and FFNs respectively

- $\mathcal{D}$: Training distribution over token sequences

- $\mathcal{L}(\theta) = \mathbb{E}_{(w_1,\ldots,w_T)\sim\mathcal{D}}[-\log p_\theta(w_t|w_{<t})]$: The expected cross-entropy loss

- $x^{(L)} \in \mathbb{R}^d$: Final layer representation

- $E \in \mathbb{R}^{|V|\times d}$: Embedding/unembedding matrix

- $p_\theta(w_t|w_{<t}) = \text{softmax}(x^{(L)} \cdot E^T)[w_t]$: Next-token probability

### B.2 Gradient Flow Analysis

For arithmetic tasks like "$43 \times 78 = 3354$", the model must minimize:

$$\mathcal{L}_{\text{local}} = -\log p_\theta(\text{`3354'}|\text{`43} \times 78 =\text{'})$$

This requires $x^{(L)} \cdot e_{3354} > x^{(L)} \cdot e_v$ for all $v \neq 3354$ in the vocabulary.

**Gradient w.r.t. Final Representation.** The gradient of the loss with respect to the final representation is:

$$\frac{\partial\mathcal{L}}{\partial x^{(L)}} = E^T(p_\theta - y_{\text{onehot}})$$

where $p_\theta$ is the predicted distribution and $y_{\text{onehot}}$ is the one-hot encoding of the target token.

**Backpropagation Through Components.** Since $x^{(L)} = \text{embeddings} + \sum_{l=1}^{L} \Delta h_{\text{attn}}^{(l)} + \sum_{l=1}^{L} \Delta h_{\text{FFN}}^{(l)}$, the gradient flows to:

1. **Embeddings ($\theta_E$):**
$$\frac{\partial\mathcal{L}}{\partial\theta_E} = \frac{\partial\mathcal{L}}{\partial x^{(L)}} \cdot \frac{\partial x^{(L)}}{\partial\text{embeddings}} \cdot \frac{\partial\text{embeddings}}{\partial\theta_E}$$

    But embeddings must serve multiple contexts. For token "43", the embedding must minimize loss across:

    - Mathematical contexts: "$43 + 57$", "$43 \times 2$"
    - Date contexts: "43 BC", "1943"
    - Version contexts: "version 43.0"

    The multi-objective optimization forces:

    $$\theta_E^*(\text{"43"}) = \arg\min_e \sum_{c\in\mathcal{C}(\text{"43"})} \mathcal{L}_c(e)$$

    This averaging across contexts prevents the isometric properties needed for arithmetic (see Section 3.1).

2. **Attention ($\theta_A$):** At the "$=$" position with causal masking, attention computes:

    $$\Delta h_{\text{attn}} = \text{softmax}\left(\frac{QK^T}{\sqrt{d_k}}\right) V$$

where $Q$, $K$, $V$ are projections of $\{\text{"43", "}\times\text{", "78", "="}\}$.

**Key Constraint:** The output is a weighted average:

$$\Delta h_{\text{attn}} \in \text{conv}\{v_1, v_2, v_3, v_4\}$$

where $v_i$ are the value vectors. Since $e_{3354} \notin \text{conv}\{e_{43}, e_\times, e_{78}, e_=\}$, attention cannot generate the required output direction. It can only create rich encodings of the operation context.

3. **FFNs ($\theta_F$):** FFNs must map attention's encoding to the result. The required function is:

$$F : h_{\text{attn}}(\text{"multiply 43 by 78"}) \mapsto \text{direction}(e_{3354})$$

By UAT, FFNs with sufficient width can approximate any continuous function on compact sets. Define:

$$K = \{h_{\text{attn}}(a \times b) : (a, b) \in \text{training data}\}$$

For any $\epsilon > 0$, there exist weights such that:

$$\sup_{h \in K} ||FFN(h) - F(h)||_2 < \epsilon$$

**Critical Issue:** This approximation only holds on $K$. For novel arithmetic outside $K$, no approximation guarantee exists.

## B.3 Why Memorization Is Inevitable

**Architectural Impossibility.** FFNs implement $\text{FFN}(x) = W_2 \cdot \text{ReLU}(W_1 x + b_1) + b_2$, which is piecewise linear. Multiplication $f(a, b) = a \times b$ is not piecewise linear:

$$\frac{\partial^2 f}{\partial a \partial b} = 1 \neq 0$$

No weight configuration can make a piecewise linear function compute exact multiplication (see Theorem 2 in Appendix A).

**Training Dynamics.** Given $N$ training examples $\{(a_i \times b_i, c_i)\}_{i=1}^N$, gradient descent optimizes:

$$\theta_F^* = \arg\min_{\theta_F} \sum_{i=1}^N -\log p(\text{token}(c_i)|h_{\text{attn}}(a_i \times b_i))$$

Since exact computation is impossible, the optimization converges to memorizing the mapping:

$$h_{\text{attn}}(a_i \times b_i) \mapsto e_{c_i}$$

This is pattern storage, not algorithmic learning. The UAT capacity that could theoretically approximate multiplication is instead allocated to memorizing these $N$ discrete mappings.

**Out-of-Distribution Failure.** For novel multiplication $(a_{\text{new}}, b_{\text{new}}) \notin$ training:

- $h_{\text{attn}}(a_{\text{new}} \times b_{\text{new}}) \notin K$

- FFN has no memorized pattern

- Output is interpolation between nearest memorized patterns

- Result is essentially random from neighboring vocabulary tokens

### B.4 Conclusion

The role specialization is inevitable under gradient descent:

1. **Embeddings** must average across diverse contexts, losing arithmetic structure

2. **Attention** can only compute weighted averages, unable to generate novel tokens

3. **FFNs** cannot implement multiplication algorithmically, so they memorize patterns

This specialization emerges not from insufficient capacity—FFNs have universal approximation capability—but from how the next-token prediction objective allocates that capacity. The result is pattern storage rather than algorithmic computation, explaining why LLMs can articulate algorithms perfectly while failing to execute them reliably. □

## C Metacognitive Assessment Experiments

### C.1 Experimental Design

We tested three state-of-the-art models—Claude Sonnet 4, GPT-4o, and Gemini 2.5 Flash—on multiplication problems using "golden decomposition" prompts that provide explicit step-by-step procedural guidance. Models were constrained to use only internal capabilities without external tools or self-correction.

We evaluated two complexity levels: 5-digit numbers (range 10,000–99,999) and 10-digit numbers (range 1,000,000,000–9,999,999,999), with 20 problems per complexity level per model.

### C.2 Sample Prompt

The following shows a representative prompt structure used in our experiments:

```
Problem:  Calculate 93810 × 24592

Method:  Use place value decomposition to break this into simpler steps.

Step 1 - Decompose 24592 into place values:
24592 = 2 + 90 + 500 + 4000 + 20000

Step 2 - Calculate each partial product:
Step 1:  93810 × 2 = ?
Step 2:  93810 × 90 = ?
Step 3:  93810 × 500 = ?
Step 4:  93810 × 4000 = ?
Step 5:  93810 × 20000 = ?

Step 3 - Add all partial products:
Sum:  ?  + ?  + ?  + ?  + ?  = ?

Final Answer:  ?
```

This methodology eliminates planning uncertainty by providing the exact algorithmic steps, allowing us to isolate execution limitations from decomposition failures.

### C.3 Implementation Challenges

GPT-4o exhibited systematic processing limitations: when given the full batch of 20 samples, it repeatedly failed to complete the task. When reduced to batches of 5 samples, GPT-4o often omitted intermediate calculation steps, proceeding directly to final answers without showing the required step-by-step work. This behavior required individual problem presentation to obtain reliable step-by-step responses.

Claude Sonnet 4 and Gemini 2.5 Flash processed full batches reliably and consistently followed the decomposition structure as specified.

### C.4 Key Findings

The experiments revealed universal computational split-brain syndrome across all three model families. At 5-digit complexity, performance varied significantly: GPT-4o achieved 5% overall accuracy despite 95–100% step-wise accuracy, Gemini achieved 95% overall accuracy, and Claude achieved perfect performance.

At 10-digit complexity, all models exhibited complete breakdown (0% overall accuracy) but through different failure modes. GPT-4o showed degraded step-wise accuracy (76–100%), while both Gemini and Claude maintained excellent step accuracy (95–100%) but failed completely at final summation.

These results provide direct empirical evidence for the instruction-execution disconnect: models can follow algorithmic procedures perfectly yet systematically fail at computational implementation, demonstrating fundamental limitations in bridging comprehension and competence even under optimal procedural guidance.

