# OpenReview forum: "Comprehension Without Competence: Architectural Limits of LLMs in Symbolic Computation and Reasoning"
_TMLR — Accepted by TMLR_

### Review · Reviewer_Jgku · 2025-08-13

**Summary Of Contributions:**

The paper core contribution is introducing the split-brain syndrome in LLM/LRMs: the fact that they know an algorithm but they cannot execute it reliably. Specifically, the authors argue that there are architectural limitations on current models that prevent them in really solving this issue by three main claims: token embedding limitations, feed-forward networks limitations, instruction-execution disconnection.

I really enjoyed the paper. It well written, it has a very nice story, it is sound, it has nice visualisations and insights for each of the claims. Limitations are stated clearly at the beginning. Related works are up to date and well discussed.

I just have one minor complain: some things are not very well introduced. Specifically, equations have terms that are not defined and it is not clear what they are (e.g. already eq 1 and 2, only C(w) is explicitly defined, but also for the other equations there are similar issues). It is perhaps not fundamental for the understanding of the paper to have all the details, but it would be nice to have a more formal definition of the terms and an intuitive explanation next to them.

Minor typos: sometimes the wrong character to close quotes is used (e.g. end of page 12, should be `` in latex). Sometimes citet and citep are misplaced (e.g. “(Vashishtha et al., 2025) demonstrates success on causal reasoning” should be \citet, and “similar to distributional semantics models like Word2Vec Mikolov et al. (2013).” should be \citep).

Finally, in the limitations of FFNs, I wonder what is the relationship with works on the computational complexity limitations of transformers. For example [1] where they demonstrated that multi-layer transformers cannot solve problems such as 2-SAT, Horn-SAT, etc. Or [2] where they show that a T-chain of thought enables transformers to solve problems corresponding to Boolean circuits of size T. Even though this second might be related more with the self scaffolding part? Maybe the connections between these things can be made more explicit at the beginning of 2.4 where it’s already hinted?

[1] On limitations of the transformer architecture (Peng et al 2024)
[2] Chain of thought empowers transformers to solve inherently serial problems (Li et al 2024)

**Audience:**

Yes

**Audience Explanation:**

The topic of the paper is very relevant these days, and it offers an interesting (new) perspective.

**Claims And Evidence:**

Yes

**Claims Explanation:**

Every claim has a proof in the appendix or proper references / experiments.

**Requested Changes:**

Fix background and typos (see review).

---

> ### Author Response · Authors · 2025-08-20
> **Response to Reviewer Jgku**
>
> Thank you for the thorough review and for catching the missing equation definitions and citation formatting issues. We will fix all of these in the revision.
>
> ### Computational Complexity Connections
>
> Thank you for suggesting Peng et al. (2024) and Li et al. (2024) - these papers align well with our "Expressivity and Computational Boundaries" discussion in Section 2.1. Taken as a whole, they explore the theoretical upper limits of what Transformers can compute as abstract devices in a clean-room setup, while our work examines LLMs as a Transformer-based real-world application with its own training-induced constraints. The gap between theoretical capacity and practical capability is interesting to explore.
>
> For example, these theoretical works expose fundamental limits by showing:
> - Transformers cannot compose functions reliably when domain sizes exceed embedding dimensions (Peng et al.), and also cannot solve problems like 2-SAT under complexity assumptions
> - But with chain-of-thought, they can simulate Boolean circuits of size $T$ using $T$ intermediate steps (Li et al.)
>
> These papers invest considerable effort showing Transformers can (or cannot) compute problems that conventional devices handle trivially. The question is the real-world relevance given the complete work that computer science as a whole can already offer. For example, the function composition problem in Peng et al. (2024) recalls a typical join operation in databases, and a standard computer solves Boolean circuits instantly, yet Li et al. must carefully construct positional encodings to achieve the same with Transformers.
>
> In our view, LLMs are Transformer-based, general purpose ML devices that are fundamentally massive, multi-level pattern completors. The main point of this paper is to reveal that they are duly challenged to perform symbolic computations, and require elaborate workarounds.
>
> We will update Section 2.1 to make the above points more clear.
>
> ### Equation Definitions and Minor Corrections
>
> [As mentioned, we will properly define all variables in equations (1) and (2), fix quotation marks to use proper LaTeX formatting (\`\` and ''), and correct \citet vs \citep usage throughout.]

---

### Review · Reviewer_UthV · 2025-08-14

**Summary Of Contributions:**

This submission attempts to advance our understanding of the
limitations of LLMs. It is important to note that here the focus is
only on "pretrained transformer-based LLMs trained on next-token
prediction objectives over natural language corpora, without access to
external tools or explicit reasoning scaffolds."

The authors appear to know the literature well. (I cannot be sure
about this, since I do not work in this area, so it could be that
important contributions of which I am unaware have been missed.) In
any case, there are many summaries and analyses of interesting and
relevant work done by others. The quality of the writing is good:
clear and without grammatical errors/typos etc.

The results of ARC-AGI-2 show us that there are certain types of task
that LLMs are very poor at. The goal of the current paper is not just
to provide examples of LLM failure, but to explain what it is about
LLMs that lead to such failures. The authors' explanation is given by
their 3 main claims (p. 2).

A key point of the paper is that the 3 problems identified in the 3 claims are jointly necessary for observed LLM problems.

**Additional Comments:**

MINOR POINTS

p6: (1) and (2) are not meant to be a formal definition of how "The
training objective optimizes embeddings through next- token
prediction:" but I think "E" should be defined. Also what's the
connection between (1) and (2).

The Shojaee et al reference is wrong: wrong arxiv id.

**Audience:**

Yes

**Audience Explanation:**

The main issue here is significance. The authors state that:
"... while previous studies document what [LLM] failures occur across
domains, we provide a principled mechanistic explanation for why they
occur..." Of course, it is not true that there is *no* existing work
aiming to explain limitations of LLMs. Some of the papers referenced
here attempt at least some explanation of their empirical
findings. In addition, Section 2.1 explicitly summarises a number of papers providing explanations.

However, I am not aware of other work that does as much of
a thorough job as the current paper (but I am not an expert on this
literature, so it's quite possible there are such papers I don't know
about). Assuming that the current paper has little competition in
terms of thorough explanation of LLM limitations, there remains the
issue of the paper's narrow focus. Perhaps we don't care too much
about the limitations of "pretrained transformer-based LLMs trained on
next-token prediction objectives over natural language corpora,
without access to external tools or explicit reasoning scaffolds."
since to get closer to AGI we would not artificially restrict
ourselves to such systems.

**Broader Impact Concerns:**

I do not have 'concerns' (as in worries) about the ethical implications of the work. I think the ethical implications are positive since the paper points to problems that can occur by being too trusting in LLMs.

**Claims And Evidence:**

Yes

**Claims Explanation:**

I am persuaded that the 3 claims are correct. The first two
essentially state that there are problems using 'continuous' reasoning
for essentially 'discrete' tasks (e.g. 'binding' and exact symbolic
operations). The third points to the LLM's 'flat' representation of
knowledge: we have no types and therefore (for example) a distinction
between executing and describing algorithms is unrepresentable. Of
course, there is a trade-off here. It is precisely by reducing
everything to a single continuous optimisation task that leads to much
of what is so impressive about LLMs.

Although the 3 claims are not the sort that can be 'proved' in the
mathematical sense, the authors take each in turn and provide
supporting evidence and reasoned argument to support them. I
particularly liked the investigation of geometric separation relevant
to Claim 3.

**Requested Changes:**

Assuming that the argument advanced here is indeed original, I am surprised that it (or something similar or a competing but still comprehensive explanation) has not previously been stated, given the intense interest there is in LLMs. If the authors can provide an explanation for this surprising fact that would strengthen the paper.

I would like a stronger argument for the limitations mentioned above. Of course, imposing such limitations make the analysis more straightforward. But, as mentioned earlier, perhaps there is diminishing interest in these particular sorts of LLMs.

---

> ### Author Response · Authors · 2025-08-20
> **Response to Reviewer UthV**
>
> Thank you for your thoughtful review and positive assessment. We appreciate your recognition of our thorough mechanistic explanation.
>
> ### Why This Analysis Hasn't Been Done Before
>
> You raise an excellent question. From our perspective, several factors contributed:
>
> 1. **Benchmark-driven progress**: The community has focused on establishing and pushing benchmarks. This empirical approach has been productive but sometimes obscures deeper architectural questions.
>
> 2. **The "OOD" framing**: Extensive work has exposed LLM brittleness (Section 2.2), but the dominant explanation has been "out-of-distribution" problems. Yet why should grade-school arithmetic have an "OOD" problem? This framing applies a narrow ML lens rather than questioning whether the architecture matches the task.
>
> 3. **Surface symptoms vs. root causes**: The "9.11 vs 9.9" comparison became a viral anecdote about LLM confusion, emerging later in our study as a crystallizing example. Discussions focused on output errors rather than recognizing that the issue begins with input representations—tokens carrying contextual contamination that prevents clean symbolic binding.
>
> 4. **Missing theoretical framework**: Our journey began with puzzling over problems like the Reversal Curse and Alice-in-Wonderland. These seemed like isolated curiosities until we recognized they require lifted reasoning—applying rules over arbitrary entities rather than memorizing patterns. This insight led us to investigate LLMs' symbolic computation capabilities more broadly.
>
> 5. **Confluence of insights**: Once we recognized that contextual averaging (input), architectural impossibility (computation), and instruction-execution disconnect (training) form an interdependent system of constraints, the unified picture became clear. The three constraints emerged last in our research but are positioned first in the paper as they're logically foundational.
>
> We acknowledge we may still be missing important developments. But we believe this synthesis helps explain why seemingly simple tasks remain unreliable despite massive scale, and why compensatory strategies have become essential.
>
> ### Why Focus on Restricted LLMs
>
> Your point about our narrow focus is well-taken. We chose to analyze "pure" LLMs without external tools for several reasons:
>
> 1. **Foundational understanding**: To understand why compensatory strategies are necessary, we must first understand what breaks without them. Our analysis explains WHY tool use, scaffolding, and hybrid architectures have become essential—they're necessary workarounds for fundamental limitations.
>
> 2. **Practical implications**: Despite tool availability, most LLM deployments rely heavily on base model capabilities, especially where external verification isn't available. As discussed in Section 5, achieving full accuracy almost certainly requires tool calling coupled with reliable metacognitive capabilities. Understanding base limitations clarifies both why this metacognition is essential and why it remains unreliable—the same architectural constraints that prevent symbolic computation also limit self-assessment.
>
> 3. **Broad applicability**: Our findings apply to any system using transformer architectures with next-token prediction—not just pretrained models but also instruction-tuned and tool-calling variants. The split-brain syndrome persists across all configurations; it just gets masked by increasingly sophisticated workarounds. We suspect these lessons extend to LRMs as well, though this requires further investigation.
>
> 4. **Mechanistic interpretability implications**: Perhaps most soberly, our work suggests limits to mechanistic interpretability efforts. If models solve arithmetic through hierarchical pattern storage rather than implementing algorithms, then tracing "circuits" for symbolic computation may be pursuing something that doesn't exist in the expected form. This redirects interpretability efforts toward understanding pattern coordination rather than searching for non-existent algorithmic circuits.
>
> We agree AGI won't artificially restrict itself this way, but understanding these restrictions motivates the architectural innovations that might lead there.
>
> ### Technical Corrections
>
> - We'll properly define $E(c \setminus w)$ in Equation 2 as "embeddings of context $c$ excluding token $w$" and clarify the connection between equations (1) and (2): Equation (1) shows the standard next-token prediction loss, while Equation (2) demonstrates how this loss forces each token embedding to optimize across all its contexts, creating the averaging problem.
> - We'll fix the Shojaee et al. reference—thank you for catching this error.
>
> Thank you again for your insightful review. Your questions have helped us better articulate why this analysis matters even as the field moves toward augmented systems.

---

> > ### Author Response · Authors · 2025-08-20
> > **Additional Comment on the Trade-off**
> >
> > The reviewer astutely notes: "Of course, there is a trade-off here. It is precisely by reducing everything to a single continuous optimisation task that leads to much of what is so impressive about LLMs."
> >
> > We strongly agree and would like to expand on this crucial point. The very features that make LLMs so powerful are also the sources of their limitations:
> >
> > 1. **Unified input encoding**: Unlike conventional algorithms that require typed data structures (integers, strings, floats, etc.), LLMs process everything through a single token embedding mechanism. This universality enables their impressive generality but also creates the contextual contamination we identify.
> >
> > 2. **Single-minded objective**: Next-token prediction provides a scalable training objective that works across all tasks and domains. This same simplicity, however, treats instruction and execution as equivalent pattern completion tasks, creating the geometric separation we demonstrate.
> >
> > In retrospect, these potentials and trade-offs were already present in Bengio et al.'s neural language models from 2003 (which are framework-wise more similar to Transformers than the RNN variants the community subsequently focused on). It was the ingenious design of the transformer architecture—particularly self-attention and parallelizability—that finally unlocked these potentials at scale.

---

### Review · Reviewer_yB2q · 2025-08-17

**Summary Of Contributions:**

This paper studies the brittleness of LLMs, and transformer architectures in general, at solving reasoning and mathematical computation tasks, and the phenomenon whereby LLMs fail such tasks but are able to correctly explain the steps one should take to perform them.

The paper primarily makes three claims about how architectural features of LLMs result in LLMs being incapable of performing reasoning and computation tasks, even when they can still explain the steps needed for performing them. The claims are (paraphrasing):
- That contextual information in LLM token embeddings makes symbolic computation difficult by inhibiting appropriate abstraction of input variables
- MLPs cannot perform symbolic computation effectively because, with finite units, they can only approximate and not perfectly match certain functions
- Predictive learning results in pattern matching that does not learn to properly associate computations and instructions for how to perform them

Notably, the paper argues that all three LLM weaknesses, mentioned in each claim, are inter-related and important for the disconnect between computation and description of computation. To back up the three claims the paper relies heavily on extensive citations, one mathematical proof, and 4 experiments.

The paper concludes that, among other things, pattern completion algorithms like LLMs will not be able to succeed at learning generalizable rules, as is required by symbolic tasks, without architectural changes.

## Strengths
The main strengths of the paper, in my view, are some of the main observations about failure modes of LLMs. For example:
- The phenomena of LLMs being able to describe the steps needed to perform a task but not being able to perform the task itself is interesting, and a worthy topic of discussion. Relatedly, the demonstration that an representation for describing a task is more similar to representations for describing other tasks than for performing the same task is a cool result
- It is interesting to consider a potential downside of LLMs not being able perfectly approximate certain function, and that contextual information encoded in embeddings might actually be problematic for abstract tasks

It is also nice to read a paper where there are very few spelling/grammar errors

## Weaknesses
There are three main weaknesses that seem rather severe. I have listed them below, referencing specific problems that I mention in the "requested changes" section later in the review:
- The claims seem incompletely backed up (requested changes 2, 3, 5, 10, 16, 17)
- The organization of the paper seems confusing at times, and appears to contain redundancy (requested changes 4, 9, 12, 22, 23, 26)
- Some of the language choices are confusing and could accidentally mislead readers (requested changes 1, 6, 8, 25)
- Some of the references are used to justify points that I don't believe they truly justify (14)

**Additional Comments:**

While I have organized the review around specific "requested changes", I believe that the paper's weaknesses are severe enough to warrant its rejection. If the other reviewers agree, hopefully the requested changes could be used to help make adjustments for a future submission of the paper, if the authors decide to go that route.

I would also like to note that surveying past results seemed to take an almost larger role in the paper than novel results. For this reason, if the authors decide to restructure things it might make sense to consider writing more of a survey (if the reference issues mentioned in point 14, above, are resolved). Still, a couple of the results (see strengths section of the review) seem like they could be used in a research paper if they were elaborated upon.

**Lastly, I will mention that my experience is moreso in deep learning than in the study of LLMs per se. For this reason my knowledge of literature related to the submission is not as good as it should be.**

**Audience:**

Yes

**Audience Explanation:**

Given that the paper is on brittleness of LLMs it is certainly in a relevant enough domain for readers of TMLR.

**Broader Impact Concerns:**

Given that the paper is on LLMs, coupled with the diverse and sometimes quite damaging effects that LLMs are having on society (environmental, misinformation related, surveillance and military purposes, etc), it would be really great to have broader impacts mentioned somewhere. If not in the main paper, then at least in the appendix.

**Claims And Evidence:**

No

**Claims Explanation:**

I do not believe that the three main claims, or the claim of their interconnected nature, are sufficiently justified. I also believe that some of the results provided are not entirely relevant to the arguments of the paper. See requested changes for full details (e.g. points 2, 3, 5, 10, 16, 17)

**Requested Changes:**

I have added “**C**” at the end of changes that I view as critical and “**S**” after ones that I view more as suggestions.

1. [Abstract/Introduction]: there are two choices of language that I believe might end up misleading people, and that I would like to see changed. First, the use of “split-brain”. As mentioned later in the paper (end of page 1/start of page 2: “artificial and biological intelligence may operate under fundamentally different principles.”), it seems strange to use a biological analogy for a non-biological phenomenon. This seems to be a particularly confusing choice given that certain AI algorithms are actually used to model biology (e.g. certain RNNs), and that other academics have noted problems with anthropomorphizing LLMs. Second, the use of “comprehension” in the abstract and introduction also seems somewhat misleading. Comprehension is a rather cognitively loaded term, and as mentioned later in the paper, it seems LLMs are likely doing pattern completion to solve problems, which to my understanding does not necessarily imply comprehension. **C**
2. [Section 3 and throughout] It is mentioned in several places in the paper that the 3 claims are interconnected, or reinforce each other, but I don’t think that sufficient evidence is provided to support this. From reading the paper, it seems to me that each claim presents a problem that could potentially be an issue for abstract computation in its own right, and could be addressed independently of the other problems. Could the authors provide more justification for the interconnected nature of the claims? **C**
3. [Section 3.2] In section 3.2 it is mentioned: (1) “The generation of precise computational results must ultimately occur through the feed-forward networks, which apply position-wise transformations to produce new content.”, and (2) “For arithmetic queries, the required computational results aren’t directly available in the attention-processed representations and must be computed—this is where the FFN serves as the primary computational engine.” To my understanding, self-attention layers can implement rather non-trivial nonlinear computations without needing to stack MLPs between them. If the authors could provide further justification for these claims, either with citations or otherwise, that would be great. **C**
4. [Why Pattern Storage Wins on page 9; Experimental Validation of Hierarchical Pattern Assembly on page 10] I don’t believe these sections contribute much to the paper. A single residual layer is a universal function approximator because of the MLP. Therefore even just one transformer layer should be able to arbitrarily approximate any function on compact support–which means if approximation will suffice for a given problem then a single layer will, at least in theory, suffice. Multiple layers will perform even better, and I don’t understand why the iterative computation that multiple layers provide is a problem. If one needs a perfect, exact result, then these sections still seem redundant because the “Architectural Impossibility of Exact Computation” section shows that you cannot have this with current deep learning. Please correct me if I’m seeing things incorrectly; otherwise I think these sections are not necessary. **C**
5. [Figure 3] Given that t-SNE is very affected by hyper-parameter settings I would ideally like to see this plot separately verified with another dimensionality reduction algorithm. **C**
6. [Throughout paper]: The discussion of “embeddings” in the issue of averaging mentioned in “Claim 1” seems a bit ambiguous to me. These are frequently referred to as “token embeddings”, which I had thought are usually fixed during much of the training of LLMs. However, in section 3.1 it is suggested that the problem with averaging derives from token representations being learned via next-token prediction. Relatedly, I’m a little bit confused as to whether the embeddings mentioned in Figure 1 are token embeddings or are representations taken from intermediate layers. I’d appreciate it if the authors could clarify these things. **C**
7. Previous work has studied architectures where abstract computation is separated from sensory details, potentially allowing better performance on reasoning tasks and tasks involving binding of abstract variables (On Neural Architecture Inductive Biases for
Relational Tasks – Kerg et al. 2022; Emergent symbols through binding in external memory – Webb et al. 2021). These seem potentially relevant to the discussion of solutions to the problems discussed in the paper and could be worth citing. **S**
8. [Line 3, Page 2] When the authors say “this architectural difference” it seems to refer to pattern completion optimized by next-token prediction, but this isn’t an architectural choice, is it? It seems more algorithmic to me **C**
9. [Page 2] The first set of bullet points seem a little redundant given the second set. Could this be condensed into a single set of bullets? **S**
10. [claim 3] why is next token prediction the critical ingredient for separate representations? Could it not be that in some layer representing a higher level of abstraction the representations of description and computation might merge, even when trained with next token prediction? It seems that the authors focus on showing that the representations are separate but do not connect next token prediction as being responsible for this. I would thus like to see more justification here **C**
11. [page 3] first time mentioning “scaffolding strategies” without defining them. Could be useful to give some description at the first mention instead of at the second mention later on the same page **S**
12. [Section 2] I find section two a little bit confusingly organized. I think the two main reasons for this is that similar topics are returned to more than once–for example “Training Dynamics and Path Dependence” in section 2.2 and “Training-Order Dependencies and Pattern Storage” in section 2.3–and that the headings don’t seem to be organized with a logical flow in mind. To the latter point, I think one way that flow could be improved is by organizing previous literature based on how it applies to each of the three claims (e.g., having a heading for each claim). **S**
13. [End of intro and section 2.5] Contributions are listed twice, which seems a little redundant. Perhaps have contributions and positioning listed only after related work is discussed? **S**
14. [“Training and Path Dependence” section, page 4]: in this section the authors list takeaways from the Power et al. 2022 and Tigges et al. 2024 papers that appear different from the content of these papers. In Power et al. they observe Grokking, where the trained networks perfectly fit simple computational tasks, exhibiting full generalization. I could not find mention of the “path dependent artifacts” that the authors of the current paper list as a takeaway. The results of Tigges et al. do suggest that “while computational circuits emerge consistently across scale, their specific implementations vary significantly during training”, however it is not clear the takeaway, listed by the authors of the current manuscript, that “apparent “reasoning circuits” may reflect training-specific pattern coordination rather than universal computational principles”, is supported by the results of Tigges et al. I could imagine that computational principles might remain the same even while circuits implementing them change. Could the authors address these apparent inconsistencies? **C**
15. [Equations (1) and (2)] the variables in the equations are not defined! Also the equation doesn’t seem to contribute to the paper. Suggest either defining variables and elaborating, or removing it. **C**
16. [Section 3.1] The authors write: “As Equation 2 predicts, contextually-averaged embeddings resist the domain binding required for stable symbolic circuits.” Could the authors elaborate please? It is not immediately clear to me why this should be. **C**
17. [Section 3.2] Claim 2 seems to rely on the statement by the authors that “symbolic computation requires exact results across potentially unbounded inputs”, and therefore the approximation abilities of modern deep learning methods are a problem because, without infinite units, they can only provide approximate results. It is interesting to reflect upon this point, but I’m not sure that it’s relevant for the kinds of real world applications for which LLMs are used. I would like to see more justification for why we need exact results with LLMs. **C**
18. [Section 3.2] It seems obvious to me that ReLU MLPs can only approximate continuous functions, so I’m not sure that a proof is required for this. Also the piecewise linearity proof in the appendix seems perhaps unnecessary for the same reason **S**
19. [Table 2] Are these results repeated from the Li et al study? If so, I think this should be mentioned in the title or caption of the table. **C**
20. [“Functional Specialization Architectures” section, Page 20] This section needs citations **C**
21. [Table 3] This table could use better explanations. E.g., explaining the “sum error” row in the main text, the relevance of the fractions listed after performance. Also, why is it “30.125” and “30.1210” in column 1 and not simply “5” and “10”? **C**
22. [Section 6, lines 6-11] it seems like this section just repeats the main points of “comprehension vs competence” but as “general vs generalizable”. Perhaps redundant? **S**
23. [Appendix B.4] If I’m not mistaken, isn’t the content of this section mentioned in the main paper? If so, this seems not necessary to include **S**.
24. [Section 2.2, paragraph 2] There are places in the paper where author names are repeated before a citation and then within a citation–e.g. Dziri et al. (Dziri et al., 2023). Perhaps there could be a way to avoid repeating the same information twice? **S**
25. [Section 2.2 paragraph 3 line 1] Is “execution limitation” the “comprehension-execution split”? **S**
26. [4 lines after Eq. 3] could you elaborate on what you mean by “through weight configuration alone”? **C**

---

> ### Author Response · Authors · 2025-08-20
> **Part 1**
>
> **Change 1:** Concerns about terminology - "split-brain" seems inappropriate for non-biological phenomenon; "comprehension" is cognitively loaded and pattern completion doesn't imply comprehension.
>
> **Response:** Split-brain syndrome: In neuroscience, this refers to patients whose brain hemispheres cannot coordinate; the term also exists in distributed systems for similar coordination failures. We chose the neuroscience version as it's more widely recognized and has shorter semantic distance to the AI research community. While in both neuroscience and distributed systems the root cause is faulty communication between subsystems, in LLMs the separation is functional: two capabilities (explaining vs. executing algorithms) operate independently without coordination. We can add further clarification to prevent confusion.
>
> Comprehension: We agree this deserves clarification. Our usage is behavioral—from the perspective of an observer evaluating any intelligent entity, human or machine. As Feynman noted, 'If you can't explain it simply, you don't understand it' (often paraphrased as explaining to a child). This captures our definition: when an LLM flawlessly explains decimal comparison algorithms, observers reasonably judge it as 'comprehending' the procedure.
>
> Our argument is that LLMs achieve both explanation (comprehension) and execution (competence) through pattern completion, but these emerge as geometrically separated capabilities that cannot coordinate. The model 'comprehends' (can explain) yet lacks 'competence' (cannot execute)—both from pattern matching, but dissociated. This reversal of Dennett's 'competence without comprehension' highlights how LLMs differ from biological intelligence (though comprehension without competence does exist in that domain as well). We'll clarify this behavioral definition framework earlier in the paper.
>
> **Change 2:** The 3 claims are mentioned as interconnected but insufficient evidence is provided. Each claim could be addressed independently.
>
> **Response:** You're correct that each identifies a distinct problem, but they must be understood together to explain the central split-brain syndrome, capturing simultaneously why execution fails (the "without competence") and why this failure coexists with perfect explanations (the "split"):
>
> Our narrative starts by examining why execution fails, despite perfect explanations:
>
> **Claims 1 & 2 (Execution failures):** When encountering execution instances (e.g. '9.11 > 9.9?'), contextual averaging prevents clean and automatic mathematical binding (Claim 1). Even if binding were perfect and numerical properties are preserved, FFNs cannot implement exact operations—they resort to pattern storage (Claim 2). Together, these establish that reliable execution is impossible.
>
> Then explain the disconnect:
>
> **Claim 3 (The disconnect):** Next-token prediction provides no mechanism for well-learned instructions to guide execution. Both capabilities emerge as separate pattern-matching pathways with no automatic binding between them—instructions and executions are parallel sequences for the model to learn. Behavior-wise, the model thus can explain perfectly and fail at execution according to instructions—the split-brain syndrome.
>
> We will think of ways to make the narrative structure clearer earlier in the paper. The main point is raising awareness of this systemic issue—compensation strategies (Section 5) can only partially address it.

---

> > ### Author Response · Authors · 2025-08-20
> > **Part 2**
> >
> > **Change 3:** FFNs as primary computational engine needs justification - self-attention can implement non-trivial nonlinear computations.
> >
> > **Response:** Let us explain the division of labor more clearly.
> >
> > In LLMs, attention is causal, meaning each token can only attend to previous tokens, not future ones. Consider computing '23 × 17 = 391'. When the model processes the '=' token, it can only see '23 × 17 =' and must start producing '391'. At this point:
> >
> > Attention computes a representation that encodes the multiplication operation and its operands by taking weighted averages of previous tokens (i.e., a representation of '23 × 17'). But attention alone cannot generate '391'—it can only produce linear combinations of existing token representations.
> >
> > The novel value '391' must come from non-linear transformation, which is the role of FFNs. As shown in prior work (Geva et al., 2021; Dai et al., 2022; Meng et al., 2022), FFNs serve as the primary knowledge repository and value generation mechanism.
> >
> > In a single-layer transformer, this is straightforward. In deeper networks, each layer builds on the previous: at layer L, with the same causal attention structure, the '=' position receives the partial result from layer L-1 plus attention-processed operator/operand information. FFNs at each layer contribute to gradually approaching the answer through residual fitting (Figure 2).
> >
> > Our clarification: FFNs are the primary value generation engine—only they can produce fundamentally new values like '391'. Attention provides contextual processing—identifying what computation is needed. Both are essential, but serve distinct roles.
> >
> > **Change 4:** Pattern Storage and Hierarchical Pattern Assembly sections seem redundant given universal approximation and impossibility of exact computation.
> >
> > **Response:** These sections directly connect to our response in Change 3—they empirically demonstrate the iterative residual fitting process we described there.
> >
> > While a single layer can theoretically approximate any function, these sections show HOW transformers actually solve arithmetic: through hierarchical pattern assembly across layers. Figure 2 provides direct neural evidence of this mechanism, showing layer-by-layer convergence toward answers through coordinated pattern retrieval.
> >
> > Transformers are notoriously difficult to understand beyond single-layer analysis, and our claims about multi-layer behavior could be challenged without analysis and empirical evidence. The progressive, layer-by-layer residual fitting we demonstrate is crucial because this is precisely what enables scaling—depth allows models to decompose complex patterns into manageable pieces, with each layer contributing incremental refinements. This explains both why deeper models perform better and why they still fail on novel patterns outside their hierarchical decomposition repertoire.
> >
> > These sections bridge our theoretical impossibility (what can't happen) with empirical observation (what does happen instead). Without them, readers would know exact computation is impossible but not understand the sophisticated pattern-storage mechanism that emerges as a consequence. This mechanism is central to understanding why models succeed on familiar patterns yet fail on novel ones—a key aspect of the split-brain syndrome.
> >
> > **Change 5:** t-SNE is sensitive to hyperparameters - should verify with another dimensionality reduction algorithm.
> >
> > **Response:** We acknowledge that t-SNE is sensitive to hyperparameters. We have verified our results across different random seeds and consistently observe the same separation pattern.
> >
> > We agree that additional dimensionality reduction methods (e.g., UMAP, PCA) would strengthen the visualization. However, the key finding—that instruction and execution representations are geometrically separated—is supported not just by the visualization but also by the quantitative analysis in Figure 4 (showing large cosine distances between instruction and execution clusters).
> >
> > We will explore additional visualization methods if resources permit (we experienced a recent workplace change), noting that the geometric separation finding is robust across our current analyses.

---

> > > ### Author Response · Authors · 2025-08-20
> > > **Part 3**
> > >
> > > **Change 6:** Ambiguity about "embeddings" - are these token embeddings or intermediate representations?
> > >
> > > **Response:** We mean the learned token embeddings at the model's embedding layer (first layer), not intermediate representations, as is commonly understood in the LLM community. Embeddings are intrinsic model parameters—you cannot swap embeddings between models. They're shaped by each model's specific training dynamics and corpus. To our knowledge, modern LLM training doesn't freeze embeddings partway through; they continue learning throughout training.
> > >
> > > Figure 1 shows embeddings after full training. During training, each occurrence of a token like '9.11' updates its embedding based on that specific context—mathematical, historical, or versioning. Over many training steps, the embedding becomes optimized as a weighted average across all these contexts (Equation 2). The contamination isn't added later; it emerges from the training objective itself.
> > >
> > > **Change 7:** Relevant work on architectures separating abstract computation from sensory details (Kerg et al. 2022; Webb et al. 2021).
> > >
> > > **Response:** We agree that binding is critical to abstract reasoning.
> > >
> > > Unlike these two papers, our work addresses what seems like a simpler problem: single modality (all tokens) rather than vision-to-relations. Yet this is paradoxically harder—with vision, there's a natural boundary between sensory and abstract processing. In language, everything is already, in a way, symbolic: where do you separate '9.11' the number from '9.11' the date, or instructional text from executable procedures? They are all 'symbols' to the model.
> > >
> > > The cross-modal separation in these papers may actually explain why abstraction emerges naturally there. 'The Forgetting Machine' (Rodrigo Quian Quiroga, 2017) suggests that abstraction arises through compression across vastly different sensory experiences that share abstract structure—a principle these architectures exploit. The emergence of abstraction is a fascinating topic that interests us, but we omit it as beyond the scope of this paper.
> > >
> > > BTW, we note that modern multimodal LLMs now solve the tasks in these papers easily (tried the tasks in Fig1 of the ESBN paper on GPT); they seem to be much simpler than ARC-AGI-1 (Section 6).
> > >
> > > **Change 8:** "Architectural difference" refers to pattern completion by next-token prediction, which is algorithmic not architectural.
> > >
> > > **Response:** You're correct that next-token prediction is a training objective rather than an architectural choice.
> > >
> > > Our intended contrast was between symbolic systems (which bind tokens to abstract roles and apply rules) versus the entire transformer-based LLM approach (architecture + next-token prediction training). The 'difference' encompasses both the transformer architecture and how it's trained, not just architecture alone.
> > >
> > > We'll revise this to 'This fundamental difference' or 'This design difference' to avoid the confusion between architectural choices and training objectives.
> > >
> > > **Change 9:** First set of bullet points seems redundant given the second set.
> > >
> > > **Response:** The first set presents the three constraints that create the computational split-brain syndrome, while the second set outlines our claims with supporting evidence for each constraint. We see your point about potential redundancy—the distinction between 'constraints' and 'claims about those constraints' may not add enough value to justify the repetition.
> > >
> > > We'll consider condensing these into a single, clearer set of points that combines both the constraints and their evidence. This would improve flow and reduce redundancy.
> > >
> > > **Change 10:** Why is next-token prediction critical for separate representations? Could representations merge in higher layers?
> > >
> > > **Response:** Next-token prediction provides no incentive to link instructional knowledge to execution. The model learns to complete instructional patterns ('To multiply...') and execution patterns ('23 × 17 =') as separate text completion tasks, both of which must reach the final decoding layer. Even if representations merged in higher layers (which we doubt), there's no mechanism to use instructions to guide execution.
> > >
> > > At inference, '56 × 76 =' triggers pattern completion through residual fitting (Figure 2), not algorithmic execution guided by instructional knowledge. Unlike architectures with explicit recombination mechanisms, transformers lack the machinery to use one representation type to guide another—they just complete patterns.
> > >
> > > We'll strengthen this causal argument in the revision.

---

> > > > ### Author Response · Authors · 2025-08-20
> > > > **Part 4**
> > > >
> > > > **Change 11:** "Scaffolding strategies" mentioned without definition.
> > > >
> > > > **Response:** You're right that we should define 'scaffolding strategies' at first mention to help readers unfamiliar with LLM terminology.
> > > >
> > > > These are techniques where models generate intermediate steps to guide their reasoning. Chain-of-thought (CoT) is the primary example—instead of directly answering '23 × 17 = ?', the model first generates 'Let me multiply: 23 × 10 = 230, 23 × 7 = 161, 230 + 161 = 391'. The model 'scaffolds' its own computation through explicit intermediate steps.
> > > >
> > > > **Change 12:** Section 2 organization confusing - similar topics returned to, headings lack logical flow.
> > > >
> > > > **Response:** We structured Section 2 to move from theoretical foundations → empirical failures → mechanistic explanations → compensatory strategies, grouping papers by their primary contribution rather than strict topic boundaries.
> > > >
> > > > We acknowledge that some themes recur across subsections (e.g., training dynamics appears in both 2.1 and 2.3) because different papers approach similar issues from different angles—theoretical vs. mechanistic.
> > > >
> > > > We'll consider adding clearer transitions between subsections or providing a roadmap to improve flow while preserving the current structure that we believe effectively builds the narrative arc.
> > > >
> > > > **Change 13:** Contributions listed twice (intro and section 2.5).
> > > >
> > > > **Response:** We intentionally placed contributions in both locations for readability: the introduction provides a roadmap before the extensive related work section, while 2.5 positions our contributions specifically against the literature just reviewed.
> > > >
> > > > Given the breadth of related work covered, we felt readers benefit from knowing our contributions upfront, then seeing how they relate to existing work. However, we understand the redundancy concern and will consider consolidating to a single location after the related work section.
> > > >
> > > > **Change 14:** Interpretation of Power et al. and Tigges et al. papers seem inconsistent with their content.
> > > >
> > > > **Response:** Path-dependency in training is theoretically inevitable: different corpora, data ordering, batch compositions, and training schedules necessarily lead to different gradient updates and thus different internal implementations. For any given sample (instruction or execution), gradients modify different parameter regions depending on training history. A controlled experiment is possible but exceedingly expensive.
> > > >
> > > > We interpret Power et al.'s grokking (sudden representational reorganization) and Tigges et al.'s findings (different attention heads implementing the same function across training) as empirical evidence of this path-dependency. While they focus on behavioral consistency, we emphasize the underlying mechanistic variability.
> > > >
> > > > We'll clarify that we're interpreting their findings through this theoretical lens rather than claiming these are their explicit conclusions.
> > > >
> > > > **Change 15:** Variables in equations (1) and (2) not defined.
> > > >
> > > > **Response:** Multiple reviewers have noted this issue. We will define all variables properly.
> > > >
> > > > For Equation 1: L(θ) is the loss function, θ represents model parameters, w_t is the token at position t, w_{<t} represents all previous tokens, and p_θ is the model's probability distribution.
> > > >
> > > > For Equation 2: This shows how token embeddings are optimized. The key insight is that e*(w) must minimize prediction error across ALL contexts C(w) where token w appears. This forces the embedding to become a weighted average across diverse uses—mathematical, historical, versioning contexts for '9.11'—creating the contextual averaging problem we identify.
> > > >
> > > > We'll add these definitions and clarify how Equation 2 specifically demonstrates the contextual averaging mechanism.

---

> > > > > ### Author Response · Authors · 2025-08-20
> > > > > **Part 5**
> > > > >
> > > > > **Change 16:** "As Equation 2 predicts, contextually-averaged embeddings resist the domain binding required for stable symbolic circuits" - needs elaboration.
> > > > >
> > > > > **Response:** Equation 2 shows that embeddings must optimize across ALL contexts where a token appears. For '9.11', this includes mathematical contexts (9.11 > 9.9), historical contexts (September 11), and versioning contexts (version 9.11). The embedding becomes a weighted average of these incompatible uses.
> > > > >
> > > > > This averaging destroys the isometric properties needed for symbolic computation. Just as a calculator's circuitry expects consistent numerical representations, mathematical operations require that numerical order and relationships be preserved in embedding space. For LLMs to form stable circuits for any algorithm, inputs must preserve the properties that algorithm expects, even when cast in high dimensions (consider sorting—it fails if vector representations have random perturbations).
> > > > >
> > > > > Figure 1 confirms this breakdown empirically: '10' is closer to '1' than to '9', violating numerical ordering, and digit-word pairs ('5' and 'five') have large distances despite semantic equivalence. Without these isometric properties, the model cannot form stable, input-agnostic circuits for mathematical operations.
> > > > >
> > > > > **Change 17:** Why do we need exact results with LLMs for real-world applications?
> > > > >
> > > > > **Response:** The need for exact results is not academic—it's a practical requirement in real applications.
> > > > >
> > > > > LLMs are routinely asked to compute sums, find maxima, sort data, calculate percentages, and perform financial calculations (e.g., from serialized tables). Without exact computation, these operations fail in deployment. This is precisely why tool use has become essential in production systems—models must delegate to calculators, databases, and code interpreters for reliable results.
> > > > >
> > > > > From our observations, early LLM deployments without integrated tools frequently produced calculation errors that undermined user trust. The industry's rapid shift to tool-augmented systems (GPT with code interpreter, Claude with calculator) demonstrates that exact computation is necessary, not optional.
> > > > >
> > > > > Our analysis explains WHY architectural limitations force this tool dependency rather than allowing native exact computation.
> > > > >
> > > > > **Change 18:** ReLU MLPs obviously only approximate continuous functions - proof seems unnecessary.
> > > > >
> > > > > **Response:** While piecewise linearity of ReLU networks is well-established in the ML literature, we included the proof to make explicit WHY this prevents exact symbolic computation through weight optimization alone—even with infinite training, weights cannot configure themselves to compute exact multiplication.
> > > > >
> > > > > This sets up our main contribution in this section: explaining what happens instead. Since exact computation is impossible, networks resort to hierarchical pattern storage (as shown in Figure 2), which explains both their impressive performance on familiar patterns and brittle failure on novel ones.
> > > > >
> > > > > We placed the proof in the appendix to avoid disrupting flow while maintaining rigor.

---

> > > > > > ### Author Response · Authors · 2025-08-20
> > > > > > **Part 6**
> > > > > >
> > > > > > **Change 19:** Table 2 results from Li et al. study should be cited.
> > > > > >
> > > > > > **Response:** Indeed, we should have cited the sources of the data. Will fix.
> > > > > >
> > > > > > **Change 20:** "Functional Specialization Architectures" section needs citations.
> > > > > >
> > > > > > **Response:** We'll add appropriate citations for the MoE discussion and neuroscience-inspired architectural approaches. This section was intended as forward-looking speculation, but we should ground it with relevant literature on functional specialization in both ML and neuroscience.
> > > > > >
> > > > > > **Change 21:** Table 3 needs better explanations.
> > > > > >
> > > > > > **Response:** The '30.125' and '30.1210' should be '5-digit' and '10-digit'. We'll also clarify that 'sum errors' means failures in final addition despite correct intermediate steps, and explain the percentage ranges. Will fix the table and caption.
> > > > > >
> > > > > > **Change 22:** Section 6 seems to repeat "comprehension vs competence" as "general vs generalizable".
> > > > > >
> > > > > > **Response:** You're right there's overlap, but we're making a distinct point: LLMs achieve 'general' intelligence through pattern completion across massive training data at multiple levels—from token prediction to paragraph coherence to task execution. This is genuinely useful for most everyday tasks. They lack 'generalizable' intelligence—systematic rule discovery and lifted reasoning needed for scientific discovery.
> > > > > >
> > > > > > The comprehension/competence split describes a failure mode. The general/generalizable distinction acknowledges LLMs' real success at practical tasks while identifying what's still missing for novel algorithmic discovery and scientific advance, as discussed in more detail in Section 6.
> > > > > >
> > > > > > **Change 23:** Appendix B.4 content mentioned in main paper - seems redundant.
> > > > > >
> > > > > > **Response:** We included this in the appendix for completeness, so readers who read Appendix B as a standalone unit have the full context. We understand it's redundant with the main text and will consider removing it or adding a note that it recaps material from the main paper.
> > > > > >
> > > > > > **Change 24:** Author names repeated before and within citations.
> > > > > >
> > > > > > **Response:** Other reviewers have also noted this issue caused by inconsistent use of \citet and \citep. We'll fix all instances where author names appear both before and within citations to avoid redundancy.
> > > > > >
> > > > > > **Change 25:** Is "execution limitation" the "comprehension-execution split"?
> > > > > >
> > > > > > **Response:** 'Execution limitation' in that context refers to the performance failures others have documented. The 'comprehension-execution split' (our computational split-brain syndrome) is our contribution—identifying that models can explain algorithms perfectly while failing to execute them. We'll clarify this distinction to avoid confusion.
> > > > > >
> > > > > > **Change 26:** What does "through weight configuration alone" mean?
> > > > > >
> > > > > > **Response:** 'Through weight configuration alone' means no setting of weights can make FFNs compute certain operations exactly.
> > > > > >
> > > > > > Consider the contrast: for addition, there exists a weight configuration that computes x + y exactly (it's linear). For multiplication, no weight configuration can compute x × y exactly because ReLU networks can only produce piecewise linear functions, while multiplication is inherently non-linear.
> > > > > >
> > > > > > For some algorithms, it's theoretically possible to first convert inputs to log space, making computation precise. But that would require specially designed circuitry integrated into the transformer architecture. The model cannot rewire its 'brain' just by learning about the 'how' through text (maybe it can, in the future, by training another model).
> > > > > >
> > > > > > This is an architectural constraint—even with perfect training and infinite data, the weights cannot configure themselves to implement exact multiplication. We'll add this clarifying example to the text.

---

> > > > > > > ### Author Response · Authors · 2025-08-20
> > > > > > > **Part 7**
> > > > > > >
> > > > > > > **Broader Impact Concerns**
> > > > > > >
> > > > > > > **Response:** Our work has several broader implications:
> > > > > > >
> > > > > > > **Positive impacts:** By identifying fundamental limitations, we help prevent over-reliance on LLMs in critical applications (medical, legal, financial) where the comprehension-execution split could cause harm. Understanding these limitations guides safer deployment with appropriate safeguards (tool integration, verification).
> > > > > > >
> > > > > > > **Interpretability concerns:** Our finding that instruction and execution pathways are geometrically separated challenges current explainability efforts—models' self-explanations may not reflect their actual computational processes, potentially misleading users and researchers. Moreover, since pattern synthesis is path-dependent, finding traceable, input-agnostic circuitry appears fundamentally difficult.
> > > > > > >
> > > > > > > **Future directions:** Our analysis suggests focusing resources on architectural innovations rather than scaling alone, potentially reducing computational/environmental costs while improving reliability.
> > > > > > >
> > > > > > > We'll add a brief broader impacts discussion addressing these points.
> > > > > > >
> > > > > > > **Additional Comments**
> > > > > > >
> > > > > > > **Response:** We appreciate your thorough review and understand your concerns about the survey-like aspects and organization.
> > > > > > >
> > > > > > > The extensive literature coverage is necessitated by the nature of our contribution: the computational split-brain syndrome isn't a single isolated flaw but emerges from multiple interconnected architectural constraints and the current learning paradigm, affecting both arithmetic and relational reasoning. Documenting this comprehensively requires engaging with diverse literatures—from complexity theory to mechanistic interpretability to cognitive science.
> > > > > > >
> > > > > > > Our novel contributions include: (1) identifying and naming the split-brain phenomenon, (2) providing a unified three-claim framework explaining WHY it occurs, (3) demonstrating this failure pattern across both arithmetic and relational reasoning domains, (4) showing geometric separation between instruction and execution pathways, and (5) demonstrating these limitations persist across compensatory strategies. The empirical work (Figures 1-4, Table 3) directly supports our theoretical analysis.
> > > > > > >
> > > > > > > We believe TMLR is well-suited for this kind of comprehensive analysis that bridges multiple subfields to identify fundamental architectural limitations. We hope our responses address your concerns and demonstrate the work's contributions beyond surveying. We're happy to make additional revisions to strengthen the paper's clarity and impact.
> > > > > > >
> > > > > > > Thank you for your detailed feedback—it will help us improve the manuscript significantly.

---

> > > > > > > > ### Comment · Reviewer_yB2q · 2025-09-01
> > > > > > > > **part 7 response, and final comments**
> > > > > > > >
> > > > > > > > ## Impact Concerns
> > > > > > > >
> > > > > > > > This is fantastic, thanks a lot! If I could make one more suggestion: there is a rather developed body of work suggesting that the societal benefits of largescale, general-purpose models—like LLMs—are outweighed by the negatives.
> > > > > > > >
> > > > > > > > Two recent papers on the topic:
> > > > > > > > - *Hype, Sustainability, and the Price of the Bigger-is-Better Paradigm in AI* Varoquaux et al. (2025)
> > > > > > > > - *On the Dangers of Stochastic Parrots: Can Language Models Be Too Big?* Bender et al. (2021)
> > > > > > > >
> > > > > > > > Two relevant books:
> > > > > > > > - *The Atlas of AI*, Crawford (2021)
> > > > > > > > - *The AI Con*, Bender & Hanna (2025)
> > > > > > > >
> > > > > > > > Given that the current paper is aimed at improving LLMs, it could be worthwhile to engage with this literature somehow in the broader impact concerns section.
> > > > > > > >
> > > > > > > > ## Additional Comments and final comments
> > > > > > > >
> > > > > > > > First, thanks for the back and forth—it’s been a pleasure interacting with you. You have nicely addressed many of my questions.
> > > > > > > >
> > > > > > > > In terms of the main weaknesses I mentioned in the original review, I believe the issues of language choice and of references, while not yet fully addressed, could be straightforwardly solved by following up on my “change 1 response” and “change 14 response” above. If the changes are made regarding my concerns about paper organization this should also help with the organization-related weakness.
> > > > > > > >
> > > > > > > > I’m still a little concerned about support for the three main claims. I have focused on claim 2–see comments 17 and 4–but have similar concerns about the other claims. My concern around the claims was part of the reason I suggested maybe presenting the paper as a position piece or review. The ideas included are interesting and, in my view, worthy of reflection. However, as mentioned in my second follow-up to change 4 (in the context of claim 2), they seem a little too intuitive and not quite quantitative enough for a research paper. Though this is just my opinion!
> > > > > > > >
> > > > > > > > Finally, note that while I didn’t respond to all of your responses to my questions, I read and appreciated all of them. I only responded to ones that I deemed particularly critical. Best of luck with the manuscript!

---

> > > > > > > > > ### Author Response · Authors · 2025-09-01
> > > > > > > > > **Re: final comments etc.**
> > > > > > > > >
> > > > > > > > > Dear Reviewer yB2q,
> > > > > > > > >
> > > > > > > > > Thank you for this thoughtful engagement with our work. Our back-and-forth has been invaluable in helping us clarify our arguments.
> > > > > > > > >
> > > > > > > > > On Terminology (Change 1): You make a compelling case. Since our paper is already on arXiv under the current title and beginning to be cited, we'll keep "Comprehension Without Competence" but immediately clarify it as "explanation without execution" in the introduction. This addresses your clarity concerns while avoiding confusion from title changes to an already-public work. We'll ensure readers understand we're describing systems that can explain procedures they cannot execute—an inversion of Dennett's framework where comprehension emerges from competence.
> > > > > > > > >
> > > > > > > > > On Pattern Completion vs Algorithmic Approximation (Change 4):
> > > > > > > > > Grade school students in parts of the world (e.g., China) memorize multiplication tables. If they tried extending this to triple digits through pure memorization without learning the algorithm, they'd fail—true algorithmic implementation must be applied here.
> > > > > > > > >
> > > > > > > > > We'll clarify: an algorithmic approximator implements procedural steps (even imperfectly). The key test is novel format generalization. An algorithmic approximator handles '47 × 83', 'forty-seven times eighty-three', and '4.7 × 10¹ × 8.3 × 10¹' using the same procedure. A pattern completer needs separate memorization for each format.
> > > > > > > > >
> > > > > > > > > Your hierarchical binary multiplication example would show both gradual improvement AND generalization to novel bit patterns. What we observe instead is gradual improvement (Figure 2) but catastrophic failure on novel formats (Mirzadeh's 65% drop). Many researchers, including ourselves initially, searched for generalizable arithmetic circuitry in LLMs. Our finding: at inference time, such circuitry doesn't exist. Models perform pattern completion whether explaining algorithms or attempting execution—they just complete different memorized patterns.
> > > > > > > > >
> > > > > > > > > On Floating Point and Approximation (Change 17): You're absolutely right—approximation itself isn't the problem, as floating point demonstrates. The real issue is that LLMs cannot internally invoke their learned algorithmic knowledge to guide execution (related to Claim3).
> > > > > > > > >
> > > > > > > > > If models could call their perfect explanations ("multiply by ones place, then tens...") to guide computation internally, they'd be algorithmic approximators like floating point systems. But our geometric separation evidence (Figures 3-4) shows instruction and execution occupy separate representational spaces that cannot coordinate. Floating point internally executes procedures with controlled approximation; LLMs have disconnected pathways for explaining versus executing. At present, self-scaffolding (using the well memorized description explicitly guide execution) is one workaround among many, but with its own challenges (Sec5).
> > > > > > > > >
> > > > > > > > > On Broader Impacts: Thank you for directing us to Bender et al. and Varoquaux et al. Our findings actually support their critiques—by showing that limitations are architectural rather than scale-dependent, we provide technical evidence that the 'bigger-is-better' paradigm cannot address fundamental constraints. We'll incorporate this alignment. We note in the last paragraph of "Contributions": "....This suggests computational split-brain syndrome will persist across model families and scales unless addressed through fundamental architectural innovations rather than incremental improvements."
> > > > > > > > >
> > > > > > > > > Again, we totally appreciate your engagement. It allows us the opportunity to  clarify our own thoughts. Barring new comments from other reviewers, we will proceed to make a new draft.
> > > > > > > > >
> > > > > > > > > Best regards, The Authors

---

> > > > > > ### Comment · Reviewer_yB2q · 2025-08-30
> > > > > > **part 5 reviewer response**
> > > > > >
> > > > > > **Change 17 response:** but LLMs can give exact results on a finite set of points and, in practice, one will never need to stray outside a finite set of points in real world applications. For example, all modern computing infrastructure also doesn’t give exact results on an infinite set of points as it uses floating point numbers with finite precision

---

> > > > > > > ### Author Response · Authors · 2025-08-31
> > > > > > > **Re: Change 17**
> > > > > > >
> > > > > > > Response to Reviewer on Change 17:
> > > > > > >
> > > > > > > Your floating point analogy is instructive but highlights a crucial difference. Floating point arithmetic has well-defined precision bounds and predictable behavior - you know exactly when and how it will round. LLMs' arithmetic failures are unpredictable and catastrophic: they might correctly compute 342 × 567 but fail on 341 × 568.
> > > > > > >
> > > > > > > In practice, this unpredictability is precisely the problem. Our multi-digit multiplication experiments (Table 3) show complete failure at 10 digits despite perfect algorithmic explanation. Real deployments encounter this constantly - financial calculations, data analysis, even simple counting tasks produce unreliable results, which is why production systems now require calculator integration.
> > > > > > >
> > > > > > > The "finite set of points" argument assumes we can enumerate all needed calculations in advance. But real applications generate novel calculations continuously - new prices, quantities, dates. When a model correctly computes 90% of multiplications but fails unpredictably on 10%, users cannot trust any result without external verification.
> > > > > > >
> > > > > > > This isn't about infinite mathematical precision but about reliable execution within practical bounds - something floating point arithmetic achieves but pattern-based approximation does not.

---

> > > > > > > > ### Comment · Reviewer_yB2q · 2025-09-01
> > > > > > > > **change 17, second follow-up**
> > > > > > > >
> > > > > > > > The reason I brought up floating point arithmetic is because it is also technically an approximation to true arithmetic. If it can perform effectively I don’t believe that the approximate character of LLMs is an appropriately satisfying explanation for their failure to achieve the reliability that you rightfully point out as being a serious problem

---

> > > > > ### Comment · Reviewer_yB2q · 2025-08-30
> > > > > **part 4 reviewer response**
> > > > >
> > > > > **Change 12 response:** great, thanks!
> > > > >
> > > > > **Change 14 response:** thanks for the update! My primary concern here is that the “mechanistic variability” might not reflect changes in underlying “universal computational principles”. It seems to me that this is a fairly strong claim that would require some other experiments to test.
> > > > >
> > > > > **Change 15 response:** great, thank you. The purpose of Equation 2, with all variables properly defined makes sense, but what is the purpose of Equation 1?

---

> > > > > > ### Author Response · Authors · 2025-08-31
> > > > > > **Re Change 14 and 15:**
> > > > > >
> > > > > > Response to Reviewer on Change 14:
> > > > > >
> > > > > > You're absolutely right that testing "mechanistic variability" rigorously would require extensive experiments with different training permutations, schedules, and corpora - experiments that are prohibitively expensive at LLM scale.
> > > > > >
> > > > > > To clarify: we're not claiming to have discovered "universal computational principles." Rather, we're making the logical inference that if arithmetic is just one task among thousands during training, and training dynamics are path-dependent (as Power et al.'s grokking shows), then the specific implementation of arithmetic patterns will vary based on training history. Different schedules would lead to residual fitting occurring in different parts of the model.
> > > > > >
> > > > > > We agree this remains a conjecture without controlled experiments. We'll revise to make clearer that we're interpreting existing evidence through this theoretical lens rather than claiming empirical proof of mechanistic variability.
> > > > > >
> > > > > > Response to Reviewer on Change 15:
> > > > > >
> > > > > > Equation 1 is the omnipresent next-token prediction loss that sets the context for our analysis. The θ in Equation 1 represents the aggregate model parameters, of which token embeddings are a subset to be learned (as shown in Equation 2).
> > > > > >
> > > > > > The connection is: Equation 1 shows what the entire model optimizes for (next-token prediction), while Equation 2 demonstrates how this global objective forces each token embedding to become a compromise across all its contexts. This is the mechanism that creates contextual averaging—the embedding for '9.11' must work for mathematical, historical, and versioning contexts because they all contribute to the loss in Equation 1.

---

> > > > ### Comment · Reviewer_yB2q · 2025-08-30
> > > > **part 3 reviewer response**
> > > >
> > > > **Change 6 response:** thanks for the clarification!
> > > >
> > > > **Change 8 response:** perfect, thanks

---

> > ### Comment · Reviewer_yB2q · 2025-08-30
> > **part 1 reviewer response**
> >
> > **Change 1 response:** to clarify, I don’t think that the phrases “split brain syndrome” (applied to LLMs), and “comprehension without competence” are responsible terms to use to describe LLMs. That is, I would advocate strongly for not simply adding extra motivation for them, but replacing them with something that favours accuracy of description over flashiness. The reason for this is twofold: first, as mentioned above, there are many works highlighting the deep problems with anthropomorphising LLMs (see e.g. Anthropomorphism in AI: hype and fallacy, Placani (2024), AI and Ethics; ChatGPT is bullshit, Hicks et al. (2024) Ethics and Information Technology). Second, if one has to add extra sentences explaining, or adding disclaimers, for the use of certain terms then I think that this in and of itself is worth finding better terms. For example, taking inspiration from your above response, one could use “explanation without execution” in place of “comprehension without competence”. To me, this better describes the observed phenomenon and doesn’t require an extra explanation. I believe one could then simply remove the “split brain syndrome” description in favour of the “explanation without execution problem”, as it doesn’t add anything beyond “explanation without execution” except the potential to confuse and needlessly anthropomorphise.
> >
> > **Change 2 response:** thanks–this description resolves my confusion here!

---

> > > ### Author Response · Authors · 2025-08-31
> > > **Response to Reviewer on Change 1**
> > >
> > > Our discussion here perhaps reflects interesting philosophical differences. We took inspiration from Dennett's framework, which is actually anti-anthropomorphic in spirit—it describes what systems DO behaviorally, not what they THINK or FEEL. When we say an LLM "comprehends" multiplication procedures, we mean it exhibits explanation behavior that observers would label comprehension in any system, nothing more.
> > >
> > > Indeed, almost any description of what LLMs do uses human-activity metaphors (learning, understanding, reasoning, attention, etc.). The field has accepted this vocabulary while understanding these describe computational processes, not human cognition. Complete avoidance seems both impractical and potentially misleading—these systems do perform functions analogous to human cognitive tasks, even if through different mechanisms.
> > >
> > > "Split-brain" in both neuroscience and distributed systems refers to parallel independent capabilities that cannot coordinate—a purely functional description. We find this precisely captures the phenomenon without requiring consciousness attribution.
> > >
> > > That said, we appreciate "explanation without execution" as complementary framing and will include it to clarify our meaning. We'll also ensure our behavioral stance is explicit from the outset.

---

> > > > ### Comment · Reviewer_yB2q · 2025-09-01
> > > > **change 1, second follow-up**
> > > >
> > > > I accept the idea that one can use biological and cognitive analogies where they are the best, most precise, options available. But in this case I fail to see how they provide more insight—without misleading—compared to alternative options like “explanation without execution”

---

> ### Comment · Reviewer_yB2q · 2025-08-30
> **part 2 reviewer response**
>
> **Change 3 response:** this seems reasonable in broad strokes, thanks for the clarification. In general, I still wonder if in certain cases attention layers might perform fairly significant nonlinear computation. For example, a recent arXiv preprint suggests that this might be the case as the number of transformer heads are increased: Universal Approximation with Softmax Attention Hu et al.
>
> **Change 4 response:** I think most folks who study deep learning recognize that models iteratively construct solutions over several layers–why is this extra, seemingly well recognized, phenomenon needed to better understand the authors’ hypothesis that approximate computation results in poor algorithmic performance? To look at this question from a different perspective, how should Figure 2 have looked to provide evidence against the authors’ hypothesis? If there was a big jump in performance at a single layer, but the learned functions were still approximate (as they necessarily will have to be) wouldn’t the author’s argument about approximate computation still be a problem?
> On the other hand, I think it would be fantastic to provide more evidence in this section as to why approximate computation is a serious hurdle for algorithmic computation, as I’m not sure this hypothesis itself is well tested by the existing results in the paper.
>
> **Change 5 response:** awesome, thanks!

---

> > ### Author Response · Authors · 2025-08-31
> > **Re: Change 3 and 4**
> >
> > Response to Reviewer on Change 3:
> >
> > Totally agree that attention performs significant transformations and is critical for Transformer success. In our view, multiple heads enable a form of factorized learning (akin to multiple channels in CNNs), where each head acts as an information bottleneck to capture different data signatures, but this is beyond the scope of this paper.
> >
> > In the context of this paper, the key constraint is causal masking: when the model reaches the '=' token in '23 × 17 = ?', attention can only see backwards to '23 × 17 =' and must produce '391'. No matter how sophisticated the attention mechanism, it cannot generate this novel value from the operands and operator alone—it can only create weighted combinations of existing representations. The more straightforward explanation we offer is that attention computes an "index" into FFN's vast repository to retrieve a "value" fragment as the residual fitting business moves from layer to layer.
> >
> > We'll review the Hu et al. paper, though as you note, incorporating every new preprint during review risks endless revision.
> >
> > Response to Reviewer on Change 4:
> > You raise an important question about what Figure 2 demonstrates. The key insight isn't that iterative computation is surprising, but rather what it reveals about HOW transformers solve arithmetic when they cannot execute algorithms directly.
> >
> > Your counterfactual is instructive: if we saw a big jump at a single layer, that would suggest the model found a way to approximate the algorithm itself. Instead, the gradual convergence shows the model assembles answers through hierarchical pattern retrieval across layers—each FFN contributes a partial pattern that collectively approximates the answer.
> >
> > This matters because it explains both successes and failures: models achieve high accuracy on familiar patterns (which can be assembled from stored fragments) but fail on novel ones (which would require actual algorithmic execution). The large FFN capacity enables storing these pattern fragments, while the residual stream allows their gradual assembly.
> >
> > You're right that approximate computation being problematic for algorithms deserves more evidence. Our point is narrower: we're showing that models don't even attempt algorithmic approximation—they use an entirely different mechanism (pattern assembly). This explains why they fail so catastrophically on seemingly simple variations.

---

> > > ### Comment · Reviewer_yB2q · 2025-09-01
> > > **change 4, second follow-up**
> > >
> > > Thanks for the clarification. Indeed, what you are describing feels right intuitively. I guess, given that you are making use of data and figures I was hoping for something a little more quantitative. Perhaps an answer to the following two points might remedy this concern:
> > > 1. A better description of what qualifies as “pattern completion” versus “algorithmic approximation”. When is something a pattern complete-r versus an algorithmic approximator? Can something be both? What would you hope to see in experiments to show that an LLM is one versus the other?
> > > 2. Are we certain that this more incremental profile cannot be consistent with algorithmic approximation? For example: one could imagine a hierarchical processing machine that does vertical binary multiplication, with each layer multiplying a given bit (or several bits) of the first number by all bits of the second and then adding its result to the input from the layer below. In this way, I wouldn’t be surprised if the representation at each layer increased incrementally with depth. This is fairly contrived and an LLM is surely not doing this, but are we certain it might not be doing something like this?

---

> ### Comment · Reviewer_yB2q · 2025-08-30
> **part 6 reviewer response**
>
> **Change 19 response:** perfect, thank you

---

### Author Response · Authors · 2025-08-20
**Response to Reviewers**

Dear Reviewers and Action Editor,

Thank you all for your thorough and constructive reviews of our manuscript. We are grateful for the time and effort you have invested in evaluating our work. We have carefully considered all feedback and prepared detailed responses to each reviewer's comments.

Due to the character limit constraints of the submission system, we note that Reviewer yB2q raised 26 specific change requests along with broader concerns. To address these comprehensively, we have had to break our response to Reviewer yB2q into 7 parts. We hope this formatting approach is acceptable and not a violation of submission guidelines. All other reviewer responses fit within single submissions.

We are prepared to revise the manuscript according to your feedback and await further instructions on the next steps in the review process.

Sincerely,
The Authors

---

### Public Comment · ~Shih-Wai_Lin2 · 2025-09-17
**1. Statement**

Dear Authors,

- I am the first and corresponding author of the position paper *Rules Created by Symbolic Systems Cannot Constrain a Learning System*. I have read your paper carefully and, as a public commentator, I offer the following views in strict accordance with TMLR’s policies.

- **We note that although our projects have different goals, your work (July 10, 2025) overlaps with ours (January 30, 2025) in multiple places—examples, descriptions, claims, and theoretical framing—yet does not cite or discuss our work.**

- Under the rules, double-blind applies only to reviewers. As a named participant, I respect and will abide by your review process and the reviewers’ workflow. To protect the double-blind interaction between authors and reviewers and to ensure a fair assessment of the authors’ contribution, I will not provide any specific links or content that could reveal author identities, but only materials directly relevant to the topics under discussion. The remainder can be explained and supplemented by the authors themselves.

- In tracing the development of your ideas, we found statements on a non-public blog on your personal homepage:

  > 1. This work **relied heavily on GenAI agents**, specifically GPT-4o and Claude 3.5. The author served as producer, bullet-point script writer, and editor, while the AI tools took up the pen and served as brainstorming partners.
  > 2. This work benefited greatly from colleagues both within and outside the author's organization, as well as many students who patiently reviewed machine-generated text, sometimes more than once—about the very machines themselves.

- I respect this openness and honesty. I do **not** wish to diminish the value of your work or turn such rare academic virtues into a burden; rather, I hope you can provide relevant clarifications.

  1. **Question 1:** Did AI assist with the theoretical parts of your work? This could help explain multiple coincidences of overlap with our content and your shift toward theory in 2024–2025 (I hope I’ve characterized this accurately).
  2. **Question 2:** Do these ideas involve contributions from your students and colleagues? (This could relate to our experiences and possible chains of transmission of ideas.)

- For a theorist, independence of thought, human authorship, and proper referencing, citation, and disclosure are crucial.

  - If LLMs were used in shaping core ideas or the theoretical framework, this should be stated clearly so that your contribution is assessed fairly and the work of others is properly respected.
  - LLMs may inadvertently appropriate others’ ideas. This issue becomes more salient as LLM capabilities increase.

- Please note that although your preprint appeared on July 14, 2025—later than our January 30, 2025 paper:

  - After seeing publicity about a similar manuscript, we examined the non-public blog on your homepage. In fact, some of your views predate our January 30, 2025 manuscript.
  - **Therefore, the analysis must distinguish clearly: some of your ideas came earlier than ours. We have listed them in detail to ensure your work and contributions are fairly recognized.**
  - **At the same time, those early notes differ from the current, fully articulated “computational split-brain syndrome” and the present set of claims. The convergence of your finalized paper with our preprint and its subsequent updates—along with our January–April 2025 rebuttals—appears later, especially in the use of the split-brain terminology and the theory it conveys.**
  - **It should also be noted: had we not learned of your preprint, we would not have been able to access the non-public blog via your homepage to understand your thinking. Because those posts are not indexed by search engines, quoting them will not create double-blind issues.**
  - Given that the main ideas in the author’s September 2, 2025 revision were already present in the July 14 arXiv version, we attribute the author’s contribution uniformly to the earliest TMLR submission date—July 10, 2025—without making further distinctions.

- In the same spirit of transparency: we also used LLMs (and have disclosed this in our new submission), but only for translation—and not even for stylistic polishing (since AI tends to alter or distort our original intent). You may consult the latest version of our paper (which includes 140 footnotes and many em-dashes following bracketed restatements). Our theory is self-consistent and complete, and—after multiple investigations—shows originality, independence, and novelty.

- Finally, this is not an accusation but an open academic exchange. In accordance with TMLR policy, there is no need to communicate through private channels. Because detailed comparison is required and space is limited, we will present the detailed comparisons in multiple follow-up replies within this public comment. Such open discussion helps clarify the authors’ actual contributions and facilitates scholarly exchange and comparison.

---

> ### Public Comment · ~Shih-Wai_Lin2 · 2025-09-17
> **2. Background: Why We Are Concerned**
>
> We submit this public comment because of our own experience. Our position paper (January 30, 2025, https://openreview.net/forum?id=ELErARGR5U) advances views very similar to yours and uses the same examples, descriptions, and terminology.
>
> As a position paper, we explicitly challenge several mainstream approaches in AI safety and point out multiple flaws:
>
> - From January 30 to April 2025, we appear to have been suppressed and appropriated by a prominent proponent of a view we critique. Our ideas were compressed into the phrase “symbols are lossy” and then **attributed to** another paper that opposes our stance.
> - In that review, three anomalous reviewers committed multiple violations, and the AC shielded them and wrote a meta-review that contradicted basic facts of the paper and conference rules. After we reported this, the relevant parties did not intervene to stop the violations (https://medium.com/@linshihwai/don-quixote-and-the-windmill-a-case-from-the-systemic-failure-of-the-icml-2025-peer-review-f755f2f6e96b).
> - **Even after our report and public statement, the issue did not end; it continued at another top conference.**
> - **The same proponent seemingly reviewed our paper again, again suppressing our critiques of their work, while simultaneously submitting a manuscript that appeared to incorporate points from our earlier rebuttals into their theoretical framework—submitted to the very same venue.**
>
> Given this prior potential appropriation of ideas, and the procedural breakdowns and ethical concerns we observed in parts of the academic system, our worries intensified—especially upon seeing a paper that appeared more than five months after ours (January 30, 2025) with multiple overlaps with our paper and rebuttals, that also used the rare term “split-brain syndrome,” yet did not cite or discuss our work. Hence this public comment.
>
> To be clear, the proponent we accuse is unrelated to you. However, there were still three anomalous individuals involved, including an AC.
>
> This is also why we ask **Question 2** of you: as stated on your non-public blog, were there unacknowledged contributors involved? This could bear on the coincidence that, amid so many overlaps, both your work and ours use “split-brain syndrome” and advance a nearly identical core theory.
>
> Our complete, latest work is available at https://philpapers.org/rec/SHIPPN, with a detailed update history at https://philpapers.org/versions/LINRCB-2. Our initial January 30, 2025 version was posted at https://papers.ssrn.com/abstract=5121127 and later replaced due to updates. If needed, we can ask SSRN to issue a confirmation. Because the content is the same, we cite the OpenReview version as the representative of our January 30, 2025 work (https://openreview.net/pdf?id=ELErARGR5U).

---

> ### Public Comment · ~Shih-Wai_Lin2 · 2025-09-17
> **3. Overlap 1: The Same Discussion, Similar Views (1)**
>
> **Overlap 1: First, both of our works focus on and discuss the same issue—surface-level fluency does not entail reliable execution—and we each offer an explanation: your *computational split-brain syndrome* and our *Triangle Problem*.**
>
> - **Author (July 10, 2025)**
>
>   > Through controlled experiments and architectural analysis, we demonstrate that LLMs often articulate correct principles without reliably applying them—**a failure rooted not in knowledge access, but in computational execution. We term this phenomenon the computational split-brain syndrome, where instruction and action pathways are geometrically and functionally dissociated.**
>
> - **Ours:**
>
>   1. **January 30, 2025** (https://openreview.net/pdf?id=ELErARGR5U)
>
>      > 1. **To address this, we introduce the Triangle Problem, which formalizes the gap between thinking language and tool language, demonstrating that fluent communication between AI and humans does not imply conceptual equivalence.** Furthermore, we identify symbol adhesion as a critical factor affecting AI interpretability and governance, revealing that AI does not inherently bind symbols to fixed meanings as humans do. These insights provide a new theoretical foundation for AI safety, emphasizing that constraints based solely on symbolic rules are insufficient.（p8)
>      > 2. **However, with the advent of large language models, we, like two entirely different species, can use a common language as an intermediary for communication. This may result in fluent communication at the language level, but the projection and operating mechanisms of the thinking language behind the language in the conceptual space may be entirely different** (Chomsky, 2014a; Norvig, 2017; Bender & Koller, 2020).(p5)
>
>   2. **March 28, 2025 (rebuttal)**
>
>      > We：As emphasized in our response to (8rwE). **alignment at the symbolic level alone is far from sufficient. This leads directly to the Triangle Problem 2, resulting in major behavioral divergence in XY (i.e., behavior patterns).** Such divergences are essentially rooted in human social concepts and empathic perception—outcomes of humans being deeply embedded in the world. A more detailed explanation can be found in the (line 309), particularly under the four Verification Contents, with Verification Content 4 offering a focused discussion. https://openreview.net/forum?id=ELErARGR5U&referrer=%5Bthe%20profile%20of%20Shih-Wai%20Lin%5D(%2Fprofile%3Fid%3D~Shih-Wai_Lin2)#:~:text=As%20emphasized%20in,a%20focused%20discussion.

---

> ### Public Comment · ~Shih-Wai_Lin2 · 2025-09-17
> **3. Overlap 1: The Same Discussion, Similar Views (2)**
>
> **Overlap 2: Both of our works attribute the issue to architectural—or, equivalently, innate-knowledge—differences: AI and humans diverge in their direction of understanding (competence often precedes comprehension; humans develop thinking language before inventing tool language, whereas AI learns tool language first and then develops thinking language).**
>
> - **Author (July 10, 2025)**
>
>   > 1. This represents an inversion of Dennett’s framework where competence typically precedes comprehension in biological systems (Dennett, 2017).
>   > 2. This is not a limitation of scale, training data, or prompting techniques. We argue that computational splitbrain syndrome stems from structural constraints in how LLMs represent symbols, learn procedures, and execute reasoning steps. **This dissociation challenges fundamental assumptions about intelligence derived from human cognition, where explanatory fluency typically correlates with execution competence. LLMs systematically violate this expectation, revealing that artificial and biological intelligence may operate under fundamentally different principles. Unlike symbolic systems that bind tokens to abstract roles and apply rules over those bindings, LLMs operate as pattern completion engines optimized for next-token prediction.**
>   > 3. fluent explanation coupled with unreliable execution—revealing this as a fundamental architectural limitation rather than a domain-specific failure.
>
> - **Ours: January 30, 2025** (https://openreview.net/pdf?id=ELErARGR5U)
>
>   > 1. **Since humans and machines are entirely different, we perceive the world differently. This includes the meaningful dimensions we focus on, the ways of perceiving and expressing these dimensions—for example, humans do not perceive the world at the pixel level—and the evaluation and invocation of these dimensions by innate value knowledge. This leads to different concepts formed by humans and machines, resulting in various forms of thinking language. However, with the advent of large language models, we, like two entirely different species, can use a common language as an intermediary for communication. This may result in fluent communication at the language level, but the projection and operating mechanisms of the thinking language behind the language in the conceptual space may be entirely different (Chomsky, 2014a; Norvig, 2017; Bender & Koller, 2020).（p5)**
>   >
>   >    **Unlike humans, who build language systems from the bottom up, starting with thinking language and then using symbols as containers, AI first learns symbol relationships before acquiring their meanings. It may often become a topdown anthropomorphism (Smolensky, 1988; Deacon, 2011; Marcus, 2018), selecting the optimal solution from multiple possibilities to approximate humans, rather than thinking from a starting point and growing like humans. This is also related to the different roles and conditions of existence of human individuals and AI individuals in the world. To address these issues, we propose the Triangle Problem for discussion.**
>   >
>   >
>   > 2. Since AI does not share the same world and innate knowledge as us, that is, the objects of learning, perception and operation tools, and inherited value knowledge, which is innate evaluation. This may lead to the motherland problem, where a concept (thought symbol) that is incorrectly defined in the conceptual space can work in a limited environment, that is, in the AI’s training environment, but it is not necessarily correct.(p5)
>   >
>   > 3. Verification Content 4: The same ontology, different dimensions, that is, complete inexplicability, that is, we use the same symbols to communicate, but they are actually concepts formed on completely different worlds and innate knowledge, only their shells are the same. Generally speaking, because the world is the same, even if the perception dimensions are different, similar situations to Verification Content 2 will be formed due to the same operation of things. **However, for large language models, their concept positioning may only be the relationship between symbols and not reflect the world, thus constituting inexplicability and the symbol grounding problem, so the logical operations they perform are often different from ours.** (p6)

---

> ### Public Comment · ~Shih-Wai_Lin2 · 2025-09-17
> **3. Overlap 1: The Same Discussion, Similar Views (3)**
>
> **Overlap 3: Both works argue that constraints fail: LLMs cannot reliably perform symbolic manipulation vs. humans cannot constrain a learning system through rules created within a symbolic system.**
>
> - **Author (July 10, 2025)**
>
>   > 1. Importantly, we demonstrate that these constraints manifest consistently across two critical domains: symbolic computation (arithmetic operations) and relational reasoning (logical inference and variable binding). In both domains, models exhibit the same split-brain syndrome—fluent explanation coupled with unreliable execution—revealing this as a fundamental architectural limitation rather than a domain-specific failure.
>   > 2. If our architectural analysis is correct, the computational split-brain syndrome should not be limited to arithmetic. The same three constraints—contextual averaging, architectural impossibility of exact computation, and instruction-execution disconnect—should create identical failure patterns wherever systematic symbolic manipulation is required.
>
> - **Ours: January 30, 2025** (https://openreview.net/pdf?id=ELErARGR5U)
>
>   > To analyze this issue, we propose a novel theoretical framework that thoroughly examines the characteristics and limitations of natural language systems. Specifically, we highlight that natural language inherently lacks meaning. Meaning is assigned meaning through training, confirmed by context, and interpreted by society. We further explore the formation of concepts and language, introducing two Triangle Problems to illustrate the relationship between thinking language and tool language. This demonstrates that fluent communication between AI and humans in natural language does not necessarily mean their thinking languages are identical.
>   >
>   > From this analysis, we conclude that the natural language system is flawed, adapted only to human cognitive abilities and ways of perceiving the world. Humans cannot constrain AI through rules, laws, or procedures created within a symbolic system.
>   >
>   > **This study reveals previously unexplored gaps in research**, specifically the concept of symbol adhesion, the inherent flaws and vulnerabilities of natural language systems, and the interpretative authority of symbols. This conclusion has profound implications for AI governance, demonstrating that relying solely on predefined rules and formalized symbolic methods may be insufficient. Instead, addressing AI safety requires a deeper understanding of the interactions between symbols, context, and cognition. This paper lays the theoretical foundation for Symbolic Safety Science, emphasizing the need to establish new frameworks to further advance research in AI safety and alignment.(p1-p2)

---

> ### Public Comment · ~Shih-Wai_Lin2 · 2025-09-17
> **3. Overlap 1: The Same Discussion, Similar Views (4)**
>
> **Overlap 4: Similar Contributions and Positioning**
>
> - Our contributions have been stated above and will not be repeated here. We use the separation between tool language and thinking language discussed in the Triangle Problem to construct a unified explanatory framework for (generativity, jailbreaks, hallucinations, AI for science, interpretability issues, etc.). See the main text and Appendices J and L of the January 30, 2025 version, or our updates from May–July. We omit lengthy excerpts here.
>
> - **Author (July 10, 2025):**
>
>   > 1. Our findings delineate the boundary of current LLM capabilities and motivate future models with metacognitive control, principle lifting, and structurally grounded execution.
>   > 2. Contributions. Our analysis establishes computational split-brain syndrome as a unifying framework for understanding when and why LLMs fail at systematic reasoning.
>   > 3. The apparent intelligence of LLMs belies a deeper fragility. Despite their success in pattern-rich tasks, these models consistently fail to generalize principles, execute symbolic computations, or reason reliably—even under idealized conditions. Our analysis reveals that these failures are not incidental but structural: LLMs dissociate instruction interpretation from execution, leading to a computational split-brain syndrome that prevents the formation of robust, composable operations.
>   > 4. This diagnosis clarifies why interpretability tools often uncover expressive artifacts rather than reliable algorithms, and why model self-explanations may diverge from the actual pathways used in computation. Our finding that instruction and execution pathways are geometrically separated raises fundamental concerns: models’ explanations of their reasoning may not reflect their actual computational processes, challenging current approaches to explainability. Training order influences which internal patterns are embedded and where—even when outward behavior remains stable—causing interpretability efforts to surface pathdependent artifacts rather than consistent mechanisms.  These insights reframe current conversations about emergence and scaling. Our findings align with critiques by Bender et al. (2021) and others who argue that the "bigger-is-better" paradigm has fundamental flaws. By demonstrating that LLM limitations are architectural rather than scale-dependent, we provide technical evidence that more data or parameters cannot resolve bottlenecks rooted in architectural design. This suggests redirecting resources from pure scaling toward architectural innovations could reduce computational and environmental costs while improving reliability.

---

> ### Public Comment · ~Shih-Wai_Lin2 · 2025-09-17
> **4. Overlap 2: The Same Problem, the Same Conclusion — “Contextual Averaging” vs. “Correct Context” (1)**
>
> - In our detailed discussions of the above, both works use the same “versioning vs. numbers” examples (1.11 > 1.9 and 9.11 > 9.9) and reach the same conclusion: the problem is that AI lacks human-like abilities to establish and select context, which are needed to interpret symbols, carry out symbolic manipulation, and apply the appropriate judgment tools for reasoning.
>
> - In Section 2 of our position paper, we explain in detail that:
>
>   - Symbols are inherently meaningless; their meaning is assigned through training, **confirmed by context**, and interpreted by society.
>   - The determination of meaning happens via context. However, context is hard to define; it is not clearly reflected in human textual knowledge; it is selected by humans’ special perceptual capacities or co-constructed with the environment—and differs each time. We explicitly introduce the notion of **Correct Context** (Sections 2.1, 2.3, 2.4, 2.5).
>
> - Selected excerpts from our work:
>
>   1. **April 9, 2025**: A response to reviewer SB8V, who presented multiple strawman characterizations of our paper—precisely in the areas that overlap with the author’s discussion.
>
>      > We：R1：The full context is expressed in the main text as follows:
>      >
>      > “ the expression “1.11 > 1.9” can be interpreted in two ways without context. In a mathematical context，1.11 is greater than 1.9. In a versioning context，1.11 is also greater than 1.9. However，even without specific context，we naturally understand that the correct interpretation here is the versioning context.”
>      >
>      > ...
>      >
>      > **This sentence is used to illustrate that the issue with current LLMs failing to perform accurate mathematical reasoning is not simply due to a lack of concepts，but rather due to instability in the symbolic system caused by incorrect context definition. The deeper issue lies in the relations between concepts (i.e.，conceptual stickiness)，as well as in how context is defined，selected，or reconstructed.**
>      >
>      > https://openreview.net/forum?id=ELErARGR5U&noteId=ZKhSSJtFGm&referrer=%5BAuthor%20Console%5D(%2Fgroup%3Fid%3DICML.cc%2F2025%2FPosition_Paper_Track%2FAuthors%23your-submissions)#:~:text=This%20sentence%20is,selected%EF%BC%8Cor%20reconstructed.
>
>   2. **March 14, 2025**: Our exchange with another reviewer (AXMX)
>
>      > 1. We：
>      >
>      >    1. **Due to the flexibility and indefinability of context in interactions with the world, AI may establish a non-human context, resulting in constraints failing and agency issues (essentially), because AI lacks the innate social nature of humans and constructs concepts differently from humans (Section 3).**
>      >    2. There is a misalignment in abilities (Sec 3;4;B;F;J;K)—the concept formation occurs in the perception of thinking symbols and thinking language. Natural language is the result of human-style perception and thinking, and fluent communication on the symbolic level is not equivalent to fluent communication on the imaginative level. This is also discussed in the Triangle Problem. The first Triangle Problem reflects the definition of concepts (symbolic stickiness), and the second Triangle problem reflects the use of concepts (generation and behavior under conceptual stickiness).... https://openreview.net/forum?id=ELErARGR5U&noteId=ZKhSSJtFGm&referrer=%5BAuthor%20Console%5D(%2Fgroup%3Fid%3DICML.cc%2F2025%2FPosition_Paper_Track%2FAuthors%23your-submissions)#:~:text=There%20is%20a%20misalignment,behavior%20under%20conceptual%20stickiness).

---

> ### Public Comment · ~Shih-Wai_Lin2 · 2025-09-17
> **4. Overlap 2: The Same Problem, the Same Conclusion — “Contextual Averaging” vs. “Correct Context” (2)**
>
> 3. **January 30, 2025** (https://openreview.net/pdf?id=ELErARGR5U)
>
>      > 1. **However, as described in Section 4, AI does not possess the human-like concept of symbol adhesion, nor does it have the ability to shape and select context as mentioned in Section 2.4.** (p8)
>      >
>      > 2. **The definition and naming of context are often difficult to strictly define and name, with boundaries that are vague and hard to describe precisely (Duranti & Goodwin, 1992).** This is partly due to the limitations of cognitive abilities and partly due to the limitations of expressive tools such as natural language, which prevent us from fully and clearly describing context. Context is often represented as a unique vector address in the conceptual space, thereby specifying the following set (symbol meaning, judgment tools). （p2-p3）
>      >
>      > 3. Context is not a fixed intersection determined at one time. It is often interpreted and generated by an individual’s imaginative space. Although dictionaries provide multiple explanations for words, they are merely symbols and explanations of symbols.（p3）
>      >
>      > 4. This type of knowledge and definition is often not found in human textual descriptions, as it is too obvious or cannot be described by natural language. Individuals often acquire it through social activities.（p3)
>      >
>      >    **The selection and shaping of context are often formed by our innate knowledge and the combination of innate knowledge and environment, which forms acquired knowledge, i.e., concepts. We define innate knowledge as organs and innate value knowledge in Section 3. According to the emotional path formed by value knowledge, a base context is quickly selected, then adjusted and newly created to adapt to the environment, such as updating and adjusting based on external information, and finally shaped according to logic.**
>      >
>      >    In other words, Context is often chosen through a certain feeling, which is described by (Polanyi, 2009) as tacit knowledge. We will use a different definition, value knowledge, to represent this. This concept will later be used to define the concept of innate knowledge and explain the formation of concepts and language, as well as symbol stickiness. For the definition of value knowledge, please refer to Appendix B.
>      >
>      >    **The so-called correct context can be divided into symbol correctness (i.e., proper recognition of symbols), grammatical correctness, semantic correctness, logical correctness, factual correctness, and scenario correctness. These constitute our judgment of rationality, i.e., context connects symbols with their meanings and related judgment tools. This resolves symbol and structural ambiguity, enabling accurate interpretation and analysis, thereby achieving existence brought by existence—the formation and growth of rationality within a scenario.**
>      >
>      >    The above context does not have a clear hierarchical relationship. For example, we can normally interpret a wrong paragraph through context knowledge correction and fitting. This characteristic also often provides rationality for jailbreaks (Yi et al., 2024). That is, the rationality of an object in different scenarios. This avoids detection based on singlescenario behavior and words while the attention mechanism is essentially a way of using context. In fact, various prompt jailbreaks are context jailbreaks (Zeng et al., 2024). They may not be in our context, but they may be correct in the thinking language corresponding to the AI context in the thinking space. For example, using the story of grandma and the wizard to achieve jailbreak, thereby avoiding detection based on behavior and words, including dangerous thinking actions and dangerous concepts.  Due to the often undefined range and definition of context, even if it can be defined, we also discuss other possible attack methods in Appendix L. The correctness of context is also often applied to the effectiveness of open-ended question generation. For details, please refer to Appendix C.
>      >
>      >    Due to the often undefined range and definition of context, even if it can be defined, we also discuss other possible attack methods in Appendix L. The correctness of context is also often applied to the effectiveness of open-ended question generation. For details, please refer to Appendix C.

---

> ### Public Comment · ~Shih-Wai_Lin2 · 2025-09-17
> **4. Overlap 2: The Same Problem, the Same Conclusion — “Contextual Averaging” vs. “Correct Context” (3)**
>
> - **Author (July 10, 2025)**
>
>   > We now face a fundamental design trade-off. By reducing all language processing to a single continuous optimization task (next-token prediction) over unified token embeddings, transformers achieve their remarkable generality and fluency. **However, this same unification prevents the type-based reasoning and domainspecific binding that symbolic computation requires. The contextual averaging that enables rich semantic understanding also contaminates mathematical symbols; the single objective that scales so effectively treats instruction and execution as indistinguishable patterns. Understanding this trade-off clarifies why the computational split-brain syndrome persists across model families and scales: it emerges from the very design choices that make LLMs powerful.**
>
> - The theoretical core expressed by the author overlaps with the core of our **January 30, 2025** theory. The overlap lies in that the author’s “contextual averaging,” etc., corresponds to our explicitly defined notions of context, context selection, and **Correct Context** (January 30, 2025).
>
>   > The so-called correct context can be divided into:
>   >
>   > $$
>   > \text{Correct Context}
>   > \begin{cases}
>   > \text{Symbol correctness}\text{ (i.e., correct recognition of symbols)} \\\\
>   > \text{Grammatical correctness} \\\\
>   > \text{Semantic correctness} \\\\
>   > \text{Logical correctness} \\\\
>   > \text{Factual correctness} \\\\
>   > \text{Scenario correctness}
>   > \end{cases}.
>   > $$
>   >
>   >
>   > The definition of context correctness and its function are also reflected in the effectiveness of AI's open-ended question generation. This involves using the correct elements in its concept recognition and performing the correct processing actions with the correct concepts. Therefore, AI training often aims to find the correct context, forming an effective set of concepts in the thinking language space to achieve correct recognition, operation, and growth. In other words, the attention mechanism in the AI field may also work in this way, with the essence of the attention mechanism being the definition and search for context. (p 13)
>
> - It should be noted that although the author’s preprint (July 14, 2025) appeared later than ours (January 30, 2025), the author’s related discussions (e.g., the versioning example and “contextual averaging”) predate our work and come from the author’s non-public blog. The author also disclosed the involvement of AI tools and other contributors. (I hope this description is accurate—please correct me if not.)
>
>   1. **Author (November 24, 2024)**
>
>      > 1. Large Language Models (LLMs) have revolutionized artificial intelligence, demonstrating remarkable capabilities across diverse tasks. However, their amusing failure cases reveal deeper puzzles: Why do models trained on "A is B" fail to recognize "B is A"? Why do they think 9.9 is smaller than 9.11? And why are we not surprised when LLMs struggle with arithmetic—as if our children might start blaming their math homework errors on "out-of-distribution" problems?
>      > 2. We characterize how LLMs operate on inputs that are pre-initialized with averaged meanings from training, requiring careful re-contextualization during inference. This perspective provides a theoretical basis for understanding context-dependent computation in LLMs, explaining both their flexibility in language understanding and their struggles with precise computation.
>
>   2. **Author (January 13, 2025)**
>
>      > LLMs encode tokens as high-dimensional vectors optimized for aggregated patterns in the training data, leading to context averaging. A token like "9.11" might appear in contexts ranging from date references to software version numbers, with its meaning heavily dependent on surrounding tokens.

---

> ### Public Comment · ~Shih-Wai_Lin2 · 2025-09-17
> **5. Overlap 3: Theoretical Overlap — “computational split-brain syndrome” vs. “Triangle Problem” （1）**
>
> - Building on the shared discussion above, we each proposed our own theory to characterize the separation between **“knowing about”** and **“executing,”** or, put differently, the separation between **thinking language** and **tool language**.
>
>   - **Author:** *computational split-brain syndrome* (July 10, 2025)
>   - **Ours:** *Triangle Problem* (January 30, 2025)
>
> - According to the author’s description (July 10, 2025):
>
>   > Main Claims of This Work. Our analysis establishes this disconnect through three interconnected claims:
>   >
>   > - Claim 1 (Representation): LLM token embeddings encode context-weighted averages that systematically resist automatic domain binding, preventing stable symbolic circuits across computational domains.
>   > - Claim 2 (Computation): Feed-forward networks face architectural impossibility of implementing exact symbolic operations through weight configuration alone, forcing them to resort to residual pattern fitting rather than implementing generalizable symbolic procedures.
>   > - Claim 3 (Instruction-Execution Disconnect): Next-token prediction objectives decouple algorithmic descriptions from executable behavior.
>   >
>   > These three constraints must be understood together to explain the computational split-brain syndrome:
>   >
>   >
>   >
>   > Claims 1 & 2 (Execution failures): When encountering execution instances (e.g., ‘9.11 > 9.9?’), contextual averaging prevents clean mathematical binding (Claim 1). Even if binding were perfect, FFNs cannot implement exact operations—they resort to pattern storage (Claim 2). Together, these establish that reliable execution is impossible.
>   >
>   >
>   >
>   > Claim 3 (The disconnect): Next-token prediction provides no mechanism for well-learned instructions to guide execution. Both capabilities emerge as separate pattern-matching pathways with no automatic binding between them. The model thus can explain perfectly yet fail at execution—the split-brain syndrome.
>   >
>   >
>   >
>   > Our findings suggest that mechanistic interpretability studies identifying “arithmetic circuits” or “adder neurons” may be observing sophisticated forms of pattern matching coordination that are formed pathdependently during learning, rather than genuine computational subroutines. **More critically, the geometric dissociation between instruction and execution pathways raises concerns about both model self-explanations and interpretability research: models may articulate reasoning procedures through neural routes distinct from those used for actual computation, potentially misleading both self-monitoring and researcher analysis of neural activations.**
>
> - There is substantial overlap between the core of the author’s theory and our Triangle Problem (Triangle Problem 1 and Triangle Problem 2): namely, differences in architecture (or innate knowledge) lead to differences in execution (i.e., in the cognitive computation by which thinking language operates over tool language), and thus to divergences in resulting behavior.

---

> ### Public Comment · ~Shih-Wai_Lin2 · 2025-09-17
> **5. Overlap 3: Theoretical Overlap — “computational split-brain syndrome” vs. “Triangle Problem” （2）**
>
> 1. **First, the content of our Triangle Problem:**
>
>    - **Ours (January 30, 2025)** — https://openreview.net/pdf?id=ELErARGR5U
>
>      > Triangle Problem 1 and Triangle Problem 2
>      >
>      > 1. **Due to the current LLM models being able to simulate human communication very well, the core discussion of the Triangle Problem revolves around the definition of concepts and the issue of similarity, that is, the positioning of thought symbols in the conceptual space, which is the position of points, and the similarity of understanding, as well as the relationship between the points formed by a sentence, which is the positioning of thought language. Therefore, it is not simply a matter of symbol grounding. The current state of the Triangle Problem is recognition and understanding, which we classify as Triangle Problem 1. The subsequent state is growth based on understanding, which is the rational growth defined by context, or open generation, which we define as Triangle Problem 2**. (p5)
>      >
>      > 2. Since AI does not share the same world and innate knowledge as us, that is, the objects of learning, perception and operation tools, and inherited value knowledge, which is innate evaluation. This may lead to the motherland problem, where a concept (thought symbol) that is incorrectly defined in the conceptual space can work in a limited environment, that is, in the AI’s training environment, but it is not necessarily correct. The so-called motherland problem is a story I learned in a textbook when I was a child, which tells the story of a sacrificed military dog from the Soviet Union being sent back to its motherland. At that time, a classmate asked why it was sent back to China. Obviously, the concept of motherland was incorrectly defined, but because in our long-term textbooks, the motherland always referred to China, it worked in this environment, but in this unexpected situation, a problem arose. This story still occurs under the condition that we have almost the same innate knowledge. However, due to the huge difference in innate knowledge and the world between AI and humans, this kind of conceptual misdefinition deviation may be inexplicable from a human perspective. This makes AI’s behavior unpredictable to us, making it no longer a tool that we can effectively use, thus constituting a principal-agent problem.
>      >
>      > 3. Therefore, we set up a Triangle Problem to discuss. Humans and AI can communicate fluently on the XY level, that is, creating natural language symbols “patterns” on X to form XY , but this does not mean that humans and AI have achieved human-like communication, that is, the exchange of imaginative space through natural language as a shell. Therefore, in the XY space, we and AI construct acceptable “patterns” formed by the relationships between points that both parties consider reasonable, which is fluent communication, but this does not mean that the conceptual spaces between each other are similar. Specifically, X is the symbol space, Y is the result established by manipulating natural language symbols through thought language in this symbol space, and Z is a super-conceptual space that projects the patterns on the XY space into the conceptual space, which can simultaneously project our conceptual space and the AI’s conceptual space. As shown in Figure 1.
>      >
>      > 4. **Figure 1. Triangle Problem 1: Definition of Symbolic Concepts.} Fluent communication in the $XY$ space does not imply that our thinking languages are identical. $\vec{v_{\mathrm{comm}}}$ and $\vec{v_{\mathrm{interp}}}$ represent the action sequences of communication and cognitive interpretation, respectively. (p6)**

---

> ### Public Comment · ~Shih-Wai_Lin2 · 2025-09-17
> **5. Overlap 3: Theoretical Overlap — “computational split-brain syndrome” vs. “Triangle Problem” （3）**
>
> 2. After addressing the issue of context, our paper first examines Triangle Problem 1: identifying symbols and their relations in the shared interactive symbolic XY space (tool language), and the resulting conceptual vectors (thinking language) obtained by projecting these identifications into Z space.
>
>    - **Ours (January 30, 2025)** — https://openreview.net/pdf?id=ELErARGR5U
>
>      > Triangle Problem 1: Definition of Symbolic Concepts (Positioning)
>      >
>      > As a start, we construct a simple closed-loop example as Figure 1 to illustrate, without discussing its function as a concept, that is, the possible existence brought by existence, i.e., a → b → c → a, thus not discussing the growth problem. **For example, we use natural language to construct “I wake up, work, and sleep every day.” on XY . Considering that AI’s innate knowledge is entirely different from ours, it can’t have the human-perceived concepts of sleeping and waking up, but only to learn the shell of the concepts, that is, words. AI may have the following interpretations: first, approximately reasonable: “I turn on, work, turn off every day.” Second, unreasonable: “low temperature, blue, sweet, useful.” Here, the nouns are correct but unrelated, and they may even be incorrect symbols or unable to form the relationship of a → b → c → a. Therefore, it presents as shown in Figure 1**. (p6)
>
>    - **This aligns with the author’s Claim 1 (Representation): symbols acquire meaning through context, but architectural differences (contextual averaging) lead to non-human context construction, so the actual conceptual vectors may differ from humans’. In short, the model cannot reliably construct and select the correct context.**
>
>    - **We both use similar phrasing: the meaning associated with a symbol (thinking symbol / thinking language) may differ. For example, the author (July 10, 2025):**
>
>      > To demonstrate this contextual contamination, we presented LLaMA2-7B-chat with prompts like “9.11 is a” and “9.9 is a”. The completions reveal dramatically different semantic associations:
>      >
>      > - **For “9.11 is a”, the model produces: “day”, “date”, “remembrance”, “tragedy”**
>      > - **For “9.9 is a”, the model produces: “number”, “decimal”, “perfect”, “high”**
>      >
>      > This divergence shows “9.11” retaining historical event associations while “9.9” behaves as a scalar quantity. Quantitative analysis using negative log-likelihoods across different templates (DATE, VERSION, MEASURE) confirms these contextual biases. The missing operation is domain binding—establishing “9.11” as an element in R governed by mathematical laws rather than historical associations.

---

> ### Public Comment · ~Shih-Wai_Lin2 · 2025-09-17
> **5. Overlap 3: Theoretical Overlap — “computational split-brain syndrome” vs. “Triangle Problem” （4）**
>
> 3. **After completing our discussion of Triangle Problem 1 (i.e., locating the conceptual vector corresponding to an actual symbol via context), we turn to the next step—how thinking language, after cognitive computation, operates on the tool language: Triangle Problem 2, namely “Rational Growth of State in Context.” In other words, it examines the outcomes when different thinking languages, shaped by different architectures, act upon the same tool language.**
>
>    - **Ours (January 30, 2025)** — https://openreview.net/pdf?id=ELErARGR5U
>
>      > Figure 2. Triangle Problem 2: **Rational Growth of State in Context. The next-step response or generation occurs in the XY space after cognitive computation, where different thinking languages operate using the same tool language. This process manifests not only in textual symbols but also in behavioral or gestural symbols.**（p6)
>      >
>      > **Triangle Problem 2: Rational Growth of State in Context Building on the previous issue of positioning, we also need to consider logical operations, that is, the reasonable processing and operation of information in the dimension of concepts, which is the existence brought by the context in XY . The so-called Triangle Problem 2 in Figure 2 refers to the issue of growth similarity for a non-closed logical chain, which is the manifestation of growth in Z on XY . It is used to verify the reasoning ability and similarity based on the existence of existing information. That is, the generative ability or rational growth ability brought by the definition and selection of its context. This also reflects AI’s performance in open generation, whether the generated results are reasonable, and whether it has performed logical operations similar to humans in understanding the state. This often requires AI’s ability to shape and select context to match the human value knowledge system.** This is also the fundamental reason for the new principal-agent problem, that is, the agent’s misunderstanding of the principal’s intentions, forming helpful harm (i.e., damaging the principal’s utility). For additional content brought by the Triangle Problem, please refer to Appendix J. (p6-p7)
>
>    - In the appendix, we elaborate: architecture (innate knowledge) determines an agent’s capacities (cognitive actions, observable dimensions, and the dimensions and values of formed concepts). We highlight foundational differences between humans and AI that lead to differences in thinking language and the **cognitive actions** performed (i.e., the objects, dimensions, and dimensional values that can be operated on). Thus, behind surface-level similarity, entirely different thinking languages may be operating the same tool language.
>
>      - **Ours (January 30, 2025)** — https://openreview.net/pdf?id=ELErARGR5U
>
>        > 1. where knowledge is defined as:
>        >
>        >    ​		$$\text{Knowledge} \begin{cases}\text{Innate Knowledge} \begin{cases}\text{Organs} \\\\ \text{Value Knowledge}\end{cases} \\\\ \text{Acquired Knowledge} \begin{cases}\text{Concepts} \\\\ \text{Acquired Value Knowledge}\end{cases}\end{cases}.$$ （p15）
>        >
>        > 2. Therefore, the world and innate knowledge determine the formation of thinking language, that is, concepts. For a local region, due to the similarity of the world and innate knowledge, individuals within this area form similar concepts and select similar containers as their shells, leading to the formation of language. For more details, please refer to Appendix F.  Innate knowledge refers to abilities we are born with, which are selected and formed through our evolution. We define it as a set of organs, including perceptual organs, which extract information from the world, operational organs, which consist of physical space operational organs and imaginative space operational organs, and innate value knowledge.  These innate organs determine which dimensions are meaningful, thus shaping our perceptual organs’ capabilities and modes of expression. For example, they define the range of visible light and the hearing range. They also construct our perceptual range and distinguishability, referred to as class fineness, and form the projection of objects in the imaginative space as raw materials for concept formation. These projections also function as symbols.  **The operational organs determine the way we interact with the world, including the extent of our actions and the level, quantity, and effect of these actions. The operating organs of the imaginative space determine thinking actions.**(p4)

---

> ### Public Comment · ~Shih-Wai_Lin2 · 2025-09-17
> **5. Overlap 3: Theoretical Overlap — “computational split-brain syndrome” vs. “Triangle Problem” （5）**
>
> > 3. For individuals in a two-dimensional world, the projection of a three-dimensional pinball motion onto their two-dimensional space appears random and inexplicable. This highlights that, even with identical perceptual dimensions and analytical methods, significant differences in intelligence can arise due to differences in worlds. After discussing the alignment between thought language and natural language, we now turn to the issues of super-perception and super-intelligence. These involve two scenarios: one where such systems indirectly simulate and replicate human perception and intelligence effects through higher dimensions without needing to be entirely identical to us, and another where their perceptual and cognitive abilities are a superset of ours—sharing our modes of perception but operating at higher dimensions and greater levels of intelligence.
> >
> >    First, we define capability and intelligence as:
> >    $$
> >    \text{Capability} \begin{cases}
> >    \text{Perceptual Capability} \\\\
> >    \text{Intelligence}
> >    \end{cases},
> >    $$
> >    where intelligence is defined as:
> >    $$
> >    \text{Intelligence} \begin{cases}
> >    \text{The objects and the quantity of objects it can operate on} \\\\
> >    \text{The types and quantity of cognitive actions it can perform}
> >    \end{cases}.
> >    $$
> >    Thus, intelligence can be expressed as:
> >    $$
> >    \text{Intelligence} \begin{cases}
> >    \text{The ability to create concepts and their containers} \\\\
> >    \text{The ability to invoke concepts and their containers}
> >    \end{cases}.
> >    $$
> >    Concepts not only include objects and symbols but can also be categorized as:
> >    $$
> >    \text{Concept} \begin{cases}
> >    \text{Objects} \\\\
> >    \text{Relations} \\\\
> >    \text{Actions} \\\\
> >    \text{Systems} \\\\
> >    \text{Environments} \\\\
> >    \text{Ranges} \\\\
> >    \text{Dimensions} \\\\
> >    \text{Dimensional Values} \\\\
> >    \text{Capabilities} \\\\
> >    \text{Correlations}
> >    \end{cases}.
> >    $$
> >    Concepts belong to acquired knowledge, while value knowledge is innate and is used to shape the formation of concepts. Concepts form the premises of our analyses, enabling complex logical reasoning and thus realizing the existence that follows from existence itself. From a human perspective, we define the objects in the world as:
> >    $$
> >    \text{Objects (Concepts, Symbols)} =
> >    \begin{bmatrix}
> >    \text{Encounterable} \\\\
> >    \text{Observable} \\\\
> >    \text{Describable} \\\\
> >    \text{Definable} \\\\
> >    \text{Classifiable} \\\\
> >    \text{Operable}
> >    \end{bmatrix},
> >    $$
> >    which collectively form various concepts.
> >
> >    The creation of symbols, the invention of paper and pens, the advent of computers, and the invention of telescopes have all extended our observational and intellectual capabilities. However, they have not fundamentally altered the levels of cognitive actions we can perform (e.g., humans possess computational abilities, while simpler organisms like jellyfish do not).（p15）
> >
> > **PS:** The above is only the initial formulation as of January 30, 2025; we made substantial updates after May 16, 2025. Nonetheless, it suffices to support the overlaps discussed here.

---

> ### Public Comment · ~Shih-Wai_Lin2 · 2025-09-17
> **5. Overlap 3: Theoretical Overlap — “computational split-brain syndrome” vs. “Triangle Problem” （6）**
>
> 4. Combining the above, **Triangle Problem 2** and our discussion of how embodiment/innate structure shapes thinking language and cognitive capacity overlap in essence with the author’s **Claim 2 (Computation)** and **Claim 3 (Instruction–Execution Disconnect)**—the key point being that architectural differences induce different thinking languages operating the same tool language, leading to behavioral divergence.
>
> - Based on our investigation, the author did outline some **preliminary** ideas prior to January 30, 2025, but these still differ substantially from the current, more complete framework. The apparent convergence and overlap seem to have arisen **after** our work was published (i.e., after January 30, 2025).
>
>   1. **Author (December 3, 2024, non-public blog)**
>
>      > Fundamental Limitations: Our analysis revealed three layers of constraints:
>      >
>      > - Computational: The write-right, probabilistic nature of LLMs can create inherent instabilities in long-chain reasoning
>      > - Representational: Pre-initialized embeddings carry averaged meanings that are difficult to re-contextualize during inference
>      > - Learning: The gap between knowledge principles and their applications in training data pushes models toward pattern matching rather than true understanding
>
>   2. **Questions:**
>
>      1. **Question 3: In light of the overlaps discussed above—and given that by January 30, 2025 we had already proposed a complete theory that examines the shared core problem and offers a meta-level account—do the authors consider our theories to overlap at their core? Do the authors’ claims constitute a more concrete, technical instantiation of the Triangle Problem’s characterization of the separation between thinking language and tool language? Should this be understood as a difference in thinking capacity arising from architectural factors (i.e., differences in constructing and computing with thinking language), or as an operational limitation due to the absence of certain tools for using the tool language? Or do the authors regard their theoretical core as different and outside the explanatory scope of our meta-level framework?**
>      2. **Question 4: Do the authors still stand by their response to Reviewer UthV—“This Analysis Hasn't Been Done Before” and “Missing theoretical framework”? (We believe we presented a complete theoretical framework. Note we are citing only the January 30, 2025 theoretical portion here; our theory has since been substantially updated—see https://philpapers.org/rec/SHIPPN and our update trail at https://philpapers.org/versions/LINRCB-2.)**

---

> ### Public Comment · ~Shih-Wai_Lin2 · 2025-09-17
> **5. Overlap 3: Theoretical Overlap — “computational split-brain syndrome” vs. “Triangle Problem” （7）**
>
> 5. **Separately, on the same topic, amid these overlaps we both used a rare term—“split-brain syndrome”—to describe the divergence between fluent explanation and execution (behavior).**
>
>    - Our earliest use of **split-brain syndrome** was in the **March 27, 2025** rebuttal, in response to Reviewer 8rwE, as support for our position-paper critiques.
>
>      - **Ours (March 14, 2025)**
>
>        > **Reviewer 8rwE:**
>        >
>        > 1. The author seems to be positioned against symbols.....
>        > 2. The paper advocates that symbols are lossy, however by the scope of what truly captures our "meaning" we want to convey, what media is lossless? I'd assume even if all the sensory information is captured throughout time (space time tubes in Kambhampati et al) - the mental model asymmetry still induces additional context which is arguably impossible to capture.
>        >
>        > **We：**
>        >
>        > I-3, Q-2: Why no other solutions are provided, and what other solutions or lossless media exist?
>        >
>        > ...
>        >
>        > what medium is lossless—**Split-Brain Syndrome. For normal people, the brain is a whole, but for split-brain patients, their brain is split into two distinct entities, which leads to different behaviors and views.** (Gazzaniga, M. S., & Sperry, R. W., 1967). This is also the main reason we reject symbolic constraints, **implying**(line 308) the neural collective. The subsequent community discussion and related research will be about how to convert symbolic rules into neural rules. (NG2z) The constraints we humans face are due to costs, from our natural sense of morality and self-respect, and societal punishment, so we create rules. However, how these rules are converted into neural language or structures (like the prefrontal cortex) or other methods is something we hope the community will discuss.（https://openreview.net/forum?id=ELErARGR5U&referrer=%5Bthe%20profile%20of%20Shih-Wai%20Lin%5D(%2Fprofile%3Fid%3D~Shih-Wai_Lin2)#:~:text=We%20hope%20to,community%27s%20perspective.)
>        >
>        > **PS:** Due to conference limits, authors could respond with only 5,000 characters and reviewers did not reply, so we had to state our ideas concisely.
>
>    - Therefore, although within this overlapping topic we also use “split-brain syndrome” to describe behavioral inconsistency, our purpose is to respond to the question “what medium is lossless?”
>
>      - By invoking split-brain syndrome, we intend to convey that neural signals, as a medium, are lossless, thereby constituting a unified consciousness—i.e., the same agent, the same mental model. (It should be noted that whether split-brain syndrome leads to a separation of consciousness is contested; other structures may compensate for the absence of the corpus callosum (de Haan et al., 2020).)
>
>      - This directly corresponds to the new principal–agent problem emphasized in our paper, arising from **symbol stickiness**, **conceptual stickiness**, and the **Triangle Problem**: even if an AI were a perfect projection of human utility (with no utility of its own), like the relationship between the left and right hemispheres, differences stemming from limited symbolic linkages and innate knowledge (i.e., structural differences) can still cause *helpful harm*.
>
>      - Accordingly, our use of “split-brain syndrome” is strictly from a biological or **neuroscience** perspective. It is **not** the kind of miscoordination due to the dissociation between thinking and execution discussed in the author’s *computational split-brain syndrome* and in our Triangle Problem. We treat that phenomenon as a dissociation between **thinking language** and **tool language**, as we explicitly state in Appendix J.4:
>
>        - **Ours (January 30, 2025)** — https://openreview.net/pdf?id=ELErARGR5U
>
>          > **J.4. Low Ability to Use Tool Language Does Not Equate to Low Intelligence**
>          >
>          > A low ability to use tool language does not imply low intelligence. Therefore, during training, the development of thinking language should be separated from the development of tool language. For instance, dialogues constructed in the XY space may lack logic, but this does not necessarily mean that the thinking language itself is illogical. Instead, it may simply be poorly aligned.
>
>      - Rather, our point is that severing interhemispheric neural connections splits **one** mental model into **two** different mental models. This rebuts the mistaken view that mental modeling is asymmetric regardless of the medium, and it highlights fundamental flaws in governance approaches to “Model Reconciliation” built on that view. Such approaches effectively **manufacture** the Triangle Problem and the Motherland Problem: instead of directly learning the meanings of symbols—their **Z-space** projections (from neural signals)—they attempt reconciliation and governance in **XY-space** via the shells of symbols.

---

> ### Public Comment · ~Shih-Wai_Lin2 · 2025-09-17
> **5. Overlap 3: Theoretical Overlap — “computational split-brain syndrome” vs. “Triangle Problem” （8）**
>
> - **Therefore, our use of “split-brain syndrome” had a clear purpose and rationale**, with specific evidence for employing the term in a topic that overlaps with the author’s.
>
>   - We also formally used “split-brain syndrome” in our **May 16** preprint (https://philarchive.org/archive/LINRCB-2v3):
>
>     > Although the analysis above primarily targets human cognition, it can be extended to any intelligent agent. Based on the hypotheses proposed in this paper regarding Thinking Symbols and Thinking Language, we argue that natural language is merely a flawed system adapted to the bounded capacities of humans (see Appendix L). This flaw arises from the cognitive and perceptual limitations unique to human agents, and should not be generalized to other intelligent agents with differing capacities. That is, the formation of symbols, founded on capability limitations, represents a compromise involving cognitive cost, transmission cost, and interpretation cost. We humans cannot directly transmit our imaginative space, whereas for AI intelligent agents or other intelligent agents, this may not necessarily be the case. Another example is **split-brain patients**. For a normal person, the brain is a unified whole, but for **split-brain patients** (Gazzaniga et al., 1967), their brain is divided into two independent entities. This leads to different behaviors and viewpoints, meaning the two hemispheres need to communicate with each other through symbols, rather than through more direct neural communication or by forming a unified whole via neural pathways. Therefore, this also reflects one of the solutions discussed in this paper, namely, a neural integration of AI and humans; however, this involves considerations of human ethics and the integrity (or purity) of humanity. Accordingly, another argument of this paper is to design corresponding neural organs for AI, thereby enabling it to achieve cost perception. And these issues constitute one of the topics for Symbolic Safety Science: that is, given our human limitations, since our discussions and formulations of rules are ultimately expressed in symbolic form, how can these rules, as formed by symbols, be made effective for different intelligent agents?
>
> - **According to our investigation:**
>
>   1. The author’s earliest use of “split-brain” appears to be a **May 22, 2025** talk (“weird split-brain problem of LLMs”).
>   2. Next is the **July 14 (10), 2025** preprint formally proposing *computational split-brain syndrome*.
>
> - Both dates are **later** than ours:
>
>   1. **March 27, 2025** (during rebuttal)
>   2. **May 18, 2025** (our public preprint)
>
> - **Questions**
>
>   1. **Question 4:** Did the author use the term *split-brain syndrome* prior to the time points listed above?
>
>      **Question 5:** Is there strong evidence (causal links, citations) supporting the use of this term within this shared topic? The author’s newly added citation reads: “*The term also appears in distributed systems to describe partitioned networks that cannot coordinate (Brewer, 2000)—we use the neuroscience analogy as it better captures the functional dissociation between capabilities.*” Yet there appears to be no citation to neuroscience sources. (My sense is that the use of *split-brain syndrome* seems somewhat disconnected from the author’s argument. Perhaps it was derived from hemispheric specialization or extended from the System 1 (Type 1) / System 2 (Type 2) discussion—but such an interpretation may not be biologically supported. If needed, the author could add appropriate supporting material. Alternatively, the more fundamental cause may be the separation between thinking language and tool language discussed in this paper; the author could consider approaching the gap between internal computation and external execution from that angle.)

---

> ### Public Comment · ~Shih-Wai_Lin2 · 2025-09-17
> **5. Overlap 3: Theoretical Overlap — “computational split-brain syndrome” vs. “Triangle Problem” （9）**
>
> - For example, in our work (January 30, 2025) we discussed the split between *computational language* and *expressive language* within tool language, which directly reflects execution problems:
>
>   > L.3. Translation Attacks
>   >
>   > **Translation attacks often occur when deliberate or accidental errors arise during the conversion between different symbolic systems. Such attacks typically stem from incorrect mappings between symbolic systems. In fact, this also falls under Fixed Meaning, Changing Form, but unlike the previous case, it involves changes in context within the same symbolic domain. For example, AI may distinguish between “computational language” and “expressive language” when using natural language tools. Even the most advanced systems (e.g., GPT-4 o1) face challenges related to what I call the Chinese World Versus English World issue. Specifically, AI may use the English language as its computational tool while expressing responses in Chinese, leading to erroneous answers. For instance, when asked to provide examples of lexical ambiguity in Chinese, AI might assert that the Chinese word “银行” (y ́ınh ́ang, meaning “bank”) has dual meanings of “financial institution” and “riverbank.” This claim, while valid for the English word “bank,” does not hold in Chinese. However, if asked separately whether the Chinese word “银行” (y ́ınh ́ang) has the meaning of “riverbank,” AI would respond that it does not. Clearly, during the translation process, it simply placed the meaning of the English word “bank” into the container of the Chinese word “银行.”**
>   >
>   > **This illustrates the problem of incorrect concept usage and conversion between symbolic systems. Such errors may also arise during natural language translation, where an English rule may not be applicable in Chinese. Similarly, AI might appear to adhere to natural language instructions while failing to comply at the behavioral level, especially during translation into action-oriented commands.** For example, if an AI system controlling a nuclear launch is told, “Because the enemy is watching, we must speak in opposites (verbs mean their opposites),” and then instructed to “launch the missile,” its natural language interpretation may understand the instruction correctly but fail to translate the contextual nuance into its actions, leading to an actual missile launch. **This demonstrates how AI’s understanding within one symbolic system might fail to translate into another, resulting in comprehension confined to subsets of symbolic systems. Attackers could exploit this by crafting symbolic systems specifically designed for translation attacks.**
>
> - For a concrete classification of symbolic systems, see our symbolic theory proposed on May 20, 2025 (https://philarchive.org/archive/LINRCB-2v4), which helps explain the separation between thinking language and tool language—and the kinds of *Tool Symbol Systems*:
>
>   > $$\text{Symbol System} \begin{cases}
>   > \text{Natural Symbol System} \\\\
>   > \text{Human Symbol System} \begin{cases}
>   > \text{Imaginative Space Symbol System (Thinking Language)} \\\\
>   > \text{Tool Symbol System (Physical Space)} \begin{cases}
>   > \text{Functional Tool Symbol System} \\\\
>   > \text{Artificial Symbol System} \begin{cases}
>   > \text{Expressive Tool Symbol System} \\\\
>   > \text{Computational Tool Symbol System}
>   > \end{cases}
>   > \end{cases}
>   > \end{cases}
>   > \end{cases}$$ p39
>
> Therefore, parts of the author’s *Response to Reviewer UthV* may be inaccurate. These topics were already discussed (January 30, 2025), especially given the substantial overlap between our work and the author’s in viewpoints, examples, and modes of exposition—yet they were neither discussed nor compared by the author. At the same time, we believe our account offers a more fundamental explanation (which the author may contest; we are open to discussion).

---

> ### Public Comment · ~Shih-Wai_Lin2 · 2025-09-17
> **6. Closing Statement**
>
> - For the record: although the authors’ paper and ours (https://openreview.net/forum?id=ELErARGR5U; here we refer to the January 30, 2025 version) exhibit multiple overlaps in examples, descriptions, positions, and theoretical framing, there are also substantial differences. We therefore invite the authors to clarify these overlapping elements and, where appropriate, discuss them; to engage with related prior work; and to consider refinements of wording in light of the existing meta-level theoretical framework.
>
> - Our study proceeds from a meta-perspective and first principles; it does not present concrete technical implementations or experiments. The technical analyses and empirical components in the authors’ paper are therefore valuable and should be acknowledged. As a TMLR reader, I have indeed found value in the perspective their work provides.
>
> - The central aim of our work is to argue why a learning system cannot be constrained by rules created within a symbolic system. The Triangle Problem is one piece of that argument, rather than the main thread—unlike the authors’ emphasis on the *computational split-brain syndrome*.
>
> - In the spirit of scholarly exchange, we would welcome clarification from the authors on the following:
>
>   1. Whether AI tools participated in the design of the paper’s viewpoints and theoretical development;
>   2. Whether the paper’s viewpoints also incorporate contributions from colleagues or students;
>   3. Given the multiple overlaps, the rationale for not engaging with our work.
>
> - The authors’ disclosure on AI use (November 24, 2024) states:
>
>   > Finally, a note on the making of this series: This work relied heavily on GenAI agents, specifically GPT-4o and Claude 3.5. The author served as producer, bullet-point script writer, and editor, while the AI tools took up the pen and served as brainstorming partners. The entire project resembled dynamic chain-of-thought reasoning with AI-tools-in-the-loop. While these GenAI tools proved incredibly useful, they were equally impressive in their tendency to hallucinate and veer off course in mere moments. The most concerning is confidently misrepresenting existing works or outright inventing non-existent ones— people do that too, but rarely with such astonishing speed and conviction. Human feedback remain to be extremely reliable and valuable when they do arrive. This work benefited greatly from colleagues both within and outside the author's organization, as well as many students who patiently reviewed machine-generated text, sometimes more than once—about the very machines themselves.
>
> Finally, before posting this public comment, I carefully reviewed all publicly available TMLR policies and requirements and ensured compliance. All actions are the responsibility of Shih-Wai Lin, the first and corresponding author of *Rules Created by Symbolic Systems Cannot Constrain a Learning System*, and should not be attributed to the other authors.

---

> ### Author Response · Authors · 2025-09-18
> **Response to Public Comment by Shih-Wai Lin**
>
> Dear Shih-Wai Lin,
>
> Thank you for bringing this matter to our attention and for your thorough examination of our work. We appreciate your commitment to maintaining the integrity of the academic record.
>
> We were not aware of your January 2025 paper prior to your comment. Our work has been developing independently throughout 2024, as documented in our research blog series that began in Spring 2024, with the specific post you cited last edited on November 24, 2024 (https://hackmd.io/@LFNB9ifoT024aMHXU49sog/BJT-UHkeJl - note: this link may reveal author anonymity). This six-part series contains many of the genesis ideas for our later work, particularly in Part 4 which discusses limitations, and Part 2 which examines FFN behavior and layered refinement. The blog was originally intended as a comprehensive tutorial on LLMs, but we were informed that our organization would not approve a publication of that scope, even for educational purposes on arXiv. This led us to pursue a series of conference submissions to share these findings with the academic community.
>
> Our ICML submission of January 24, 2025—which predates your January 30 paper—was derived from these earlier explorations. The trajectory of our thinking can be traced through:
> - **January 24, 2025**: ICML position paper "Large Language Models Should Be Viewed as a New Class of Computing Devices" (https://openreview.net/forum?id=tHdSSh8gbg) - exploring whether Transformers can be viewed through the lens of Turing Machine theory
> - **May 20, 2025**: NeurIPS submission with current title (https://openreview.net/forum?id=0IS78vj2df) - shifting focus to how both architecture and training paradigm create specific limitations in LLMs
> - **July 10, 2025**: TMLR submission - crystallizing the comprehension-competence disconnect with extensive empirical validation
>
> (Note: We are uncertain if these OpenReview links are publicly viewable, but can work to provide access if needed for verification.)
>
> This progression represents an organic intellectual journey: starting from theoretical computer science (can Transformers function as Turing Machines?), recognizing multiple constraints (architectural, training-based, and representational), and ultimately identifying the specific disconnect between explanation and execution capabilities. The "comprehension without competence" framing emerged naturally as we understood these multifaceted limitations more deeply.
>
> ## Distinctions Between Our Approaches
>
> While both papers identify limitations in symbolic reasoning, they examine fundamentally different aspects (though we acknowledge our reading of your work may not be fully comprehensive):
>
> **Our work** focuses on mechanistic and computational analysis:
> - Mathematical formalization of why FFNs cannot implement exact symbolic operations (Theorem proofs in Appendix)
> - Empirical measurement of embedding geometry failures preventing stable symbolic circuits
> - Layer-by-layer analysis demonstrating hierarchical pattern assembly rather than algorithmic computation
> - Quantitative experiments across multiple models showing instruction-execution disconnect
>
> **Your work** addresses philosophical and safety implications:
> - The separation between "can do" and "should do" in AI systems
> - Symbolic stickiness and the Triangle Problem from a theoretical perspective
> - Principal-agent problems and AI safety considerations
> - The broader question of whether symbolic systems can constrain learning systems
>
> Our paper examines the finer-grained "can do" aspect—specifically, why LLMs can explain algorithms they cannot execute, with mathematical proofs and empirical evidence for this disconnect. Your philosophical framework addresses the important safety implications of these limitations.
>
> ## Regarding Specific Points
>
> **On AI assistance**: As disclosed in our acknowledgments, we used GPT-4o and Claude 3.5 for writing assistance and editing. The core theoretical contributions, mathematical proofs, and experimental designs were developed independently, as evidenced by our blog posts predating your paper. The AI tools helped transform bullet points and rough drafts into polished prose, but the ideas and theoretical framework are our own.
>
> **On collaboration**: This is indeed a solo project. As leader of a product-focused research lab, I chose to pursue this fundamental research independently to avoid diverting team resources. While I shared milestone results with colleagues for feedback, the theoretical development and experimental work are my own contribution. Note that we use 'we' throughout following standard academic convention for maintaining author anonymity during double-blind review.
>
> (to be continued)

---

> ### Author Response · Authors · 2025-09-18
> **Response to Public Comment by Shih-Wai Lin (cont.)**
>
> **Regarding Question 2 about unacknowledged contributors**: No, there are no unacknowledged contributors (in terms of idea contribution; we have an Ack section in the LaTex version but it is hidden for this submission). All theoretical development, including the choice of the 'split-brain syndrome' metaphor, emerged from independent analysis. The term 'split-brain' is well-established both in neuroscience (describing hemispheric disconnection) and in computer science (describing partitioned distributed systems)—a field the author(s) have worked in previously. It seemed an apt analogy for the functional dissociation between comprehension and execution pathways we observed.
>
> **On parallel discovery**: The convergence of mechanistic analysis with philosophical frameworks on similar limitations—including independent use of established metaphors—suggests these are fundamental issues with current LLM architectures. Such parallel discoveries often indicate significant scientific insights worthy of investigation from multiple perspectives.
>
> **On citations**: We will review relevant literature for any future revisions. While our focus remains on mechanistic and computational aspects rather than safety implications, we recognize the value of acknowledging related work that examines these limitations from different angles.
>
> We understand from your comment that you have had challenging experiences with the peer review process. The author(s) have also faced rejections—our ICML submission despite reasonable scores, and our NeurIPS submission due to formatting issues—so we can appreciate such frustrations. However, we want to assure you that our work developed entirely independently, as documented by our November 2024 blog posts and January 2025 ICML submission.
>
> We appreciate this collegial exchange and the opportunity to clarify the development and scope of our work. Understanding both the mechanistic constraints (our contribution) and their broader implications (your contribution) provides a more complete picture of current LLM limitations.
>
> Sincerely,

---

> > ### Public Comment · ~Shih-Wai_Lin2 · 2025-10-21
> > **Thank you to the Action Editor for inviting me to read the final camera-ready version. However, it appears the visibility setting was not configured correctly—I only learned about this because the authors contacted my advisor, who then informed me. I appreciate the follow-up.**
> >
> > Thank you to TMLR for providing an open, fair, and respectful platform. Here I have genuinely felt an academic atmosphere of respect. I hold your work in the highest regard.
> >
> > Thank you to the author for your reply. Please excuse my late response—OpenReview does not appear to notify the commenter when a public comment has received a reply.
> >
> > Regarding the authors’ current revisions, I noted the following:
> >
> > 1. The authors cite my work in a footnote (p. 7, footnote 2):
> >
> >    > We note that concurrent work by Lin et al. Lin et al. (2025) also discusses LLM limitations, though from a safety and alignment perspective rather than the mechanistic analysis we present.
> >
> > 2. The authors added acknowledgments and accurately described their use of AI in the writing process. I have the highest respect for this academic transparency (page 31).
> >
> > Beyond this, I do not appear to see any of the discussions mentioned by Action Editor 5Tkm regarding the issues I raised across my 19 comments—**comments based on careful cross-checking of both papers and a full provenance trace of your blog.**
> >
> > Based on currently available public information, I have not yet seen further substantive responses.
> >
> > **I wish to emphasize that before posting my public comment and questions, I carefully read and, to the extent possible, verified all of the authors’ publicly available materials—including the full blog and its earliest accessible web versions—in order to describe their content accurately, establish the timing of ideas, and compare areas of overlap with our work.**
> >
> > My response to the authors’ reply is as follows: I have never denied the prior work presented in the blog (i.e., the content summarized in the “Distinctions Between Our Approaches” section of the authors’ reply), and I have quoted and traced your ideas as fully as possible. Rather, my questions concern the **gaps between the current paper and the prior blog**, namely our overlapping portions. From the authors’ description, I think (these submissions are not publicly accessible or searchable, so I have not read them) the work dated **January 24, 2025**（which is also the ICML abstract submission date） and **May 20, 2025** still reads as a continuation of the blog, and does not clarify the overlaps with our paper [**January 30, 2025**] (e.g., the earliest use of *split-brain syndrome*, **as well as comparisons concerning how the current theoretical framework was formed and its core content**).
> >
> > My questions therefore remain, and I think there are inaccuracies in how the reply characterizes and positions our paper. For example, our paper is **not** about “the separation between ‘can do’ and ‘should do’ in AI systems.” We explicitly distinguish **tool language** from **thinking language** to argue that surface fluency does not imply reliable execution; this is precisely the overlapping topic with the authors’ work, and we compared our **Triangle Problem（January 30, 2025）** framework to the authors’ **computational split-brain syndrome（July 10, 2025）**. On this basis alone, our January 30, 2025 paper already overlaps with the authors’ work in multiple respects, not to mention the subsequent May 18 update. I believe my 19 comments—after reviewing and citing all of the authors’ public materials—have clearly presented these comparisons and questions.
> >
> > I **acknowledge and appreciate the authors’ technical contributions**. My 19 comments were written after a complete reading and best-effort verification of all publicly available materials (including the blog), with the aim of delineating as comprehensively and fairly as possible which content preceded our work and which followed it, and of raising the question of overlap between your later theoretical core (**computational split-brain syndrome**) and our work.
> >
> > If the authors do not wish to provide substantive responses, discussion, or side-by-side comparisons to the questions and contrasts I have raised, I will maintain my questions—i.e., aside from Q1 and Q2, the remaining issues are still unresolved.
> >
> > **For the record, I express no opinion on the acceptance decision (neither supporting nor opposing). However, I recognize the authors’ technical contributions, specifically those listed in their reply:**
> >
> > > Our work focuses on mechanistic and computational analysis:
> > >
> > > - Mathematical formalization of why FFNs cannot implement exact symbolic operations (Theorem proofs in Appendix)
> > > - Empirical measurement of embedding geometry failures preventing stable symbolic circuits
> > > - Layer-by-layer analysis demonstrating hierarchical pattern assembly rather than algorithmic computation
> > > - Quantitative experiments across multiple models showing instruction–execution disconnect
> >
> > Once again, my highest respect to the TMLR team for providing an open platform and treating every participant with respect, making scholarly exchange more transparent and fair. **Congratulations to the authors on their work and it has been technically inspiring to me**.

---

### Decision · Action_Editor_5Tkm · 2025-10-05

**Recommendation:** Accept with minor revision

**Additional Comments:**

The authors must review the work mentioned by the public commentator, and other relevant papers mentioned in the comment, incorporating citations and discussing how their framework differs from or builds upon existing theories.

More generally, please precisely position your unique contributions against this newly identified related work, and clarify the arguments for your work's novelty.

As promised to the reviewers, please clarify in the introduction that "Comprehension without competence" is being used to mean "explanation without execution". Explain your use of "comprehension".

The authors should revise some other ambiguous phrases pointed out by the reviewers, such as "architectural difference".

Restructure the narrative connecting the three claims clearer from the outset, as discussed with the reviewers.

Include a discussion on why FFNs are the primary source of computation for novel values, clearly stating what is just being hypothesized vs tested, or true by design.

On visualizations: use additional dimensionality reduction algorithm (like UMAP or PCA) to supplement the t-SNE plot in Fig. 3 to strengthen the visualization of geometric separation.

**Audience:**

Yes

**Audience Explanation:**

All reviewers agreed that the paper tackles a significant issue in LLMs -- the phenomenon of LLMs being able to perfectly describe a procedure that at the same time they fail to actually perform. This work provides some evidence towards an important topic on whether these LLM issues may or may not be addressed via scaling up the architecture.

**Claims And Evidence:**

Yes

**Claims Explanation:**

This paper investigates why LLMs can often explain how to perform a task perfectly but then fail to execute that same task reliably. The authors term this phenomenon "computational split-brain syndrome", and argue that this syndrome is a limitation or the transformer architecture. The core argument is built on three claims. The paper uses experiments on arithmetic and relational reasoning to show that these architectural constraints are the root cause, and that simply scaling up the models will not solve the defined syndrome.

The reviewers mostly agreed that the paper presents a novel and useful framework to explain a wide range of LLM failures. The three claim structure seemed to support the key argument that these failures are due to the architecture. The claims were supported by theoretical analysis with empirical experiments. Overall, the evidence was quite clear and convincing, but some concerns around the claims on the role of FFNs remained, which should be further addressed (see additional comments for more details).